# ANALYTIC-DPM: AN ANALYTIC ESTIMATE OF THE OPTIMAL REVERSE VARIANCE IN DIFFUSION PROBABILISTIC MODELS

**Fan Bao**[1] [*], **Chongxuan Li**[2 3] [†], **Jun Zhu**[1] [†], **Bo Zhang**[1]
[1]Dept. of Comp. Sci. & Tech., Institute for AI, Tsinghua-Huawei Joint Center for AI
BNRist Center, State Key Lab for Intell. Tech. & Sys., Tsinghua University, Beijing, China
[2]Gaoling School of Artificial Intelligence, Renmin University of China, Beijing, China
[3]Beijing Key Laboratory of Big Data Management and Analysis Methods , Beijing, China
bf19@mails.tsinghua.edu.cn,chongxuanli1991@gmail.com,
{dcszj, dcszb}@tsinghua.edu.cn

## ABSTRACT

Diffusion probabilistic models (DPMs) represent a class of powerful generative models. Despite their success, the inference of DPMs is expensive since it generally needs to iterate over thousands of timesteps. A key problem in the inference is to estimate the variance in each timestep of the reverse process. In this work, we present a surprising result that both the optimal reverse variance and the corresponding optimal KL divergence of a DPM have analytic forms w.r.t. its score function. Building upon it, we propose *Analytic-DPM*, a training-free inference framework that estimates the analytic forms of the variance and KL divergence using the Monte Carlo method and a pretrained score-based model. Further, to correct the potential bias caused by the score-based model, we derive both lower and upper bounds of the optimal variance and clip the estimate for a better result. Empirically, our analytic-DPM improves the log-likelihood of various DPMs, produces high-quality samples, and meanwhile enjoys a $20\times$ to $80\times$ speed up.

## 1 INTRODUCTION

A diffusion process gradually adds noise to a data distribution over a series of timesteps. By learning to reverse it, diffusion probabilistic models (DPMs) (Sohl-Dickstein et al., 2015; Ho et al., 2020; Song et al., 2020b) define a data generative process. Recently, it is shown that DPMs are able to produce high-quality samples (Ho et al., 2020; Nichol & Dhariwal, 2021; Song et al., 2020b; Dhariwal & Nichol, 2021), which are comparable or even superior to the current state-of-the-art GAN models (Goodfellow et al., 2014; Brock et al., 2018; Wu et al., 2019; Karras et al., 2020b).

Despite their success, the inference of DPMs (e.g., sampling and density evaluation) often requires to iterate over thousands of timesteps, which is two or three orders of magnitude slower (Song et al., 2020a) than other generative models such as GANs. A key problem in the inference is to estimate the variance in each timestep of the reverse process. Most of the prior works use a handcrafted value for all timesteps, which usually run a long chain to obtain a reasonable sample and density value (Nichol & Dhariwal, 2021). Nichol & Dhariwal (2021) attempt to improve the efficiency of sampling by learning a variance network in the reverse process. However, it still needs a relatively long trajectory to get a reasonable log-likelihood (see Appendix E in Nichol & Dhariwal (2021)).

In this work, we present a surprising result that both the optimal reverse variance and the corresponding optimal KL divergence of a DPM have analytic forms w.r.t. its score function (i.e., the gradient of a log density). Building upon it, we propose *Analytic-DPM*, a training-free inference framework to improve the efficiency of a pretrained DPM while achieving comparable or even superior performance. Analytic-DPM estimates the analytic forms of the variance and KL divergence using the Monte Carlo method and the score-based model in the pretrained DPM. The corresponding trajectory is calculated via a dynamic programming algorithm (Watson et al., 2021). Further, to

---

[*]Work done during an internship at Huawei Noah's Ark Lab. [†]Correspondence to: C. Li and J. Zhu.

correct the potential bias caused by the score-based model, we derive both lower and upper bounds of the optimal variance and clip its estimate for a better result. Finally, we reveal an interesting relationship between the score function and the data covariance matrix.

Analytic-DPM is applicable to a variety of DPMs (Ho et al., 2020; Song et al., 2020a; Nichol & Dhariwal, 2021) in a plug-and-play manner. Empirically, Analytic-DPM consistently improves the log-likelihood of these DPMs and meanwhile enjoys a $20\times$ to $40\times$ speed up. Besides, Analytic-DPM also consistently improves the sample quality of DDIMs (Song et al., 2020a) and requires up to 50 timesteps (which is a $20\times$ to $80\times$ speed up compared to the full timesteps) to achieve a comparable FID to the corresponding baseline.

## 2 BACKGROUND

Diffusion probabilistic models (DPMs) firstly construct a forward process $q(\boldsymbol{x}_{1:N}|\boldsymbol{x}_0)$ that injects noise to a data distribution $q(\boldsymbol{x}_0)$, and then reverse the forward process to recover it. Given a forward noise schedule $\beta_n \in (0,1), n = 1, \cdots, N$, denoising diffusion probabilistic models (DDPMs) (Ho et al., 2020) consider a Markov forward process:

$$q_{\mathrm{M}}(\boldsymbol{x}_{1:N}|\boldsymbol{x}_0) = \prod_{n=1}^{N} q_{\mathrm{M}}(\boldsymbol{x}_n|\boldsymbol{x}_{n-1}), \quad q_{\mathrm{M}}(\boldsymbol{x}_n|\boldsymbol{x}_{n-1}) = \mathcal{N}(\boldsymbol{x}_n|\sqrt{\alpha_n}\boldsymbol{x}_{n-1}, \beta_n \boldsymbol{I}), \quad (1)$$

where $\boldsymbol{I}$ is the identity matrix, $\alpha_n$ and $\beta_n$ are scalars and $\alpha_n := 1 - \beta_n$. Song et al. (2020a) introduce a more general non-Markov process indexed by a non-negative vector $\lambda = (\lambda_1, \cdots, \lambda_N) \in \mathbb{R}_{\geq 0}^N$:

$$q_\lambda(\boldsymbol{x}_{1:N}|\boldsymbol{x}_0) = q_\lambda(\boldsymbol{x}_N|\boldsymbol{x}_0) \prod_{n=2}^{N} q_\lambda(\boldsymbol{x}_{n-1}|\boldsymbol{x}_n, \boldsymbol{x}_0), \quad (2)$$

$$q_\lambda(\boldsymbol{x}_N|\boldsymbol{x}_0) = \mathcal{N}(\boldsymbol{x}_N|\sqrt{\overline{\alpha}_N}\boldsymbol{x}_0, \overline{\beta}_N \boldsymbol{I}),$$

$$q_\lambda(\boldsymbol{x}_{n-1}|\boldsymbol{x}_n, \boldsymbol{x}_0) = \mathcal{N}(\boldsymbol{x}_{n-1}|\tilde{\boldsymbol{\mu}}_n(\boldsymbol{x}_n, \boldsymbol{x}_0), \lambda_n^2 \boldsymbol{I}),$$

$$\tilde{\boldsymbol{\mu}}_n(\boldsymbol{x}_n, \boldsymbol{x}_0) = \sqrt{\overline{\alpha}_{n-1}}\boldsymbol{x}_0 + \sqrt{\overline{\beta}_{n-1} - \lambda_n^2} \cdot \frac{\boldsymbol{x}_n - \sqrt{\overline{\alpha}_n}\boldsymbol{x}_0}{\sqrt{\overline{\beta}_n}}.$$

Here $\overline{\alpha}_n := \prod_{i=1}^{n} \alpha_i$ and $\overline{\beta}_n := 1 - \overline{\alpha}_n$. Indeed, Eq. (2) includes the DDPM forward process as a special case when $\lambda_n^2 = \tilde{\beta}_n$, where $\tilde{\beta}_n := \frac{\overline{\beta}_{n-1}}{\overline{\beta}_n}\beta_n$. Another special case of Eq. (2) is the denoising diffusion implicit model (DDIM) forward process, where $\lambda_n^2 = 0$. Besides, we can further derive $q_\lambda(\boldsymbol{x}_n|\boldsymbol{x}_0) = \mathcal{N}(\boldsymbol{x}_n|\sqrt{\overline{\alpha}_n}\boldsymbol{x}_0, \overline{\beta}_n \boldsymbol{I})$, which is independent of $\lambda$. In the rest of the paper, we will focus on the forward process in Eq. (2) since it is more general, and we will omit the index $\lambda$ and denote it as $q(\boldsymbol{x}_{1:N}|\boldsymbol{x}_0)$ for simplicity.

The reverse process for Eq. (2) is defined as a Markov process aimed to approximate $q(\boldsymbol{x}_0)$ by gradually denoising from the standard Gaussian distribution $p(\boldsymbol{x}_N) = \mathcal{N}(\boldsymbol{x}_N|\boldsymbol{0}, \boldsymbol{I})$:

$$p(\boldsymbol{x}_{0:N}) = p(\boldsymbol{x}_N) \prod_{n=1}^{N} p(\boldsymbol{x}_{n-1}|\boldsymbol{x}_n), \quad p(\boldsymbol{x}_{n-1}|\boldsymbol{x}_n) = \mathcal{N}(\boldsymbol{x}_{n-1}|\boldsymbol{\mu}_n(\boldsymbol{x}_n), \sigma_n^2 \boldsymbol{I}),$$

where $\boldsymbol{\mu}_n(\boldsymbol{x}_n)$ is generally parameterized [1] by a time-dependent score-based model $\boldsymbol{s}_n(\boldsymbol{x}_n)$ (Song & Ermon, 2019; Song et al., 2020b):

$$\boldsymbol{\mu}_n(\boldsymbol{x}_n) = \tilde{\boldsymbol{\mu}}_n\left(\boldsymbol{x}_n, \frac{1}{\sqrt{\overline{\alpha}_n}}(\boldsymbol{x}_n + \overline{\beta}_n \boldsymbol{s}_n(\boldsymbol{x}_n))\right). \quad (3)$$

The reverse process can be learned by optimizing a variational bound $L_{\mathrm{vb}}$ on negative log-likelihood:

$$L_{\mathrm{vb}} = \mathbb{E}_q\left[-\log p(\boldsymbol{x}_0|\boldsymbol{x}_1) + \sum_{n=2}^{N} D_{\mathrm{KL}}(q(\boldsymbol{x}_{n-1}|\boldsymbol{x}_0, \boldsymbol{x}_n)||p(\boldsymbol{x}_{n-1}|\boldsymbol{x}_n)) + D_{\mathrm{KL}}(q(\boldsymbol{x}_N|\boldsymbol{x}_0)||p(\boldsymbol{x}_N))\right],$$

---

[1] Ho et al. (2020); Song et al. (2020a) parameterize $\boldsymbol{\mu}_n(\boldsymbol{x}_n)$ with $\tilde{\boldsymbol{\mu}}_n(\boldsymbol{x}_n, \frac{1}{\sqrt{\overline{\alpha}_n}}(\boldsymbol{x}_n - \sqrt{\overline{\beta}_n}\boldsymbol{\epsilon}_n(\boldsymbol{x}_n)))$, which is equivalent to Eq. (3) by letting $\boldsymbol{s}_n(\boldsymbol{x}_n) = -\frac{1}{\sqrt{\overline{\beta}_n}}\boldsymbol{\epsilon}_n(\boldsymbol{x}_n)$.

which is equivalent to optimizing the KL divergence between the forward and the reverse process:

$$\min_{\{\boldsymbol{\mu}_n, \sigma_n^2\}_{n=1}^N} L_{\text{vb}} \Leftrightarrow \min_{\{\boldsymbol{\mu}_n, \sigma_n^2\}_{n=1}^N} D_{\text{KL}}(q(\boldsymbol{x}_{0:N}) || p(\boldsymbol{x}_{0:N})). \tag{4}$$

To improve the sample quality in practice, instead of directly optimizing $L_{\text{vb}}$, Ho et al. (2020) consider a reweighted variant of $L_{\text{vb}}$ to learn $\boldsymbol{s}_n(\boldsymbol{x}_n)$:

$$\min_{\{\boldsymbol{s}_n\}_{n=1}^N} \mathbb{E}_n \overline{\beta}_n \mathbb{E}_{q_n(\boldsymbol{x}_n)} || \boldsymbol{s}_n(\boldsymbol{x}_n) - \nabla_{\boldsymbol{x}_n} \log q_n(\boldsymbol{x}_n) ||^2 = \mathbb{E}_{n,\boldsymbol{x}_0,\boldsymbol{\epsilon}} || \boldsymbol{\epsilon} + \sqrt{\overline{\beta}_n} \boldsymbol{s}_n(\boldsymbol{x}_n) ||^2 + c, \tag{5}$$

where $n$ is uniform between 1 and $N$, $q_n(\boldsymbol{x}_n)$ is the marginal distribution of the forward process at timestep $n$, $\boldsymbol{\epsilon}$ is a standard Gaussian noise, $\boldsymbol{x}_n$ on the right-hand side is reparameterized by $\boldsymbol{x}_n = \sqrt{\overline{\alpha}_n} \boldsymbol{x}_0 + \sqrt{\overline{\beta}_n} \boldsymbol{\epsilon}$ and $c$ is a constant only related to $q$. Indeed, Eq. (5) is exactly a weighted sum of score matching objectives (Song & Ermon, 2019), which admits an optimal solution $\boldsymbol{s}_n^*(\boldsymbol{x}_n) = \nabla_{\boldsymbol{x}_n} \log q_n(\boldsymbol{x}_n)$ for all $n \in \{1, 2 \cdots, N\}$.

Note that Eq. (5) provides no learning signal for the variance $\sigma_n^2$. Indeed, $\sigma_n^2$ is generally handcrafted in most of prior works. In DDPMs (Ho et al., 2020), two commonly used settings are $\sigma_n^2 = \beta_n$ and $\sigma_n^2 = \tilde{\beta}_n$. In DDIMs, Song et al. (2020a) consistently use $\sigma_n^2 = \lambda_n^2$. We argue that these handcrafted values are not the true optimal solution of Eq. (4) in general, leading to a suboptimal performance.

## 3 ANALYTIC ESTIMATE OF THE OPTIMAL REVERSE VARIANCE

For a DPM, we first show that both the optimal mean $\boldsymbol{\mu}_n^*(\boldsymbol{x}_n)$ and the optimal variance $\sigma_n^{*2}$ to Eq. (4) have analytic forms w.r.t. the score function, which is summarized in the following Theorem 1.

**Theorem 1.** *(Score representation of the optimal solution to Eq. (4), proof in Appendix A.2)*

*The optimal solution $\boldsymbol{\mu}_n^*(\boldsymbol{x}_n)$ and $\sigma_n^{*2}$ to Eq. (4) are*

$$\boldsymbol{\mu}_n^*(\boldsymbol{x}_n) = \tilde{\boldsymbol{\mu}}_n \left( \boldsymbol{x}_n, \frac{1}{\sqrt{\overline{\alpha}_n}} (\boldsymbol{x}_n + \overline{\beta}_n \nabla_{\boldsymbol{x}_n} \log q_n(\boldsymbol{x}_n)) \right), \tag{6}$$

$$\sigma_n^{*2} = \lambda_n^2 + \left( \sqrt{\frac{\overline{\beta}_n}{\alpha_n}} - \sqrt{\overline{\beta}_{n-1} - \lambda_n^2} \right)^2 \left( 1 - \overline{\beta}_n \mathbb{E}_{q_n(\boldsymbol{x}_n)} \frac{|| \nabla_{\boldsymbol{x}_n} \log q_n(\boldsymbol{x}_n) ||^2}{d} \right), \tag{7}$$

*where $q_n(\boldsymbol{x}_n)$ is the marginal distribution of the forward process at the timestep $n$ and $d$ is the dimension of the data.*

The proof of Theorem 1 consists of three key steps:

- The first step (see Lemma 9) is known as the *moment matching* (Minka, 2013), which states that approximating arbitrary density by a Gaussian density under the KL divergence is equivalent to setting the first two moments of the two densities as the same. To our knowledge, the connection of moment matching and DPMs has not been revealed before.

- In the second step (see Lemma 13), we carefully use the *law of total variance* conditioned on $\boldsymbol{x}_0$ and convert the second moment of $q(\boldsymbol{x}_{n-1}|\boldsymbol{x}_n)$ to that of $q(\boldsymbol{x}_0|\boldsymbol{x}_n)$.

- In the third step (see Lemma 11), we surprisingly find that the second moment of $q(\boldsymbol{x}_0|\boldsymbol{x}_n)$ can be represented by the score function, and we plug the score representation into the second moment of $q(\boldsymbol{x}_{n-1}|\boldsymbol{x}_n)$ to get the final results in Theorem 1.

The results in Theorem 1 (and other results to appear later) can be further simplified for the DDPM forward process (i.e., $\lambda_n^2 = \tilde{\beta}_n$, see Appendix D for details). Besides, we can also extend Theorem 1 to DPMs with continuous timesteps (Song et al., 2020b; Kingma et al., 2021), where their corresponding optimal mean and variance are also determined by the score function in an analytic form (see Appendix E.1 for the extension).

Note that our analytic form of the optimal mean $\boldsymbol{\mu}_n^*(\boldsymbol{x}_n)$ in Eq. (6) and the previous parameterization of $\boldsymbol{\mu}_n(\boldsymbol{x}_n)$ (Ho et al., 2020) in Eq. (3) coincide. The only difference is that Eq. (3) replaces the

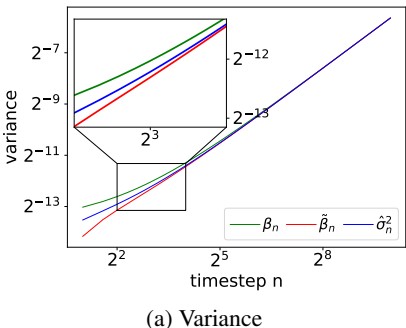 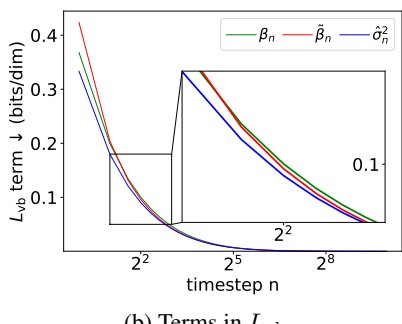

|(a) Variance|(b) Terms in $L_{\mathrm{vb}}$|

Figure 1: Comparing our analytic estimate $\hat{\sigma}_n^2$ and prior works with handcrafted variances $\beta_n$ and $\tilde{\beta}_n$. (a) compares the values of the variance for different timesteps. (b) compares the term in $L_{\mathrm{vb}}$ corresponding to each timestep. The value of $L_{\mathrm{vb}}$ is the area under the corresponding curve.

score function $\nabla_{\boldsymbol{x}_n} \log q_n(\boldsymbol{x}_n)$ in Eq. (6) with the score-based model $\boldsymbol{s}_n(\boldsymbol{x}_n)$. This result explicitly shows that Eq. (5) essentially shares the same optimal mean solution to the $L_{\mathrm{vb}}$ objective, providing a simple and alternative perspective to prior works.

In contrast to the handcrafted strategies used in (Ho et al., 2020; Song et al., 2020a), Theorem 1 shows that the optimal reverse variance $\sigma_n^{*2}$ can also be estimated without any extra training process given a pretrained score-based model $\boldsymbol{s}_n(\boldsymbol{x}_n)$. In fact, we first estimate the expected mean squared norm of $\nabla_{\boldsymbol{x}_n} \log q_n(\boldsymbol{x}_n)$ by $\Gamma = (\Gamma_1, ..., \Gamma_N)$, where

$$\Gamma_n = \frac{1}{M} \sum_{m=1}^{M} \frac{||\boldsymbol{s}_n(\boldsymbol{x}_{n,m})||^2}{d}, \quad \boldsymbol{x}_{n,m} \overset{iid}{\sim} q_n(\boldsymbol{x}_n). \tag{8}$$

$M$ is the number of Monte Carlo samples. We only need to calculate $\Gamma$ once for a pretrained model and reuse it in downstream computations (see Appendix H.1 for a detailed discussion of the computation cost of $\Gamma$). Then, according to Eq. (7), we estimate $\sigma_n^{*2}$ as follows:

$$\hat{\sigma}_n^2 = \lambda_n^2 + \left( \sqrt{\frac{\overline{\beta}_n}{\alpha_n}} - \sqrt{\overline{\beta}_{n-1} - \lambda_n^2} \right)^2 \left( 1 - \overline{\beta}_n \Gamma_n \right). \tag{9}$$

We empirically validate Theorem 1. In Figure 1 (a), we plot our analytic estimate $\hat{\sigma}_n^2$ of a DDPM trained on CIFAR10, as well as the baselines $\beta_n$ and $\tilde{\beta}_n$ used by Ho et al. (2020). At small timesteps, these strategies behave differently. Figure 1 (b) shows that our $\hat{\sigma}_n^2$ outperforms the baselines for each term of $L_{\mathrm{vb}}$, especially at the small timesteps. We also obtain similar results on other datasets (see Appendix G.1). Besides, we show that only a small number of Monte Carlo (MC) samples (e.g., $M$=10, 100) is required to achieve a sufficiently small variance caused by MC and get a similar performance to that with a large $M$ (see Appendix G.2). We also discuss the stochasticity of $L_{\mathrm{vb}}$ after plugging $\hat{\sigma}_n^2$ in Appendix H.2.

### 3.1 BOUNDING THE OPTIMAL REVERSE VARIANCE TO REDUCE BIAS

According to Eq. (7) and Eq. (9), the bias of the analytic estimate $\hat{\sigma}_n^2$ is

$$|\sigma_n^{*2} - \hat{\sigma}_n^2| = \underbrace{\left( \sqrt{\frac{\overline{\beta}_n}{\alpha_n}} - \sqrt{\overline{\beta}_{n-1} - \lambda_n^2} \right)^2 \overline{\beta}_n}_{\text{Coefficient}} \underbrace{|\Gamma_n - \mathbb{E}_{q_n(\boldsymbol{x}_n)} \frac{||\nabla_{\boldsymbol{x}_n} \log q_n(\boldsymbol{x}_n)||^2}{d}|}_{\text{Approximation error}}. \tag{10}$$

Our estimate of the variance employs a score-based model $\boldsymbol{s}_n(\boldsymbol{x}_n)$ to approximate the true score function $\nabla_{\boldsymbol{x}_n} \log q_n(\boldsymbol{x}_n)$. Thus, the approximation error in Eq. (10) is irreducible given a pretrained model. Meanwhile, the coefficient in Eq. (10) can be large if we use a shorter trajectory to sample (see details in Section 4), potentially resulting in a large bias.

To reduce the bias, we derive bounds of the optimal reverse variance $\sigma_n^{*2}$ and clip our estimate based on the bounds. Importantly, these bounds are unrelated to the data distribution $q(\boldsymbol{x}_0)$ and hence can be efficiently calculated. We firstly derive both upper and lower bounds of $\sigma_n^{*2}$ without any assumption about the data. Then we show another upper bound of $\sigma_n^{*2}$ if the data distribution is bounded. We formalize these bounds in Theorem 2.

**Theorem 2.** *(Bounds of the optimal reverse variance, proof in Appendix A.3)*

$\sigma_n^{*2}$ *has the following lower and upper bounds:*

$$\lambda_n^2 \le \sigma_n^{*2} \le \lambda_n^2 + \left( \sqrt{\frac{\overline{\beta}_n}{\alpha_n}} - \sqrt{\overline{\beta}_{n-1} - \lambda_n^2} \right)^2 . \tag{11}$$

*If we further assume $q(\boldsymbol{x}_0)$ is a bounded distribution in $[a, b]^d$, where $d$ is the dimension of data, then $\sigma_n^{*2}$ can be further upper bounded by*

$$\sigma_n^{*2} \le \lambda_n^2 + \left( \sqrt{\overline{\alpha}_{n-1}} - \sqrt{\overline{\beta}_{n-1} - \lambda_n^2} \cdot \sqrt{\frac{\overline{\alpha}_n}{\overline{\beta}_n}} \right)^2 \left( \frac{b-a}{2} \right)^2 . \tag{12}$$

Theorem 2 states that the handcrafted reverse variance $\lambda_n^2$ in prior works (Ho et al., 2020; Song et al., 2020a) underestimates $\sigma_n^{*2}$. For instance, $\lambda_n^2 = \tilde{\beta}_n$ in DDPM. We compare it with our estimate in Figure 1 (a) and the results agree with Theorem 2. Besides, the boundedness assumption of $q(\boldsymbol{x}_0)$ is satisfied in many scenarios including generative modelling of images, and which upper bound among Eq. (11) and Eq. (12) is tighter depends on $n$. Therefore, we clip our estimate based on the minimum one. Further, we show theses bounds are tight numerically in Appendix G.3.

## 4 ANALYTIC ESTIMATION OF THE OPTIMAL TRAJECTORY

The number of full timesteps $N$ can be large, making the inference slow in practice. Thereby, we can construct a shorter forward process $q(\boldsymbol{x}_{\tau_1}, \cdots, \boldsymbol{x}_{\tau_K} | \boldsymbol{x}_0)$ constrained on a trajectory $1 = \tau_1 < \cdots < \tau_K = N$ of $K$ timesteps (Song et al., 2020a; Nichol & Dhariwal, 2021; Watson et al., 2021), and $K$ can be much smaller than $N$ to speed up the inference. Formally, the shorter process is defined as $q(\boldsymbol{x}_{\tau_1}, \cdots, \boldsymbol{x}_{\tau_K} | \boldsymbol{x}_0) = q(\boldsymbol{x}_{\tau_K} | \boldsymbol{x}_0) \prod_{k=2}^K q(\boldsymbol{x}_{\tau_{k-1}} | \boldsymbol{x}_{\tau_k}, \boldsymbol{x}_0)$, where

$$q(\boldsymbol{x}_{\tau_{k-1}} | \boldsymbol{x}_{\tau_k}, \boldsymbol{x}_0) = \mathcal{N}(\boldsymbol{x}_{\tau_{k-1}} | \tilde{\boldsymbol{\mu}}_{\tau_{k-1}|\tau_k}(\boldsymbol{x}_{\tau_k}, \boldsymbol{x}_0), \lambda_{\tau_{k-1}|\tau_k}^2 \boldsymbol{I}), \tag{13}$$

$$\tilde{\boldsymbol{\mu}}_{\tau_{k-1}|\tau_k}(\boldsymbol{x}_{\tau_k}, \boldsymbol{x}_0) = \sqrt{\overline{\alpha}_{\tau_{k-1}}} \boldsymbol{x}_0 + \sqrt{\overline{\beta}_{\tau_{k-1}} - \lambda_{\tau_{k-1}|\tau_k}^2} \cdot \frac{\boldsymbol{x}_{\tau_k} - \sqrt{\overline{\alpha}_{\tau_k}} \boldsymbol{x}_0}{\sqrt{\overline{\beta}_{\tau_k}}} .$$

The corresponding reverse process is $p(\boldsymbol{x}_0, \boldsymbol{x}_{\tau_1}, \cdots, \boldsymbol{x}_{\tau_K}) = p(\boldsymbol{x}_{\tau_K}) \prod_{k=1}^K p(\boldsymbol{x}_{\tau_{k-1}} | \boldsymbol{x}_{\tau_k})$, where

$$p(\boldsymbol{x}_{\tau_{k-1}} | \boldsymbol{x}_{\tau_k}) = \mathcal{N}(\boldsymbol{x}_{\tau_{k-1}} | \boldsymbol{\mu}_{\tau_{k-1}|\tau_k}(\boldsymbol{x}_{\tau_k}), \sigma_{\tau_{k-1}|\tau_k}^2 \boldsymbol{I}).$$

According to Theorem 1, the mean and variance of the optimal $p^*(\boldsymbol{x}_{\tau_{k-1}} | \boldsymbol{x}_{\tau_k})$ in the sense of KL minimization is

$$\boldsymbol{\mu}_{\tau_{k-1}|\tau_k}^*(\boldsymbol{x}_{\tau_k}) = \tilde{\boldsymbol{\mu}}_{\tau_{k-1}|\tau_k} \left( \boldsymbol{x}_{\tau_k}, \frac{1}{\sqrt{\overline{\alpha}_{\tau_k}}} (\boldsymbol{x}_{\tau_k} + \overline{\beta}_{\tau_k} \nabla_{\boldsymbol{x}_{\tau_k}} \log q(\boldsymbol{x}_{\tau_k})) \right),$$

$$\sigma_{\tau_{k-1}|\tau_k}^{*2} = \lambda_{\tau_{k-1}|\tau_k}^2 + \left( \sqrt{\frac{\overline{\beta}_{\tau_k}}{\alpha_{\tau_k|\tau_{k-1}}}} - \sqrt{\overline{\beta}_{\tau_{k-1}} - \lambda_{\tau_{k-1}|\tau_k}^2} \right)^2 (1 - \overline{\beta}_{\tau_k} \mathbb{E}_{q(\boldsymbol{x}_{\tau_k})} \frac{||\nabla_{\boldsymbol{x}_{\tau_k}} \log q(\boldsymbol{x}_{\tau_k})||^2}{d}),$$

where $\alpha_{\tau_k|\tau_{k-1}} := \overline{\alpha}_{\tau_k} / \overline{\alpha}_{\tau_{k-1}}$. According to Theorem 2, we can derive similar bounds for $\sigma_{\tau_{k-1}|\tau_k}^{*2}$ (see details in Appendix C). Similarly to Eq. (9), the estimate of $\sigma_{\tau_{k-1}|\tau_k}^{*2}$ is

$$\hat{\sigma}_{\tau_{k-1}|\tau_k}^2 = \lambda_{\tau_{k-1}|\tau_k}^2 + \left( \sqrt{\frac{\overline{\beta}_{\tau_k}}{\alpha_{\tau_k|\tau_{k-1}}}} - \sqrt{\overline{\beta}_{\tau_{k-1}} - \lambda_{\tau_{k-1}|\tau_k}^2} \right)^2 (1 - \overline{\beta}_{\tau_k} \Gamma_{\tau_k}),$$

where $\Gamma$ is defined in Eq. (8) and can be shared across different selections of trajectories. Based on the optimal reverse process $p^*$ above, we further optimize the trajectory:

$$\min_{\tau_1,\cdots,\tau_K} D_{\mathrm{KL}}(q(\boldsymbol{x}_0, \boldsymbol{x}_{\tau_1}, \cdots, \boldsymbol{x}_{\tau_K}) || p^*(\boldsymbol{x}_0, \boldsymbol{x}_{\tau_1}, \cdots, \boldsymbol{x}_{\tau_K})) = \frac{d}{2} \sum_{k=2}^{K} J(\tau_{k-1}, \tau_k) + c, \quad (14)$$

where $J(\tau_{k-1}, \tau_k) = \log(\sigma^{*2}_{\tau_{k-1}|\tau_k} / \lambda^2_{\tau_{k-1}|\tau_k})$ and $c$ is a constant unrelated to the trajectory $\tau$ (see proof in Appendix A.4). The KL in Eq. (14) can be decomposed into $K - 1$ terms and each term has an analytic form w.r.t. the score function. We view each term as a cost function $J$ evaluated at $(\tau_{k-1}, \tau_k)$, and it can be efficiently estimated by $J(\tau_{k-1}, \tau_k) \approx \log(\hat{\sigma}^2_{\tau_{k-1}|\tau_k} / \lambda^2_{\tau_{k-1}|\tau_k})$, which doesn't require any neural network computation once $\Gamma$ is given. While the logarithmic function causes bias even when the correct score function is known, it can be reduced by increasing $M$.

As a result, Eq. (14) is reduced to a canonical least-cost-path problem (Watson et al., 2021) on a directed graph, where the nodes are $\{1, 2, \cdots, N\}$ and the edge from $s$ to $t$ has cost $J(s, t)$. We want to find a least-cost path of $K$ nodes starting from 1 and terminating at $N$. This problem can be solved by the dynamic programming (DP) algorithm introduced by Watson et al. (2021). We present this algorithm in Appendix B. Besides, we can also extend Eq. (14) to DPMs with continuous timesteps (Song et al., 2020b; Kingma et al., 2021), where their corresponding optimal KL divergences are also decomposed to terms determined by score functions. Thereby, the DP algorithm is also applicable. See Appendix E.2 for the extension.

## 5 RELATIONSHIP BETWEEN THE SCORE FUNCTION AND THE DATA COVARIANCE MATRIX

In this part, we further reveal a relationship between the score function and the data covariance matrix. Indeed, the data covariance matrix can be decomposed to the sum of $\mathbb{E}_{q(\boldsymbol{x}_n)}\mathrm{Cov}_{q(\boldsymbol{x}_0|\boldsymbol{x}_n)}[\boldsymbol{x}_0]$ and $\mathrm{Cov}_{q(\boldsymbol{x}_n)}\mathbb{E}_{q(\boldsymbol{x}_0|\boldsymbol{x}_n)}[\boldsymbol{x}_0]$, where the first term can be represented by the score function. Further, the second term is negligible when $n$ is sufficiently large because $\boldsymbol{x}_0$ and $\boldsymbol{x}_n$ are almost independent. In such cases, the data covariance matrix is almost determined by the score function. Currently, the relationship is purely theoretical and its practical implication is unclear. See details in Appendix A.5.

## 6 EXPERIMENTS

We consider the DDPM forward process ($\lambda_n^2 = \tilde{\beta}_n$) and the DDIM forward process ($\lambda_n^2 = 0$), which are the two most commonly used special cases of Eq. (2). We denote our method, which uses the analytic estimate $\sigma_n^2 = \hat{\sigma}_n^2$, as *Analytic-DPM*, and explicitly call it *Analytic-DDPM* or *Analytic-DDIM* according to which forward process is used. We compare our Analytic-DPM with the original DDPM (Ho et al., 2020), where the reverse variance is either $\sigma_n^2 = \tilde{\beta}_n$ or $\sigma_n^2 = \beta_n$, as well as the original DDIM (Song et al., 2020a), where the reverse variance is $\sigma_n^2 = \lambda_n^2 = 0$.

We adopt two methods to get the trajectory for both the analytic-DPM and baselines. The first one is even trajectory (ET) (Nichol & Dhariwal, 2021), where the timesteps are determined according to a fixed stride (see details in Appendix F.4). The second one is optimal trajectory (OT) (Watson et al., 2021), where the timesteps are calculated via dynamic programming (see Section 4). Note that the baselines calculate the OT based on the $L_{\mathrm{vb}}$ with their handcrafted variances (Watson et al., 2021).

We apply our Analytic-DPM to three pretrained score-based models provided by prior works (Ho et al., 2020; Song et al., 2020a; Nichol & Dhariwal, 2021), as well as two score-based models trained by ourselves. The pretrained score-based models are trained on CelebA 64x64 (Liu et al., 2015), ImageNet 64x64 (Deng et al., 2009) and LSUN Bedroom (Yu et al., 2015) respectively. Our score-based models are trained on CIFAR10 (Krizhevsky et al., 2009) with two different forward noise schedules: the linear schedule (LS) (Ho et al., 2020) and the cosine schedule (CS) (Nichol & Dhariwal, 2021). We denote them as CIFAR10 (LS) and CIFAR10 (CS) respectively. The number of the full timesteps $N$ is 4000 for ImageNet 64x64 and 1000 for other datasets. During sampling, we only display the mean of $p(\boldsymbol{x}_0|\boldsymbol{x}_1)$ and discard the noise following Ho et al. (2020), and we additionally clip the noise scale $\sigma_2$ of $p(\boldsymbol{x}_1|\boldsymbol{x}_2)$ for all methods compared in Table 2 (see details in Appendix F.2 and its ablation study in Appendix G.4). See more experimental details in Appendix F.

Table 1: Negative log-likelihood (bits/dim) ↓ under the DDPM forward process. We show results under trajectories of different number of timesteps $K$. We select the minimum $K$ such that analytic-DPM can outperform the baselines with full timesteps and underline the corresponding results.

| Model \ # timesteps $K$ | | 10 | 25 | 50 | 100 | 200 | 400 | 1000 |
|---|---|---|---|---|---|---|---|---|
| **CIFAR10 (LS)** | | | | | | | | |
| ET | DDPM, $\sigma_n^2 = \tilde{\beta}_n$ | 74.95 | 24.98 | 12.01 | 7.08 | 5.03 | 4.29 | 3.73 |
| | DDPM, $\sigma_n^2 = \beta_n$ | 6.99 | 6.11 | 5.44 | 4.86 | 4.39 | 4.07 | 3.75 |
| | Analytic-DDPM | **5.47** | **4.79** | **4.38** | **4.07** | **3.84** | **3.71** | **3.59** |
| OT | DDPM, $\sigma_n^2 = \beta_n$ | 5.38 | 4.34 | 3.97 | 3.82 | 3.77 | 3.75 | 3.75 |
| | Analytic-DDPM | **4.11** | **3.68** | **3.61** | **3.59** | **3.59** | **3.59** | **3.59** |
| **CIFAR10 (CS)** | | | | | | | | |
| ET | DDPM, $\sigma_n^2 = \tilde{\beta}_n$ | 75.96 | 24.94 | 11.96 | 7.04 | 4.95 | 4.19 | 3.60 |
| | DDPM, $\sigma_n^2 = \beta_n$ | 6.51 | 5.55 | 4.92 | 4.41 | 4.03 | 3.78 | 3.54 |
| | Analytic-DDPM | **5.08** | **4.45** | **4.09** | **3.83** | **3.64** | **3.53** | **3.42** |
| OT | DDPM, $\sigma_n^2 = \beta_n$ | 5.51 | 4.30 | 3.86 | 3.65 | 3.57 | 3.54 | 3.54 |
| | Analytic-DDPM | **3.99** | **3.56** | **3.47** | **3.44** | **3.43** | **3.42** | **3.42** |
| **CelebA 64x64** | | | | | | | | |
| ET | DDPM, $\sigma_n^2 = \tilde{\beta}_n$ | 33.42 | 13.09 | 7.14 | 4.60 | 3.45 | 3.03 | 2.71 |
| | DDPM, $\sigma_n^2 = \beta_n$ | 6.67 | 5.72 | 4.98 | 4.31 | 3.74 | 3.34 | 2.93 |
| | Analytic-DDPM | **4.54** | **3.89** | **3.48** | **3.16** | **2.92** | **2.79** | **2.66** |
| OT | DDPM, $\sigma_n^2 = \beta_n$ | 4.76 | 3.58 | 3.16 | 2.99 | 2.94 | 2.93 | 2.93 |
| | Analytic-DDPM | **2.97** | **2.71** | **2.67** | **2.66** | **2.66** | **2.66** | **2.66** |

| Model \ # timesteps $K$ | | 25 | 50 | 100 | 200 | 400 | 1000 | 4000 |
|---|---|---|---|---|---|---|---|---|
| **ImageNet 64x64** | | | | | | | | |
| ET | DDPM, $\sigma_n^2 = \tilde{\beta}_n$ | 105.87 | 46.25 | 22.02 | 12.10 | 7.59 | 5.04 | 3.89 |
| | DDPM, $\sigma_n^2 = \beta_n$ | 5.81 | 5.20 | 4.70 | 4.31 | 4.04 | 3.81 | 3.65 |
| | Analytic-DDPM | **4.78** | **4.42** | **4.15** | **3.95** | **3.81** | **3.69** | **3.61** |
| OT | DDPM, $\sigma_n^2 = \beta_n$ | 4.56 | 4.09 | 3.84 | 3.73 | 3.68 | 3.65 | 3.65 |
| | Analytic-DDPM | **3.83** | **3.70** | **3.64** | **3.62** | **3.62** | **3.61** | **3.61** |

We conduct extensive experiments to demonstrate that analytic-DPM can consistently improve the inference efficiency of a pretrained DPM while achieving a comparable or even superior performance. Specifically, Section 6.1 and Section 6.2 present the likelihood and sample quality results respectively. Additional experiments such as ablation studies can be found in Appendix G.

## 6.1 LIKELIHOOD RESULTS

Since $\lambda_n^2 = 0$ in the DDIM forward process, its variational bound $L_{\text{vb}}$ is infinite. Thereby, we only consider the likelihood results under the DDPM forward process. As shown in Table 1, on all three datasets, our Analytic-DPM consistently improves the likelihood results of the original DDPM using both ET and OT. Remarkably, using a much shorter trajectory (i.e., a much less inference time), Analytic-DPM with OT can still outperform the baselines. In Table 1, we select the minimum $K$ such that analytic-DPM can outperform the baselines with full timesteps and underline the corresponding results. Specifically, analytic-DPM enjoys a $40\times$ speed up on CIFAR10 (LS) and ImageNet 64x64, and a $20\times$ speed up on CIFAR10 (CS) and CelebA 64x64.

Although we mainly focus on learning-free strategies of choosing the reverse variance, we also compare to another strong baseline that predicts the variance by a neural network (Nichol & Dhariwal, 2021). With full timesteps, Analytic-DPM achieves a NLL of 3.61 on ImageNet 64x64, which is very close to 3.57 reported in Nichol & Dhariwal (2021). Besides, while Nichol & Dhariwal (2021) report that the ET drastically reduces the log-likelihood performance of their neural-network-parameterized variance, Analytic-DPM performs well with the ET. See details in Appendix G.6.

Table 2: FID ↓ under the DDPM and DDIM forward processes. All are evaluated under the even trajectory (ET). The result with $^\dagger$ is slightly better than 3.17 reported by Ho et al. (2020), because we use an improved model architecture following Nichol & Dhariwal (2021).

| Model \ # timesteps $K$ | 10 | 25 | 50 | 100 | 200 | 400 | 1000 |
|---|---|---|---|---|---|---|---|
| **CIFAR10 (LS)** | | | | | | | |
| DDPM, $\sigma_n^2 = \tilde{\beta}_n$ | 44.45 | 21.83 | 15.21 | 10.94 | 8.23 | 6.43 | 5.11 |
| DDPM, $\sigma_n^2 = \beta_n$ | 233.41 | 125.05 | 66.28 | 31.36 | 12.96 | 4.86 | $^\dagger$**3.04** |
| Analytic-DDPM | **34.26** | **11.60** | **7.25** | **5.40** | **4.01** | **3.62** | 4.03 |
| DDIM | 21.31 | 10.70 | 7.74 | 6.08 | 5.07 | 4.61 | 4.13 |
| Analytic-DDIM | **14.00** | **5.81** | **4.04** | **3.55** | **3.39** | **3.50** | **3.74** |
| **CIFAR10 (CS)** | | | | | | | |
| DDPM, $\sigma_n^2 = \tilde{\beta}_n$ | 34.76 | 16.18 | 11.11 | 8.38 | 6.66 | 5.65 | 4.92 |
| DDPM, $\sigma_n^2 = \beta_n$ | 205.31 | 84.71 | 37.35 | 14.81 | 5.74 | **3.40** | **3.34** |
| Analytic-DDPM | **22.94** | **8.50** | **5.50** | **4.45** | **4.04** | 3.96 | 4.31 |
| DDIM | 34.34 | 16.68 | 10.48 | 7.94 | 6.69 | 5.78 | 4.89 |
| Analytic-DDIM | **26.43** | **9.96** | **6.02** | **4.88** | **4.92** | **5.00** | **4.66** |
| **CelebA 64x64** | | | | | | | |
| DDPM, $\sigma_n^2 = \tilde{\beta}_n$ | 36.69 | 24.46 | 18.96 | 14.31 | 10.48 | 8.09 | 5.95 |
| DDPM, $\sigma_n^2 = \beta_n$ | 294.79 | 115.69 | 53.39 | 25.65 | 9.72 | **3.95** | **3.16** |
| Analytic-DDPM | **28.99** | **16.01** | **11.23** | **8.08** | **6.51** | 5.87 | 5.21 |
| DDIM | 20.54 | 13.45 | 9.33 | 6.60 | 4.96 | 4.15 | 3.40 |
| Analytic-DDIM | **15.62** | **9.22** | **6.13** | **4.29** | **3.46** | **3.38** | **3.13** |

| Model \ # timesteps $K$ | 25 | 50 | 100 | 200 | 400 | 1000 | 4000 |
|---|---|---|---|---|---|---|---|
| **ImageNet 64x64** | | | | | | | |
| DDPM, $\sigma_n^2 = \tilde{\beta}_n$ | **29.21** | **21.71** | 19.12 | 17.81 | 17.48 | 16.84 | 16.55 |
| DDPM, $\sigma_n^2 = \beta_n$ | 170.28 | 83.86 | 45.04 | 28.39 | 21.38 | 17.58 | 16.38 |
| Analytic-DDPM | 32.56 | 22.45 | **18.80** | **17.16** | **16.40** | **16.14** | **16.34** |
| DDIM | 26.06 | 20.10 | 18.09 | 17.84 | 17.74 | 17.73 | 19.00 |
| Analytic-DDIM | **25.98** | **19.23** | **17.73** | **17.49** | **17.44** | **17.57** | **18.98** |

## 6.2 SAMPLE QUALITY

As for the sample quality, we consider the commonly used FID score (Heusel et al., 2017), where a lower value indicates a better sample quality. As shown in Table 2, under trajectories of different $K$, our Analytic-DDIM consistently improves the sample quality of the original DDIM. This allows us to generate high-quality samples with less than 50 timesteps, which results in a $20\times$ to $80\times$ speed up compared to the full timesteps. Indeed, in most cases, Analytic-DDIM only requires up to 50 timesteps to get a similar performance to the baselines. Besides, Analytic-DDPM also improves the sample quality of the original DDPM in most cases. For fairness, we use the ET implementation in Nichol & Dhariwal (2021) for all results in Table 2. We also report the results on CelebA 64x64 using a slightly different implementation of the ET following Song et al. (2020a) in Appendix G.7, and our Analytic-DPM is still effective. We show generated samples in Appendix G.9.

We observe that Analytic-DDPM does not always outperform the baseline under the FID metric, which is inconsistent with the likelihood results in Table 1. Such a behavior essentially roots in the different natures of the two metrics and has been investigated in extensive prior works (Theis et al., 2015; Ho et al., 2020; Nichol & Dhariwal, 2021; Song et al., 2021; Vahdat et al., 2021; Watson et al., 2021; Kingma et al., 2021). Similarly, using more timesteps doesn't necessarily yield a better FID. For instance, see the Analytic-DDPM results on CIFAR10 (LS) and the DDIM results on ImageNet 64x64 in Table 2. A similar phenomenon is observed in Figure 8 in Nichol & Dhariwal (2021). Moreover, a DPM (including Analytic-DPM) with OT does not necessarily lead to a better FID score (Watson et al., 2021) (see Appendix G.5 for a comparison of ET and OT in Analytic-DPM).

Table 3: Efficiency comparison, based on the least number of timesteps ↓ required to achieve a FID around 6 (with the corresponding FID). To get the strongest baseline, the results with [†] are achieved by using the quadratic trajectory Song et al. (2020a) instead of the default even trajectory.

| Method | CIFAR10 | CelebA 64x64 | LSUN Bedroom |
|---|---|---|---|
| DDPM (Ho et al., 2020) | [†]90 (6.12) | > 200 | 130 (6.06) |
| DDIM (Song et al., 2020a) | [†]30 (5.85) | > 100 | Best FID > 6 |
| Improved DDPM (Nichol & Dhariwal, 2021) | 45 (5.96) | Missing model | **90** (6.02) |
| Analytic-DPM (ours) | **25** (5.81) | **55** (5.98) | 100 (6.05) |

We summarize the efficiency of different methods in Table 3, where we consider the least number of timesteps required to achieve a FID around 6 as the metric for a more direct comparison.

## 7 RELATED WORK

**DPMs and their applications.** The diffusion probabilistic model (DPM) is initially introduced by Sohl-Dickstein et al. (2015), where the DPM is trained by optimizing the variational bound $L_{\text{vb}}$. Ho et al. (2020) propose the new parameterization of DPMs in Eq. (3) and learn DPMs with the reweighted variant of $L_{\text{vb}}$ in Eq. (5). Song et al. (2020b) model the noise adding forward process as a stochastic differential equation (SDE) and introduce DPMs with continuous timesteps. With these important improvements, DPMs show great potential in various applications, including speech synthesis (Chen et al., 2020; Kong et al., 2020; Popov et al., 2021; Lam et al., 2021), controllable generation (Choi et al., 2021; Sinha et al., 2021), image super-resolution (Saharia et al., 2021; Li et al., 2021), image-to-image translation (Sasaki et al., 2021), shape generation (Zhou et al., 2021) and time series forecasting (Rasul et al., 2021).

**Faster DPMs.** Several works attempt to find short trajectories while maintaining the DPM performance. Chen et al. (2020) find an effective trajectory of only six timesteps by the grid search. However, the grid search is only applicable to very short trajectories due to its exponentially growing time complexity. Watson et al. (2021) model the trajectory searching as a least-cost-path problem and introduce a dynamic programming (DP) algorithm to solve this problem. Our work uses this DP algorithm, where the cost is defined as a term of the optimal KL divergence. In addition to these trajectory searching techniques, Luhman & Luhman (2021) compress the reverse denoising process into a single step model; San-Roman et al. (2021) dynamically adjust the trajectory during inference. Both of them need extra training after getting a pretrained DPM. As for DPMs with continuous timesteps (Song et al., 2020b), Song et al. (2020b) introduce an ordinary differential equation (ODE), which improves sampling efficiency and enables exact likelihood computation. However, the likelihood computation involves a stochastic trace estimator, which requires a multiple number of runs for accurate computation. Jolicoeur-Martineau et al. (2021) introduce an advanced SDE solver to simulate the reverse process in a more efficient way. However, the log-likelihood computation based on this solver is not specified.

**Variance Learning in DPMs.** In addition to the reverse variance, there are also works on learning the forward noise schedule (i.e., the forward variance). Kingma et al. (2021) propose variational diffusion models (VDMs) on continuous timesteps, which use a signal-to-noise ratio function to parameterize the forward variance and directly optimize the variational bound objective for a better log-likelihood. While we primarily apply our method to DDPMs and DDIMs, estimating the optimal reverse variance can also be applied to VDMs (see Appendix E).

## 8 CONCLUSION

We present that both the optimal reverse variance and the corresponding optimal KL divergence of a DPM have analytic forms w.r.t. its score function. Building upon it, we propose *Analytic-DPM*, a training-free inference framework that estimates the analytic forms of the variance and KL divergence using the Monte Carlo method and a pretrained score-based model. We derive bounds of the optimal variance to correct potential bias and reveal a relationship between the score function and the data covariance matrix. Empirically, our analytic-DPM improves both the efficiency and performance of likelihood results, and generates high-quality samples efficiently in various DPMs.

## ACKNOWLEDGMENTS

This work was supported by NSF of China Projects (Nos. 62061136001, 61620106010, 62076145), Beijing NSF Project (No. JQ19016), Beijing Outstanding Young Scientist Program NO. BJJWZYJH012019100020098, Beijing Academy of Artificial Intelligence (BAAI), Tsinghua-Huawei Joint Research Program, a grant from Tsinghua Institute for Guo Qiang, and the NVIDIA NVAIL Program with GPU/DGX Acceleration, Major Innovation & Planning Interdisciplinary Platform for the "Double-First Class" Initiative, Renmin University of China.

## ETHICS STATEMENT

This work proposes an analytic estimate of the optimal variance in the reverse process of diffusion probabilistic models. As a fundamental research in machine learning, the negative consequences are not obvious. Though in theory any technique can be misused, it is not likely to happen at the current stage.

## REPRODUCIBILITY STATEMENT

We provide our codes and links to pretrained models in `https://github.com/baofff/Analytic-DPM`. We provide details of these pretrained models in Appendix F.1. We provide details of data processing, log-likelihood evaluation, sampling and FID computation in Appendix F.2. We provide complete proofs of all theoretical results in Appendix A.

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

# A    PROOFS AND DERIVATIONS

## A.1    LEMMAS

**Lemma 1.** *(Cross-entropy to Gaussian) Suppose $q(\boldsymbol{x})$ is a probability density function with mean $\boldsymbol{\mu}_q$ and covariance matrix $\boldsymbol{\Sigma}_q$ and $p(\boldsymbol{x}) = \mathcal{N}(\boldsymbol{x}|\boldsymbol{\mu}, \boldsymbol{\Sigma})$ is a Gaussian distribution, then the cross-entropy between $q$ and $p$ is equal to the cross-entropy between $\mathcal{N}(\boldsymbol{x}|\boldsymbol{\mu}_q, \boldsymbol{\Sigma}_q)$ and $p$, i.e.,*

$$H(q, p) = H(\mathcal{N}(\boldsymbol{x}|\boldsymbol{\mu}_q, \boldsymbol{\Sigma}_q), p)$$
$$= \frac{1}{2}\log((2\pi)^d|\boldsymbol{\Sigma}|) + \frac{1}{2}\operatorname{tr}(\boldsymbol{\Sigma}_q\boldsymbol{\Sigma}^{-1}) + \frac{1}{2}(\boldsymbol{\mu}_q - \boldsymbol{\mu})^{\top}\boldsymbol{\Sigma}^{-1}(\boldsymbol{\mu}_q - \boldsymbol{\mu}).$$

*Proof.*

$$H(q, p) = -\mathbb{E}_{q(\boldsymbol{x})}\log p(\boldsymbol{x}) = -\mathbb{E}_{q(\boldsymbol{x})}\log \frac{1}{\sqrt{(2\pi)^d|\boldsymbol{\Sigma}|}}\exp(-\frac{(\boldsymbol{x} - \boldsymbol{\mu})^{\top}\boldsymbol{\Sigma}^{-1}(\boldsymbol{x} - \boldsymbol{\mu})}{2})$$

$$= \frac{1}{2}\log((2\pi)^d|\boldsymbol{\Sigma}|) + \frac{1}{2}\mathbb{E}_{q(\boldsymbol{x})}(\boldsymbol{x} - \boldsymbol{\mu})^{\top}\boldsymbol{\Sigma}^{-1}(\boldsymbol{x} - \boldsymbol{\mu})$$

$$= \frac{1}{2}\log((2\pi)^d|\boldsymbol{\Sigma}|) + \frac{1}{2}\mathbb{E}_{q(\boldsymbol{x})}\operatorname{tr}((\boldsymbol{x} - \boldsymbol{\mu})(\boldsymbol{x} - \boldsymbol{\mu})^{\top}\boldsymbol{\Sigma}^{-1})$$

$$= \frac{1}{2}\log((2\pi)^d|\boldsymbol{\Sigma}|) + \frac{1}{2}\operatorname{tr}(\mathbb{E}_{q(\boldsymbol{x})}\left[(\boldsymbol{x} - \boldsymbol{\mu})(\boldsymbol{x} - \boldsymbol{\mu})^{\top}\right]\boldsymbol{\Sigma}^{-1})$$

$$= \frac{1}{2}\log((2\pi)^d|\boldsymbol{\Sigma}|) + \frac{1}{2}\operatorname{tr}(\mathbb{E}_{q(\boldsymbol{x})}\left[(\boldsymbol{x} - \boldsymbol{\mu}_q)(\boldsymbol{x} - \boldsymbol{\mu}_q)^{\top} + (\boldsymbol{\mu}_q - \boldsymbol{\mu})(\boldsymbol{\mu}_q - \boldsymbol{\mu})^{\top}\right]\boldsymbol{\Sigma}^{-1})$$

$$= \frac{1}{2}\log((2\pi)^d|\boldsymbol{\Sigma}|) + \frac{1}{2}\operatorname{tr}(\left[\boldsymbol{\Sigma}_q + (\boldsymbol{\mu}_q - \boldsymbol{\mu})(\boldsymbol{\mu}_q - \boldsymbol{\mu})^{\top}\right]\boldsymbol{\Sigma}^{-1})$$

$$= \frac{1}{2}\log((2\pi)^d|\boldsymbol{\Sigma}|) + \frac{1}{2}\operatorname{tr}(\boldsymbol{\Sigma}_q\boldsymbol{\Sigma}^{-1}) + \frac{1}{2}\operatorname{tr}((\boldsymbol{\mu}_q - \boldsymbol{\mu})(\boldsymbol{\mu}_q - \boldsymbol{\mu})^{\top}\boldsymbol{\Sigma}^{-1})$$

$$= \frac{1}{2}\log((2\pi)^d|\boldsymbol{\Sigma}|) + \frac{1}{2}\operatorname{tr}(\boldsymbol{\Sigma}_q\boldsymbol{\Sigma}^{-1}) + \frac{1}{2}(\boldsymbol{\mu}_q - \boldsymbol{\mu})^{\top}\boldsymbol{\Sigma}^{-1}(\boldsymbol{\mu}_q - \boldsymbol{\mu})$$

$$= H(\mathcal{N}(\boldsymbol{x}|\boldsymbol{\mu}_q, \boldsymbol{\Sigma}_q), p).$$

$\square$

**Lemma 2.** *(KL to Gaussian) Suppose $q(\boldsymbol{x})$ is a probability density function with mean $\boldsymbol{\mu}_q$ and covariance matrix $\boldsymbol{\Sigma}_q$ and $p(\boldsymbol{x}) = \mathcal{N}(\boldsymbol{x}|\boldsymbol{\mu}, \boldsymbol{\Sigma})$ is a Gaussian distribution, then*

$$D_{\mathrm{KL}}(q||p) = D_{\mathrm{KL}}(\mathcal{N}(\boldsymbol{x}|\boldsymbol{\mu}_q, \boldsymbol{\Sigma}_q)||p) + H(\mathcal{N}(\boldsymbol{x}|\boldsymbol{\mu}_q, \boldsymbol{\Sigma}_q)) - H(q),$$

*where $H(\cdot)$ denotes the entropy of a distribution.*

*Proof.* According to Lemma 1, we have $H(q, p) = H(\mathcal{N}(\boldsymbol{x}|\boldsymbol{\mu}_q, \boldsymbol{\Sigma}_q), p)$. Thereby,

$$D_{\mathrm{KL}}(q||p) = H(q, p) - H(q) = H(\mathcal{N}(\boldsymbol{x}|\boldsymbol{\mu}_q, \boldsymbol{\Sigma}_q), p) - H(q)$$
$$= H(\mathcal{N}(\boldsymbol{x}|\boldsymbol{\mu}_q, \boldsymbol{\Sigma}_q), p) - H(\mathcal{N}(\boldsymbol{x}|\boldsymbol{\mu}_q, \boldsymbol{\Sigma}_q)) + H(\mathcal{N}(\boldsymbol{x}|\boldsymbol{\mu}_q, \boldsymbol{\Sigma}_q)) - H(q)$$
$$= D_{\mathrm{KL}}(\mathcal{N}(\boldsymbol{x}|\boldsymbol{\mu}_q, \boldsymbol{\Sigma}_q)||p) + H(\mathcal{N}(\boldsymbol{x}|\boldsymbol{\mu}_q, \boldsymbol{\Sigma}_q)) - H(q).$$

$\square$

**Lemma 3.** *(Equivalence between the forward and reverse Markov property) Suppose $q(\boldsymbol{x}_{0:N}) = q(\boldsymbol{x}_0)\prod_{n=1}^{N}q(\boldsymbol{x}_n|\boldsymbol{x}_{n-1})$ is a Markov chain, then $q$ is also a Markov chain in the reverse direction, i.e., $q(\boldsymbol{x}_{0:N}) = q(\boldsymbol{x}_N)\prod_{n=1}^{N}q(\boldsymbol{x}_{n-1}|\boldsymbol{x}_n).$*

*Proof.*

$$q(\boldsymbol{x}_{n-1}|\boldsymbol{x}_n,\cdots,\boldsymbol{x}_N) = \frac{q(\boldsymbol{x}_{n-1},\boldsymbol{x}_n,\cdots,\boldsymbol{x}_N)}{q(\boldsymbol{x}_n,\cdots,\boldsymbol{x}_N)}$$

$$= \frac{q(\boldsymbol{x}_{n-1},\boldsymbol{x}_n)\prod\limits_{i=n+1}^{N}q(\boldsymbol{x}_i|\boldsymbol{x}_{i-1})}{q(\boldsymbol{x}_n)\prod\limits_{i=n+1}^{N}q(\boldsymbol{x}_i|\boldsymbol{x}_{i-1})} = q(\boldsymbol{x}_{n-1}|\boldsymbol{x}_n).$$

Thereby, $q(\boldsymbol{x}_{0:N}) = q(\boldsymbol{x}_N)\prod\limits_{n=1}^{N}q(\boldsymbol{x}_{n-1}|\boldsymbol{x}_n)$. □

**Lemma 4.** *(Entropy of a Markov chain) Suppose $q(\boldsymbol{x}_{0:N})$ is a Markov chain, then*

$$H(q(\boldsymbol{x}_{0:N})) = H(q(\boldsymbol{x}_N)) + \sum_{n=1}^{N}\mathbb{E}_q H(q(\boldsymbol{x}_{n-1}|\boldsymbol{x}_n)) = H(q(\boldsymbol{x}_0)) + \sum_{n=1}^{N}\mathbb{E}_q H(q(\boldsymbol{x}_n|\boldsymbol{x}_{n-1})).$$

*Proof.* According to Lemma 3, we have

$$H(q(\boldsymbol{x}_{0:N})) = -\mathbb{E}_q \log q(\boldsymbol{x}_N)\prod_{n=1}^{N}q(\boldsymbol{x}_{n-1}|\boldsymbol{x}_n) = -\mathbb{E}_q \log q(\boldsymbol{x}_N) - \sum_{n=1}^{N}\mathbb{E}_q \log q(\boldsymbol{x}_{n-1}|\boldsymbol{x}_n)$$

$$= H(q(\boldsymbol{x}_N)) + \sum_{n=1}^{N}\mathbb{E}_q H(q(\boldsymbol{x}_{n-1}|\boldsymbol{x}_n)).$$

Similarly, we also have $H(q(\boldsymbol{x}_{0:N})) = H(q(\boldsymbol{x}_0)) + \sum\limits_{n=1}^{N}\mathbb{E}_q H(q(\boldsymbol{x}_n|\boldsymbol{x}_{n-1}))$. □

**Lemma 5.** *(Entropy of a DDPM forward process) Suppose $q(\boldsymbol{x}_{0:N})$ is a Markov chain and $q(\boldsymbol{x}_n|\boldsymbol{x}_{n-1}) = \mathcal{N}(\boldsymbol{x}_n|\sqrt{\alpha_n}\boldsymbol{x}_{n-1},\beta_n\boldsymbol{I})$, then*

$$H(q(\boldsymbol{x}_{0:N})) = H(q(\boldsymbol{x}_0)) + \frac{d}{2}\sum_{n=1}^{N}\log(2\pi e\beta_n).$$

*Proof.* According to Lemma 4, we have

$$H(q(\boldsymbol{x}_{0:N})) = H(q(\boldsymbol{x}_0)) + \sum_{n=1}^{N}\mathbb{E}_q H(q(\boldsymbol{x}_n|\boldsymbol{x}_{n-1})) = H(q(\boldsymbol{x}_0)) + \sum_{n=1}^{N}\frac{d}{2}\log(2\pi e\beta_n).$$

□

**Lemma 6.** *(Entropy of a conditional Markov chain) Suppose $q(\boldsymbol{x}_{1:N}|\boldsymbol{x}_0)$ is Markov, then*

$$H(q(\boldsymbol{x}_{0:N})) = H(q(\boldsymbol{x}_0)) + \mathbb{E}_q H(q(\boldsymbol{x}_N|\boldsymbol{x}_0)) + \sum_{n=2}^{N}\mathbb{E}_q H(q(\boldsymbol{x}_{n-1}|\boldsymbol{x}_n,\boldsymbol{x}_0)).$$

*Proof.* According to Lemma 4, we have

$$H(q(\boldsymbol{x}_{0:N})) = H(q(\boldsymbol{x}_0)) + \mathbb{E}_q H(q(\boldsymbol{x}_{1:N}|\boldsymbol{x}_0))$$

$$= H(q(\boldsymbol{x}_0)) + \mathbb{E}_q H(q(\boldsymbol{x}_N|\boldsymbol{x}_0)) + \sum_{n=2}^{N}\mathbb{E}_q H(q(\boldsymbol{x}_{n-1}|\boldsymbol{x}_n,\boldsymbol{x}_0)).$$

□

**Lemma 7.** *(Entropy of a generalized DDPM forward process) Suppose $q(\boldsymbol{x}_{1:N}|\boldsymbol{x}_0)$ is Markov, $q(\boldsymbol{x}_N|\boldsymbol{x}_0)$ is Gaussian with covariance $\overline{\beta}_N \boldsymbol{I}$ and $q(\boldsymbol{x}_{n-1}|\boldsymbol{x}_n,\boldsymbol{x}_0)$ is Gaussian with covariance $\lambda_n^2 \boldsymbol{I}$, then*

$$H(q(\boldsymbol{x}_{0:N})) = H(q(\boldsymbol{x}_0)) + \frac{d}{2}\log(2\pi e\overline{\beta}_N) + \frac{d}{2}\sum_{n=2}^{N}\log(2\pi e\lambda_n^2).$$

*Proof.* Directly derived from Lemma 6. $\qquad\square$

**Lemma 8.** *(KL to a Markov chain) Suppose $q(\boldsymbol{x}_{0:N})$ is a probability distribution and $p(\boldsymbol{x}_{0:N}) = p(\boldsymbol{x}_N)\prod_{n=1}^{N}p(\boldsymbol{x}_{n-1}|\boldsymbol{x}_n)$ is a Markov chain, then we have*

$$\mathbb{E}_q D_{\mathrm{KL}}(q(\boldsymbol{x}_{0:N-1}|\boldsymbol{x}_N)||p(\boldsymbol{x}_{0:N-1}|\boldsymbol{x}_N)) = \sum_{n=1}^{N}\mathbb{E}_q D_{\mathrm{KL}}(q(\boldsymbol{x}_{n-1}|\boldsymbol{x}_n)||p(\boldsymbol{x}_{n-1}|\boldsymbol{x}_n)) + c,$$

*where $c = \sum_{n=1}^{N}\mathbb{E}_q H(q(\boldsymbol{x}_{n-1}|\boldsymbol{x}_n)) - \mathbb{E}_q H(q(\boldsymbol{x}_{0:N-1}|\boldsymbol{x}_N))$ is only related to q. Particularly, if $q(\boldsymbol{x}_{0:N})$ is also a Markov chain, then $c = 0$.*

*Proof.*

$$\mathbb{E}_q D_{\mathrm{KL}}(q(\boldsymbol{x}_{0:N-1}|\boldsymbol{x}_N)||p(\boldsymbol{x}_{0:N-1}|\boldsymbol{x}_N)) = -\mathbb{E}_q\log p(\boldsymbol{x}_{0:N-1}|\boldsymbol{x}_N) - \mathbb{E}_q H(q(\boldsymbol{x}_{0:N-1}|\boldsymbol{x}_N))$$

$$= -\sum_{n=1}^{N}\mathbb{E}_q\log p(\boldsymbol{x}_{n-1}|\boldsymbol{x}_n) - \mathbb{E}_q H(q(\boldsymbol{x}_{0:N-1}|\boldsymbol{x}_N))$$

$$= \sum_{n=1}^{N}\mathbb{E}_q D_{\mathrm{KL}}(q(\boldsymbol{x}_{n-1}|\boldsymbol{x}_n)||p(\boldsymbol{x}_{n-1}|\boldsymbol{x}_n)) + \sum_{n=1}^{N}\mathbb{E}_q H(q(\boldsymbol{x}_{n-1}|\boldsymbol{x}_n)) - \mathbb{E}_q H(q(\boldsymbol{x}_{0:N-1}|\boldsymbol{x}_N)).$$

Let $c = \sum_{n=1}^{N}\mathbb{E}_q H(q(\boldsymbol{x}_{n-1}|\boldsymbol{x}_n)) - \mathbb{E}_q H(q(\boldsymbol{x}_{0:N-1}|\boldsymbol{x}_N))$, then

$$\mathbb{E}_q D_{\mathrm{KL}}(q(\boldsymbol{x}_{0:N-1}|\boldsymbol{x}_N)||p(\boldsymbol{x}_{0:N-1}|\boldsymbol{x}_N)) = \sum_{n=1}^{N}\mathbb{E}_q D_{\mathrm{KL}}(q(\boldsymbol{x}_{n-1}|\boldsymbol{x}_n)||p(\boldsymbol{x}_{n-1}|\boldsymbol{x}_n)) + c.$$

If $q(\boldsymbol{x}_{0:N})$ is also a Markov chain, according to Lemma 4, we have $c = 0$. $\qquad\square$

**Lemma 9.** *(The optimal Markov reverse process with Gaussian transitions is equivalent to moment matching) Suppose $q(\boldsymbol{x}_{0:N})$ is probability density function and $p(\boldsymbol{x}_{0:N}) = \prod_{n=1}^{N}p(\boldsymbol{x}_{n-1}|\boldsymbol{x}_n)p(\boldsymbol{x}_N)$ is a Gaussian Markov chain with $p(\boldsymbol{x}_{n-1}|\boldsymbol{x}_n) = \mathcal{N}(\boldsymbol{x}_{n-1}|\boldsymbol{\mu}_n(\boldsymbol{x}_n),\sigma_n^2\boldsymbol{I})$, then the joint KL optimization*

$$\min_{\{\boldsymbol{\mu}_n,\sigma_n^2\}_{n=1}^{N}}D_{\mathrm{KL}}(q(\boldsymbol{x}_{0:N})||p(\boldsymbol{x}_{0:N}))$$

*has an optimal solution*

$$\boldsymbol{\mu}_n^*(\boldsymbol{x}_n) = \mathbb{E}_{q(\boldsymbol{x}_{n-1}|\boldsymbol{x}_n)}[\boldsymbol{x}_{n-1}], \quad \sigma_n^{*2} = \mathbb{E}_{q_n(\boldsymbol{x}_n)}\frac{\mathrm{tr}(\mathrm{Cov}_{q(\boldsymbol{x}_{n-1}|\boldsymbol{x}_n)}[\boldsymbol{x}_{n-1}])}{d},$$

*which match the first two moments of $q(\boldsymbol{x}_{n-1}|\boldsymbol{x}_n)$. The corresponding optimal KL is*

$$D_{\mathrm{KL}}(q(\boldsymbol{x}_{0:N})||p^*(\boldsymbol{x}_{0:N})) = H(q(\boldsymbol{x}_N),p(\boldsymbol{x}_N)) + \frac{d}{2}\sum_{n=1}^{N}\log(2\pi e\sigma_n^{*2}) - H(q(\boldsymbol{x}_{0:N})).$$

**Remark.** *Lemma 9 doesn't assume the form of $q(\boldsymbol{x}_{0:N})$, thereby it can be applied to more general Gaussian models, such as multi-layer VAEs with Gaussian decoders (Rezende et al., 2014; Burda et al., 2015). In this case, $q(\boldsymbol{x}_{1:N}|\boldsymbol{x}_0)$ is the hierarchical encoders of multi-layer VAEs.*

*Proof.* According to Lemma 8, we have

$$D_{\mathrm{KL}}(q(\boldsymbol{x}_{0:N})||p(\boldsymbol{x}_{0:N})) = D_{\mathrm{KL}}(q(\boldsymbol{x}_N)||p(\boldsymbol{x}_N)) + \sum_{n=1}^{N} \mathbb{E}_q D_{\mathrm{KL}}(q(\boldsymbol{x}_{n-1}|\boldsymbol{x}_n)||p(\boldsymbol{x}_{n-1}|\boldsymbol{x}_n)) + c,$$

where $c = \sum\limits_{n=1}^{N} \mathbb{E}_q H(q(\boldsymbol{x}_{n-1}|\boldsymbol{x}_n)) - \mathbb{E}_q H(q(\boldsymbol{x}_{0:N-1}|\boldsymbol{x}_N))$.

Since $\mathbb{E}_q D_{\mathrm{KL}}(q(\boldsymbol{x}_{n-1}|\boldsymbol{x}_n)||p(\boldsymbol{x}_{n-1}|\boldsymbol{x}_n))$ is only related to $\boldsymbol{\mu}_n(\cdot)$ and $\sigma_n^2$, the joint KL optimization is decomposed into $n$ independent optimization sub-problems:

$$\min_{\boldsymbol{\mu}_n, \sigma_n^2} \mathbb{E}_q D_{\mathrm{KL}}(q(\boldsymbol{x}_{n-1}|\boldsymbol{x}_n)||p(\boldsymbol{x}_{n-1}|\boldsymbol{x}_n)), \quad 1 \le n \le N.$$

According to Lemma 2, we have

$$\begin{aligned}
&\mathbb{E}_q D_{\mathrm{KL}}(q(\boldsymbol{x}_{n-1}|\boldsymbol{x}_n)||p(\boldsymbol{x}_{n-1}|\boldsymbol{x}_n)) \\
=&\mathbb{E}_q D_{\mathrm{KL}}(\mathcal{N}(\boldsymbol{x}_{n-1}|\mathbb{E}_{q(\boldsymbol{x}_{n-1}|\boldsymbol{x}_n)}[\boldsymbol{x}_{n-1}], \mathrm{Cov}_{q(\boldsymbol{x}_{n-1}|\boldsymbol{x}_n)}[\boldsymbol{x}_{n-1}])||p(\boldsymbol{x}_{n-1}|\boldsymbol{x}_n)) \\
&+ \mathbb{E}_q H(\mathcal{N}(\boldsymbol{x}_{n-1}|\mathbb{E}_{q(\boldsymbol{x}_{n-1}|\boldsymbol{x}_n)}[\boldsymbol{x}_{n-1}], \mathrm{Cov}_{q(\boldsymbol{x}_{n-1}|\boldsymbol{x}_n)}[\boldsymbol{x}_{n-1}])) - \mathbb{E}_q H(q(\boldsymbol{x}_{n-1}|\boldsymbol{x}_n)) \\
=&\mathcal{F}(\sigma_n^2) + \mathcal{G}(\sigma_n^2, \boldsymbol{\mu}_n) + c'
\end{aligned}$$

where

$$\mathcal{F}(\sigma_n^2) = \frac{1}{2}\left(\sigma_n^{-2}\mathbb{E}_q \operatorname{tr}(\mathrm{Cov}_{q(\boldsymbol{x}_{n-1}|\boldsymbol{x}_n)}[\boldsymbol{x}_{n-1}]) + d\log\sigma_n^2\right),$$

$$\mathcal{G}(\sigma_n^2, \boldsymbol{\mu}_n) = \frac{1}{2}\sigma_n^{-2}\mathbb{E}_q||\mathbb{E}_{q(\boldsymbol{x}_{n-1}|\boldsymbol{x}_n)}[\boldsymbol{x}_{n-1}] - \boldsymbol{\mu}_n(\boldsymbol{x}_n)||^2,$$

and $c' = \frac{d}{2}\log(2\pi) - \mathbb{E}_q H(q(\boldsymbol{x}_{n-1}|\boldsymbol{x}_n))$. The optimal $\boldsymbol{\mu}_n^*(\boldsymbol{x}_n)$ is achieved when

$$||\mathbb{E}_{q(\boldsymbol{x}_{n-1}|\boldsymbol{x}_n)}[\boldsymbol{x}_{n-1}] - \boldsymbol{\mu}_n(\boldsymbol{x}_n)||^2 = 0.$$

Thereby, $\boldsymbol{\mu}_n^*(\boldsymbol{x}_n) = \mathbb{E}_{q(\boldsymbol{x}_{n-1}|\boldsymbol{x}_n)}[\boldsymbol{x}_{n-1}]$. In this case, $\mathcal{G}(\sigma_n^2, \boldsymbol{\mu}_n^*) = 0$ and we only need to consider $\mathcal{F}(\sigma_n^2)$ for the optimal $\sigma_n^{*2}$. By calculating the gradient of $\mathcal{F}$, we know that $\mathcal{F}$ gets its minimum at

$$\sigma_n^{*2} = \mathbb{E}_q \frac{\operatorname{tr}(\mathrm{Cov}_{q(\boldsymbol{x}_{n-1}|\boldsymbol{x}_n)}[\boldsymbol{x}_{n-1}])}{d}.$$

In the optimal case, $\mathcal{F}(\sigma_n^{*2}) = \frac{d}{2}(1 + \log\sigma_n^{*2})$ and

$$E_q D_{\mathrm{KL}}(q(\boldsymbol{x}_{n-1}|\boldsymbol{x}_n)||p^*(\boldsymbol{x}_{n-1}|\boldsymbol{x}_n)) = \frac{d}{2}\log(2\pi e\sigma_n^{*2}) - \mathbb{E}_q H(q(\boldsymbol{x}_{n-1}|\boldsymbol{x}_n)).$$

As a result,

$$\begin{aligned}
&D_{\mathrm{KL}}(q(\boldsymbol{x}_{0:N})||p^*(\boldsymbol{x}_{0:N})) \\
=&D_{\mathrm{KL}}(q(\boldsymbol{x}_N)||p(\boldsymbol{x}_N)) + \sum_{n=1}^{N}\frac{d}{2}\log(2\pi e\sigma_n^{*2}) - \sum_{n=1}^{N}\mathbb{E}_q H(q(\boldsymbol{x}_{n-1}|\boldsymbol{x}_n)) \\
&+ \sum_{n=1}^{N}\mathbb{E}_q H(q(\boldsymbol{x}_{n-1}|\boldsymbol{x}_n)) - (H(q(\boldsymbol{x}_{0:N})) - H(q(\boldsymbol{x}_N))) \\
=&H(q(\boldsymbol{x}_N), p(\boldsymbol{x}_N)) + \sum_{n=1}^{N}\frac{d}{2}\log(2\pi e\sigma_n^{*2}) - H(q(\boldsymbol{x}_{0:N})).
\end{aligned}$$

$\square$

**Lemma 10.** *(Marginal score function) Suppose $q(\boldsymbol{v}, \boldsymbol{w})$ is a probability distribution, then*

$$\nabla_{\boldsymbol{w}} \log q(\boldsymbol{w}) = \mathbb{E}_{q(\boldsymbol{v}|\boldsymbol{w})}\nabla_{\boldsymbol{w}} \log q(\boldsymbol{w}|\boldsymbol{v})$$

*Proof.* According to $\mathbb{E}_{q(\boldsymbol{v}|\boldsymbol{w})}\nabla_{\boldsymbol{w}}\log q(\boldsymbol{v}|\boldsymbol{w}) = \int \nabla_{\boldsymbol{w}}q(\boldsymbol{v}|\boldsymbol{w})\mathrm{d}\boldsymbol{v} = \nabla_{\boldsymbol{w}}\int q(\boldsymbol{v}|\boldsymbol{w})\mathrm{d}\boldsymbol{v} = \boldsymbol{0}$, we have

$$
\begin{aligned}
\nabla_{\boldsymbol{w}}\log q(\boldsymbol{w}) =& \nabla_{\boldsymbol{w}}\log q(\boldsymbol{w}) + \mathbb{E}_{q(\boldsymbol{v}|\boldsymbol{w})}\nabla_{\boldsymbol{w}}\log q(\boldsymbol{v}|\boldsymbol{w}) \\
=& \mathbb{E}_{q(\boldsymbol{v}|\boldsymbol{w})}\nabla_{\boldsymbol{w}}\log q(\boldsymbol{v},\boldsymbol{w}) = \mathbb{E}_{q(\boldsymbol{v}|\boldsymbol{w})}\nabla_{\boldsymbol{w}}\log q(\boldsymbol{w}|\boldsymbol{v}).
\end{aligned}
$$

$\square$

**Lemma 11.** *(Score representation of conditional expectation and covariance) Suppose $q(\boldsymbol{v},\boldsymbol{w}) = q(\boldsymbol{v})q(\boldsymbol{w}|\boldsymbol{v})$, where $q(\boldsymbol{w}|\boldsymbol{v}) = \mathcal{N}(\boldsymbol{w}|\sqrt{\alpha}\boldsymbol{v},\beta\boldsymbol{I})$, then*

$$
\mathbb{E}_{q(\boldsymbol{v}|\boldsymbol{w})}[\boldsymbol{v}] = \frac{1}{\sqrt{\alpha}}(\boldsymbol{w} + \beta\nabla_{\boldsymbol{w}}\log q(\boldsymbol{w})),
$$

$$
\mathbb{E}_{q(\boldsymbol{w})}\mathrm{Cov}_{q(\boldsymbol{v}|\boldsymbol{w})}[\boldsymbol{v}] = \frac{\beta}{\alpha}\left(\boldsymbol{I} - \beta\mathbb{E}_{q(\boldsymbol{w})}\left[\nabla_{\boldsymbol{w}}\log q(\boldsymbol{w})\nabla_{\boldsymbol{w}}\log q(\boldsymbol{w})^{\top}\right]\right),
$$

$$
\mathbb{E}_{q(\boldsymbol{w})}\frac{\mathrm{tr}(\mathrm{Cov}_{q(\boldsymbol{v}|\boldsymbol{w})}[\boldsymbol{v}])}{d} = \frac{\beta}{\alpha}\left(1 - \beta\mathbb{E}_{q(\boldsymbol{w})}\frac{||\nabla_{\boldsymbol{w}}\log q(\boldsymbol{w})||^2}{d}\right).
$$

*Proof.* According to Lemma 10, we have

$$
\nabla_{\boldsymbol{w}}\log q(\boldsymbol{w}) = \mathbb{E}_{q(\boldsymbol{v}|\boldsymbol{w})}\nabla_{\boldsymbol{w}}\log q(\boldsymbol{w}|\boldsymbol{v}) = -\mathbb{E}_{q(\boldsymbol{v}|\boldsymbol{w})}\frac{\boldsymbol{w} - \sqrt{\alpha}\boldsymbol{v}}{\beta}.
$$

Thereby, $\mathbb{E}_{q(\boldsymbol{v}|\boldsymbol{w})}[\boldsymbol{v}] = \frac{1}{\sqrt{\alpha}}(\boldsymbol{w} + \beta\nabla_{\boldsymbol{w}}\log q(\boldsymbol{w}))$. Furthermore, we have

$$
\begin{aligned}
&\mathbb{E}_{q(\boldsymbol{w})}\mathrm{Cov}_{q(\boldsymbol{v}|\boldsymbol{w})}[\boldsymbol{v}] = \frac{\beta^2}{\alpha}\mathbb{E}_{q(\boldsymbol{w})}\mathrm{Cov}_{q(\boldsymbol{v}|\boldsymbol{w})}[\frac{\boldsymbol{w} - \sqrt{\alpha}\boldsymbol{v}}{\beta}] \\
=&\frac{\beta^2}{\alpha}\mathbb{E}_{q(\boldsymbol{w})}\left(\mathbb{E}_{q(\boldsymbol{v}|\boldsymbol{w})}(\frac{\boldsymbol{w} - \sqrt{\alpha}\boldsymbol{v}}{\beta})(\frac{\boldsymbol{w} - \sqrt{\alpha}\boldsymbol{v}}{\beta})^{\top} - \mathbb{E}_{q(\boldsymbol{v}|\boldsymbol{w})}[\frac{\boldsymbol{w} - \sqrt{\alpha}\boldsymbol{v}}{\beta}]\mathbb{E}_{q(\boldsymbol{v}|\boldsymbol{w})}[\frac{\boldsymbol{w} - \sqrt{\alpha}\boldsymbol{v}}{\beta}]^{\top}\right) \\
=&\frac{\beta^2}{\alpha}\left(\frac{1}{\beta^2}\mathbb{E}_{q(\boldsymbol{v})}\mathbb{E}_{q(\boldsymbol{w}|\boldsymbol{v})}(\boldsymbol{w} - \sqrt{\alpha}\boldsymbol{v})(\boldsymbol{w} - \sqrt{\alpha}\boldsymbol{v})^{\top} - \mathbb{E}_{q(\boldsymbol{w})}\nabla_{\boldsymbol{w}}\log q(\boldsymbol{w})\nabla_{\boldsymbol{w}}\log q(\boldsymbol{w})^{\top}\right) \\
=&\frac{\beta^2}{\alpha}\left(\frac{1}{\beta^2}\mathbb{E}_{q(\boldsymbol{v})}\mathrm{Cov}_{q(\boldsymbol{w}|\boldsymbol{v})}\boldsymbol{w} - \mathbb{E}_{q(\boldsymbol{w})}\nabla_{\boldsymbol{w}}\log q(\boldsymbol{w})\nabla_{\boldsymbol{w}}\log q(\boldsymbol{w})^{\top}\right) \\
=&\frac{\beta^2}{\alpha}\left(\frac{1}{\beta^2}\mathbb{E}_{q(\boldsymbol{v})}\beta\boldsymbol{I} - \mathbb{E}_{q(\boldsymbol{w})}\nabla_{\boldsymbol{w}}\log q(\boldsymbol{w})\nabla_{\boldsymbol{w}}\log q(\boldsymbol{w})^{\top}\right) \\
=&\frac{\beta^2}{\alpha}\left(\frac{1}{\beta}\boldsymbol{I} - \mathbb{E}_{q(\boldsymbol{w})}\nabla_{\boldsymbol{w}}\log q(\boldsymbol{w})\nabla_{\boldsymbol{w}}\log q(\boldsymbol{w})^{\top}\right) = \frac{\beta}{\alpha}(\boldsymbol{I} - \beta\mathbb{E}_{q(\boldsymbol{w})}\nabla_{\boldsymbol{w}}\log q(\boldsymbol{w})\nabla_{\boldsymbol{w}}\log q(\boldsymbol{w})^{\top}).
\end{aligned}
$$

Taking the trace, we have

$$
\mathbb{E}_{q(\boldsymbol{w})}\frac{\mathrm{tr}(\mathrm{Cov}_{q(\boldsymbol{v}|\boldsymbol{w})}[\boldsymbol{v}])}{d} = \frac{\beta}{\alpha}(1 - \beta\mathbb{E}_{q(\boldsymbol{w})}\frac{||\nabla_{\boldsymbol{w}}\log q(\boldsymbol{w})||^2}{d}).
$$

$\square$

**Lemma 12.** *(Bounded covariance of a bounded distribution) Suppose $q(\boldsymbol{x})$ is a bounded distribution in $[a,b]^d$, then $\frac{\mathrm{tr}(\mathrm{Cov}_{q(\boldsymbol{x})}[\boldsymbol{x}])}{d} \leq (\frac{b-a}{2})^2$.*

*Proof.*

$$
\begin{aligned}
\frac{\mathrm{tr}(\mathrm{Cov}_{q(\boldsymbol{x})}[\boldsymbol{x}])}{d} &= \frac{\mathrm{tr}(\mathrm{Cov}_{q(\boldsymbol{x})}[\boldsymbol{x} - \frac{a+b}{2}])}{d} = \frac{\mathbb{E}_{q(\boldsymbol{x})}||\boldsymbol{x} - \frac{a+b}{2}||^2 - ||\mathbb{E}\boldsymbol{x} - \frac{a+b}{2}||^2}{d} \\
&\leq \frac{\mathbb{E}_{q(\boldsymbol{x})}||\boldsymbol{x} - \frac{a+b}{2}||^2}{d} \leq (\frac{b-a}{2})^2.
\end{aligned}
$$

$\square$

**Lemma 13.** *(Convert the moments of $q(\boldsymbol{x}_{n-1}|\boldsymbol{x}_n)$ to moments of $q(\boldsymbol{x}_0|\boldsymbol{x}_n)$) The optimal solution $\boldsymbol{\mu}_n^*(\boldsymbol{x}_n)$ and $\sigma_n^{*2}$ to Eq. (4) can be represented by the first two moments of $q(\boldsymbol{x}_0|\boldsymbol{x}_n)$*

$$\boldsymbol{\mu}_n^*(\boldsymbol{x}_n) = \tilde{\boldsymbol{\mu}}_n(\boldsymbol{x}_n, \mathbb{E}_{q(\boldsymbol{x}_0|\boldsymbol{x}_n)}\boldsymbol{x}_0)$$

$$\sigma_n^{*2} = \lambda_n^2 + \left( \sqrt{\overline{\alpha}_{n-1}} - \sqrt{\overline{\beta}_{n-1} - \lambda_n^2} \cdot \sqrt{\frac{\overline{\alpha}_n}{\overline{\beta}_n}} \right)^2 \mathbb{E}_{q(\boldsymbol{x}_n)} \frac{\mathrm{tr}(\mathrm{Cov}_{q(\boldsymbol{x}_0|\boldsymbol{x}_n)}[\boldsymbol{x}_0])}{d}$$

*where $q_n(\boldsymbol{x}_n)$ is the marginal distribution of the forward process at timestep $n$ and $d$ is the dimension of the data.*

*Proof.* According to Lemma 9, the optimal $\boldsymbol{\mu}_n^*$ and $\sigma_n^{*2}$ under KL minimization is

$$\boldsymbol{\mu}_n^*(\boldsymbol{x}_n) = \mathbb{E}_{q(\boldsymbol{x}_{n-1}|\boldsymbol{x}_n)}[\boldsymbol{x}_{n-1}], \quad \sigma_n^{*2} = \mathbb{E}_{q_n(\boldsymbol{x}_n)} \frac{\mathrm{tr}(\mathrm{Cov}_{q(\boldsymbol{x}_{n-1}|\boldsymbol{x}_n)}[\boldsymbol{x}_{n-1}])}{d}.$$

We further derive $\boldsymbol{\mu}_n^*$. Since $\tilde{\boldsymbol{\mu}}_n(\boldsymbol{x}_n, \boldsymbol{x}_0)$ is linear w.r.t. $\boldsymbol{x}_0$, we have

$$\boldsymbol{\mu}_n^*(\boldsymbol{x}_n) = \mathbb{E}_{q(\boldsymbol{x}_{n-1}|\boldsymbol{x}_n)}[\boldsymbol{x}_{n-1}] = \mathbb{E}_{q(\boldsymbol{x}_0|\boldsymbol{x}_n)}\mathbb{E}_{q(\boldsymbol{x}_{n-1}|\boldsymbol{x}_n,\boldsymbol{x}_0)}[\boldsymbol{x}_{n-1}]$$
$$= \mathbb{E}_{q(\boldsymbol{x}_0|\boldsymbol{x}_n)}\tilde{\boldsymbol{\mu}}_n(\boldsymbol{x}_n, \boldsymbol{x}_0) = \tilde{\boldsymbol{\mu}}_n(\boldsymbol{x}_n, \mathbb{E}_{q(\boldsymbol{x}_0|\boldsymbol{x}_n)}\boldsymbol{x}_0).$$

Then we consider $\sigma_n^{*2}$. According to the law of total variance, we have

$$\mathrm{Cov}_{q(\boldsymbol{x}_{n-1}|\boldsymbol{x}_n)}[\boldsymbol{x}_{n-1}] = \mathbb{E}_{q(\boldsymbol{x}_0|\boldsymbol{x}_n)}\mathrm{Cov}_{q(\boldsymbol{x}_{n-1}|\boldsymbol{x}_n,\boldsymbol{x}_0)}[\boldsymbol{x}_{n-1}] + \mathrm{Cov}_{q(\boldsymbol{x}_0|\boldsymbol{x}_n)}\mathbb{E}_{q(\boldsymbol{x}_{n-1}|\boldsymbol{x}_n,\boldsymbol{x}_0)}[\boldsymbol{x}_{n-1}]$$

$$= \lambda_n^2 \boldsymbol{I} + \mathrm{Cov}_{q(\boldsymbol{x}_0|\boldsymbol{x}_n)}\tilde{\boldsymbol{\mu}}_n(\boldsymbol{x}_n, \boldsymbol{x}_0) = \lambda_n^2 \boldsymbol{I} + (\sqrt{\overline{\alpha}_{n-1}} - \sqrt{\overline{\beta}_{n-1} - \lambda_n^2} \cdot \sqrt{\frac{\overline{\alpha}_n}{\overline{\beta}_n}})^2 \mathrm{Cov}_{q(\boldsymbol{x}_0|\boldsymbol{x}_n)}[\boldsymbol{x}_0].$$

Thereby,

$$\sigma_n^{*2} = \mathbb{E}_{q_n(\boldsymbol{x}_n)} \frac{\mathrm{tr}(\mathrm{Cov}_{q(\boldsymbol{x}_{n-1}|\boldsymbol{x}_n)}[\boldsymbol{x}_{n-1}])}{d}$$

$$= \lambda_n^2 + (\sqrt{\overline{\alpha}_{n-1}} - \sqrt{\overline{\beta}_{n-1} - \lambda_n^2} \cdot \sqrt{\frac{\overline{\alpha}_n}{\overline{\beta}_n}})^2 \mathbb{E}_{q(\boldsymbol{x}_n)} \frac{\mathrm{tr}(\mathrm{Cov}_{q(\boldsymbol{x}_0|\boldsymbol{x}_n)}[\boldsymbol{x}_0])}{d}.$$

$\square$

### A.2 PROOF OF THEOREM 1

**Theorem 1.** *(Score representation of the optimal solution to Eq. (4), proof in Appendix A.2)*

*The optimal solution $\boldsymbol{\mu}_n^*(\boldsymbol{x}_n)$ and $\sigma_n^{*2}$ to Eq. (4) are*

$$\boldsymbol{\mu}_n^*(\boldsymbol{x}_n) = \tilde{\boldsymbol{\mu}}_n \left( \boldsymbol{x}_n, \frac{1}{\sqrt{\overline{\alpha}_n}}(\boldsymbol{x}_n + \overline{\beta}_n \nabla_{\boldsymbol{x}_n} \log q_n(\boldsymbol{x}_n)) \right), \tag{6}$$

$$\sigma_n^{*2} = \lambda_n^2 + \left( \sqrt{\frac{\overline{\beta}_n}{\alpha_n}} - \sqrt{\overline{\beta}_{n-1} - \lambda_n^2} \right)^2 \left( 1 - \overline{\beta}_n \mathbb{E}_{q_n(\boldsymbol{x}_n)} \frac{||\nabla_{\boldsymbol{x}_n} \log q_n(\boldsymbol{x}_n)||^2}{d} \right), \tag{7}$$

*where $q_n(\boldsymbol{x}_n)$ is the marginal distribution of the forward process at the timestep $n$ and $d$ is the dimension of the data.*

*Proof.* According to Lemma 11 and Lemma 13, we have

$$\boldsymbol{\mu}_n^*(\boldsymbol{x}_n) = \tilde{\boldsymbol{\mu}}_n(\boldsymbol{x}_n, \mathbb{E}_{q(\boldsymbol{x}_0|\boldsymbol{x}_n)}\boldsymbol{x}_0) = \tilde{\boldsymbol{\mu}}_n(\boldsymbol{x}_n, \frac{1}{\sqrt{\overline{\alpha}_n}}(\boldsymbol{x}_n + \overline{\beta}_n \nabla_{\boldsymbol{x}_n} \log q(\boldsymbol{x}_n))),$$

and

$$\sigma_n^{*2} = \lambda_n^2 + (\sqrt{\overline{\alpha}_{n-1}} - \sqrt{\overline{\beta}_{n-1} - \lambda_n^2} \cdot \sqrt{\frac{\overline{\alpha}_n}{\overline{\beta}_n}})^2 \mathbb{E}_{q(\boldsymbol{x}_n)} \frac{\mathrm{tr}(\mathrm{Cov}_{q(\boldsymbol{x}_0|\boldsymbol{x}_n)}[\boldsymbol{x}_0])}{d}$$

$$=\lambda_n^2 + (\sqrt{\overline{\alpha}_{n-1}} - \sqrt{\overline{\beta}_{n-1} - \lambda_n^2} \cdot \sqrt{\frac{\overline{\alpha}_n}{\overline{\beta}_n}})^2 \frac{\overline{\beta}_n}{\overline{\alpha}_n}(1 - \overline{\beta}_n \mathbb{E}_{q(\boldsymbol{x}_n)} \frac{||\nabla_{\boldsymbol{x}_n} \log q(\boldsymbol{x}_n)||^2}{d})$$

$$=\lambda_n^2 + (\sqrt{\frac{\overline{\beta}_n}{\alpha_n}} - \sqrt{\overline{\beta}_{n-1} - \lambda_n^2})^2(1 - \overline{\beta}_n \mathbb{E}_{q(\boldsymbol{x}_n)} \frac{||\nabla_{\boldsymbol{x}_n} \log q(\boldsymbol{x}_n)||^2}{d}).$$

$\square$

## A.3 PROOF OF THEOREM 2

**Theorem 2.** *(Bounds of the optimal reverse variance, proof in Appendix A.3)*

$\sigma_n^{*2}$ *has the following lower and upper bounds:*

$$\lambda_n^2 \leq \sigma_n^{*2} \leq \lambda_n^2 + \left(\sqrt{\frac{\overline{\beta}_n}{\alpha_n}} - \sqrt{\overline{\beta}_{n-1} - \lambda_n^2}\right)^2. \tag{11}$$

*If we further assume $q(\boldsymbol{x}_0)$ is a bounded distribution in $[a, b]^d$, where $d$ is the dimension of data, then $\sigma_n^{*2}$ can be further upper bounded by*

$$\sigma_n^{*2} \leq \lambda_n^2 + \left(\sqrt{\overline{\alpha}_{n-1}} - \sqrt{\overline{\beta}_{n-1} - \lambda_n^2} \cdot \sqrt{\frac{\overline{\alpha}_n}{\overline{\beta}_n}}\right)^2 \left(\frac{b-a}{2}\right)^2. \tag{12}$$

*Proof.* According to Lemma 13 and Theorem 1, we have

$$\lambda_n^2 \leq \sigma_n^{*2} \leq \lambda_n^2 + (\sqrt{\frac{\overline{\beta}_n}{\alpha_n}} - \sqrt{\overline{\beta}_{n-1} - \lambda_n^2})^2.$$

If we further $q(\boldsymbol{x}_0)$ assume is a bounded distribution in $[a, b]^d$, then $q(\boldsymbol{x}_0|\boldsymbol{x}_n)$ is also a bounded distribution in $[a, b]^d$. According to Lemma 12, we have

$$\mathbb{E}_{q(\boldsymbol{x}_n)} \frac{\mathrm{tr}(\mathrm{Cov}_{q(\boldsymbol{x}_0|\boldsymbol{x}_n)}[\boldsymbol{x}_0])}{d} \leq (\frac{b-a}{2})^2.$$

Combining with Lemma 13, we have

$$\sigma_n^{*2} = \lambda_n^2 + (\sqrt{\overline{\alpha}_{n-1}} - \sqrt{\overline{\beta}_{n-1} - \lambda_n^2} \cdot \sqrt{\frac{\overline{\alpha}_n}{\overline{\beta}_n}})^2 \mathbb{E}_{q(\boldsymbol{x}_n)} \frac{\mathrm{tr}(\mathrm{Cov}_{q(\boldsymbol{x}_0|\boldsymbol{x}_n)}[\boldsymbol{x}_0])}{d}$$

$$\leq \lambda_n^2 + (\sqrt{\overline{\alpha}_{n-1}} - \sqrt{\overline{\beta}_{n-1} - \lambda_n^2} \cdot \sqrt{\frac{\overline{\alpha}_n}{\overline{\beta}_n}})^2(\frac{b-a}{2})^2.$$

$\square$

## A.4 PROOF OF THE DECOMPOSED OPTIMAL KL

**Theorem 3.** *(Decomposed optimal KL, proof in Appendix A.4)*

*The KL divergence between the shorter forward process and its optimal reverse process is*

$$D_{\mathrm{KL}}(q(\boldsymbol{x}_0, \boldsymbol{x}_{\tau_1}, \cdots, \boldsymbol{x}_{\tau_K})||p^*(\boldsymbol{x}_0, \boldsymbol{x}_{\tau_1}, \cdots, \boldsymbol{x}_{\tau_K})) = \frac{d}{2}\sum_{k=2}^{K} J(\tau_{k-1}, \tau_k) + c,$$

*where $J(\tau_{k-1}, \tau_k) = \log \frac{\sigma_{\tau_{k-1}|\tau_k}^{*2}}{\lambda_{\tau_{k-1}|\tau_k}^2}$ and $c$ is a constant unrelated to the trajectory $\tau$.*

*Proof.* According to Lemma 7 and Lemma 9, we have

$$
\begin{aligned}
&D_{\mathrm{KL}}(q(\boldsymbol{x}_0, \boldsymbol{x}_{\tau_1}, \cdots, \boldsymbol{x}_{\tau_K}) || p^*(\boldsymbol{x}_0, \boldsymbol{x}_{\tau_1}, \cdots, \boldsymbol{x}_{\tau_K})) \\
=&\mathbb{E}_q D_{\mathrm{KL}}(q(\boldsymbol{x}_0 | \boldsymbol{x}_{\tau_1}, \cdots, \boldsymbol{x}_{\tau_K}) || p^*(\boldsymbol{x}_0 | \boldsymbol{x}_1)) + D_{\mathrm{KL}}(q(\boldsymbol{x}_{\tau_1}, \cdots, \boldsymbol{x}_{\tau_K}) || p^*(\boldsymbol{x}_{\tau_1}, \cdots, \boldsymbol{x}_{\tau_K})) \\
=&\mathbb{E}_q D_{\mathrm{KL}}(q(\boldsymbol{x}_0 | \boldsymbol{x}_{\tau_1}, \cdots, \boldsymbol{x}_{\tau_K}) || p^*(\boldsymbol{x}_0 | \boldsymbol{x}_1)) + H(q(\boldsymbol{x}_N), p(\boldsymbol{x}_N)) \\
&+ \frac{d}{2} \sum_{k=2}^K \log(2\pi e \sigma^{*2}_{\tau_{k-1} | \tau_k}) - H(q(\boldsymbol{x}_{\tau_1}, \cdots, \boldsymbol{x}_{\tau_N})) \\
=&- \mathbb{E}_q \log p^*(\boldsymbol{x}_0 | \boldsymbol{x}_1) + H(q(\boldsymbol{x}_N), p(\boldsymbol{x}_N)) + \frac{d}{2} \sum_{k=2}^K \log(2\pi e \sigma^{*2}_{\tau_{k-1} | \tau_k}) - H(q(\boldsymbol{x}_0, \boldsymbol{x}_{\tau_1}, \cdots, \boldsymbol{x}_{\tau_K})) \\
=&- \mathbb{E}_q \log p^*(\boldsymbol{x}_0 | \boldsymbol{x}_1) + H(q(\boldsymbol{x}_N), p(\boldsymbol{x}_N)) + \frac{d}{2} \sum_{k=2}^K \log(2\pi e \sigma^{*2}_{\tau_{k-1} | \tau_k}) \\
&- H(q(\boldsymbol{x}_0)) - \frac{d}{2} \log(2\pi e \overline{\beta}_N) - \frac{d}{2} \sum_{k=2}^K \log(2\pi e \lambda^2_{\tau_{k-1} | \tau_k}) \\
=&- \mathbb{E}_q \log p^*(\boldsymbol{x}_0 | \boldsymbol{x}_1) + H(q(\boldsymbol{x}_N), p(\boldsymbol{x}_N)) + \frac{d}{2} \sum_{k=2}^K \log \frac{\sigma^{*2}_{\tau_{k-1} | \tau_k}}{\lambda^2_{\tau_{k-1} | \tau_k}} - H(q(\boldsymbol{x}_0)) - \frac{d}{2} \log(2\pi e \overline{\beta}_N).
\end{aligned}
$$

Let $J(\tau_{k-1}, \tau_k) = \log \frac{\sigma^{*2}_{\tau_{k-1} | \tau_k}}{\lambda^2_{\tau_{k-1} | \tau_k}}$ and $c = -\mathbb{E}_q \log p^*(\boldsymbol{x}_0 | \boldsymbol{x}_1) + H(q(\boldsymbol{x}_N), p(\boldsymbol{x}_N)) - H(q(\boldsymbol{x}_0)) - \frac{d}{2} \log(2\pi e \overline{\beta}_N)$, then $c$ is a constant unrelated to the trajectory $\tau$ and

$$
D_{\mathrm{KL}}(q(\boldsymbol{x}_0, \boldsymbol{x}_{\tau_1}, \cdots, \boldsymbol{x}_{\tau_K}) || p^*(\boldsymbol{x}_0, \boldsymbol{x}_{\tau_1}, \cdots, \boldsymbol{x}_{\tau_K})) = \frac{d}{2} \sum_{k=2}^K J(\tau_{k-1}, \tau_k) + c.
$$

$\square$

## A.5 THE FORMAL RESULT FOR SECTION 5 AND ITS PROOF

Here we present the formal result of the relationship between the score function and the data covariance matrix mentioned in Section 5.

**Proposition 1.** *(Proof in Appendix A.5) The expected conditional covariance matrix of the data distribution is determined by the score function $\nabla_{\boldsymbol{x}_n} \log q_n(\boldsymbol{x}_n)$ as follows:*

$$
\mathbb{E}_{q(\boldsymbol{x}_n)} \mathrm{Cov}_{q(\boldsymbol{x}_0 | \boldsymbol{x}_n)}[\boldsymbol{x}_0] = \frac{\overline{\beta}_n}{\overline{\alpha}_n} \left( \boldsymbol{I} - \overline{\beta}_n \mathbb{E}_{q_n(\boldsymbol{x}_n)} \left[ \nabla_{\boldsymbol{x}_n} \log q_n(\boldsymbol{x}_n) \nabla_{\boldsymbol{x}_n} \log q_n(\boldsymbol{x}_n)^\top \right] \right), \quad (15)
$$

*which contributes to the data covariance matrix according to the law of total variance*

$$
\mathrm{Cov}_{q(\boldsymbol{x}_0)}[\boldsymbol{x}_0] = \mathbb{E}_{q(\boldsymbol{x}_n)} \mathrm{Cov}_{q(\boldsymbol{x}_0 | \boldsymbol{x}_n)}[\boldsymbol{x}_0] + \mathrm{Cov}_{q(\boldsymbol{x}_n)} \mathbb{E}_{q(\boldsymbol{x}_0 | \boldsymbol{x}_n)}[\boldsymbol{x}_0]. \quad (16)
$$

*Proof.* Since $q(\boldsymbol{x}_n | \boldsymbol{x}_0) = \mathcal{N}(\boldsymbol{x}_n | \sqrt{\overline{\alpha}_n} \boldsymbol{x}_0, \overline{\beta}_n \boldsymbol{I})$, according to Lemma 11, we have

$$
\mathbb{E}_{q(\boldsymbol{x}_n)} \mathrm{Cov}_{q(\boldsymbol{x}_0 | \boldsymbol{x}_n)}[\boldsymbol{x}_0] = \frac{\overline{\beta}_n}{\overline{\alpha}_n} (\boldsymbol{I} - \overline{\beta}_n \mathbb{E}_{q_n(\boldsymbol{x}_n)} \nabla_{\boldsymbol{x}_n} \log q_n(\boldsymbol{x}_n) \nabla_{\boldsymbol{x}_n} \log q_n(\boldsymbol{x}_n)^\top).
$$

The law of total variance is a classical result in statistics. Here we prove it for completeness:

$$
\begin{aligned}
&\mathbb{E}_{q(\boldsymbol{x}_n)} \mathrm{Cov}_{q(\boldsymbol{x}_0 | \boldsymbol{x}_n)}[\boldsymbol{x}_0] + \mathrm{Cov}_{q(\boldsymbol{x}_n)} \mathbb{E}_{q(\boldsymbol{x}_0 | \boldsymbol{x}_n)}[\boldsymbol{x}_0] \\
=&\mathbb{E}_{q(\boldsymbol{x}_n)} \left( \mathbb{E}_{q(\boldsymbol{x}_0 | \boldsymbol{x}_n)} \boldsymbol{x}_0 \boldsymbol{x}_0^\top - \mathbb{E}_{q(\boldsymbol{x}_0 | \boldsymbol{x}_n)}[\boldsymbol{x}_0] \mathbb{E}_{q(\boldsymbol{x}_0 | \boldsymbol{x}_n)}[\boldsymbol{x}_0]^\top \right) \\
&+ \mathbb{E}_{q(\boldsymbol{x}_n)} \left( \mathbb{E}_{q(\boldsymbol{x}_0 | \boldsymbol{x}_n)}[\boldsymbol{x}_0] \mathbb{E}_{q(\boldsymbol{x}_0 | \boldsymbol{x}_n)}[\boldsymbol{x}_0]^\top \right) - \left( \mathbb{E}_{q(\boldsymbol{x}_n)} \mathbb{E}_{q(\boldsymbol{x}_0 | \boldsymbol{x}_n)}[\boldsymbol{x}_0] \right) \left( \mathbb{E}_{q(\boldsymbol{x}_n)} \mathbb{E}_{q(\boldsymbol{x}_0 | \boldsymbol{x}_n)}[\boldsymbol{x}_0] \right)^\top \\
=&\mathbb{E}_{q(\boldsymbol{x}_0)} \boldsymbol{x}_0 \boldsymbol{x}_0^\top - \mathbb{E}_{q(\boldsymbol{x}_0)}[\boldsymbol{x}_0] \mathbb{E}_{q(\boldsymbol{x}_0)}[\boldsymbol{x}_0]^\top = \mathrm{Cov}_{q(\boldsymbol{x}_0)}[\boldsymbol{x}_0].
\end{aligned}
$$

$\square$

---

**Algorithm 1** The DP algorithm for the least-cost-path problem (Watson et al., 2021)

---

1: **Input:** Cost function $J(s,t)$ and integers $K, N$ ($1 \leq K \leq N$)
2: **Output:** The least-cost-trajectory $1 = \tau_1 < \cdots < \tau_K = N$
3: $C \leftarrow \{\infty\}_{1 \leq k,n \leq N}, D \leftarrow \{-1\}_{1 \leq k,n \leq N}$
4: $C[1,1] \leftarrow 0$
5: **for** $k = 2$ **to** $K$ **do**                                      ▷ Calculate $C$ and $D$
6:      $CJ \leftarrow \{C[k-1,s] + J(s,n)\}_{1 \leq s < N, k \leq n \leq N}$
7:      $C[k, k :] \leftarrow (\min(CJ[:,k]), \min(CJ[:,k+1]), \cdots, \min(CJ[:,N]))$
8:      $D[k, k :] \leftarrow (\arg\min(CJ[:,k]), \arg\min(CJ[:,k+1]), \cdots, \arg\min(CJ[:,N]))$
9: **end for**
10: $\tau_K = N$
11: **for** $k = K$ **to** $2$ **do**                                      ▷ Calculate $\tau$
12:      $\tau_{k-1} \leftarrow D[k, \tau_k]$
13: **end for**
14: **return** $\tau$

---

## B   THE DP ALGORITHM FOR THE LEAST-COST-PATH PROBLEM

Given a cost function $J(s,t)$ with $1 \leq s < t$ and $k, n \geq 1$, we want to find a trajectory $1 = \tau_1 < \cdots < \tau_k = n$ of $k$ nodes starting from 1 and terminating at $n$, s.t., the total cost $J(\tau_1, \tau_2) + J(\tau_2, \tau_3) + \cdots + J(\tau_{k-1}, \tau_k)$ is minimized. Such a problem can be solved by the DP algorithm proposed by Watson et al. (2021). Let $C[k,n]$ be the minimized cost of the optimal trajectory, and $D[k,n]$ be the $\tau_{k-1}$ of the optimal trajectory. For simplicity, we also let $J(s,t) = \infty$ for $s \geq t \geq 1$. Then for $k = 1$, we have $C[1,n] = \begin{cases} 0 & n = 1 \\ \infty & N \geq n > 1 \end{cases}$ and $D[1,n] = -1$ (here $\infty$ and $-1$ represent undefined values for simplicity). For $N \geq k \geq 2$, we have

$$C[k,n] = \begin{cases} \infty & 1 \leq n < k \\ \min_{k-1 \leq s \leq n-1} C[k-1,s] + J(s,n) = \min_{1 \leq s \leq N} C[k-1,s] + J(s,n) & N \geq n \geq k, \end{cases}$$

$$D[k,n] = \begin{cases} -1 & 1 \leq n < k \\ \arg\min_{k-1 \leq s \leq n-1} C[k-1,s] + J(s,n) = \arg\min_{1 \leq s \leq N} C[k-1,s] + J(s,n) & N \geq n \geq k. \end{cases}$$

As long as $D$ is calculated, we can get the optimal trajectory recursively by $\tau_K = N$ and $\tau_{k-1} = D[k, \tau_k]$. We summarize the algorithm in Algorithm 1.

## C   THE BOUNDS OF THE OPTIMAL REVERSE VARIANCE CONSTRAINED ON A TRAJECTORY

In Section 4, we derive the optimal reverse variance constrained on a trajectory. Indeed, the optimal reverse variance can also be bounded similar to Theorem 2. We formalize it in Corollary 1.

**Corollary 1.** *(Bounds of the optimal reverse variance constrained on a trajectory)*

$\sigma^{*2}_{\tau_{k-1}|\tau_k}$ *has the following lower and upper bounds:*

$$\lambda^2_{\tau_{k-1}|\tau_k} \leq \sigma^{*2}_{\tau_{k-1}|\tau_k} \leq \lambda^2_{\tau_{k-1}|\tau_k} + \left( \sqrt{\frac{\overline{\beta}_{\tau_k}}{\alpha_{\tau_k|\tau_{k-1}}}} - \sqrt{\overline{\beta}_{\tau_{k-1}} - \lambda^2_{\tau_{k-1}|\tau_k}} \right)^2.$$

*If we further assume $q(\boldsymbol{x}_0)$ is a bounded distribution in $[a,b]^d$, where $d$ is the dimension of data, then $\sigma^{*2}_n$ can be further upper bounded by*

$$\sigma^{*2}_{\tau_{k-1}|\tau_k} \leq \lambda^2_{\tau_{k-1}|\tau_k} + \left( \sqrt{\overline{\alpha}_{\tau_{k-1}}} - \sqrt{\overline{\beta}_{\tau_{k-1}} - \lambda^2_{\tau_{k-1}|\tau_k}} \cdot \sqrt{\frac{\overline{\alpha}_{\tau_k}}{\overline{\beta}_{\tau_k}}} \right)^2 (\frac{b-a}{2})^2.$$

## D  SIMPLIFIED RESULTS FOR THE DDPM FORWARD PROCESS

The optimal solution $\boldsymbol{\mu}_n^*(\boldsymbol{x}_n)$ and $\sigma_n^{*2}$ in Theorem 1 and the bounds of $\sigma_n^{*2}$ in Theorem 2 can be directly simplified for the DDPM forward process by letting $\lambda_n^2 = \tilde{\beta}_n$. We list the simplified results in Corollary 2 and Corollary 3.

**Corollary 2.** *(Simplified score representation of the optimal solution)*

*When $\lambda_n^2 = \tilde{\beta}_n$, the optimal solution $\boldsymbol{\mu}_n^*(\boldsymbol{x}_n)$ and $\sigma_n^{*2}$ to Eq. (4) are*

$$\boldsymbol{\mu}_n^*(\boldsymbol{x}_n) = \frac{1}{\sqrt{\alpha_n}}(\boldsymbol{x}_n + \beta_n \nabla_{\boldsymbol{x}_n} \log q_n(\boldsymbol{x}_n)),$$

$$\sigma_n^{*2} = \frac{\beta_n}{\alpha_n}(1 - \beta_n \mathbb{E}_{q_n(\boldsymbol{x}_n)} \frac{||\nabla_{\boldsymbol{x}_n} \log q_n(\boldsymbol{x}_n)||^2}{d}).$$

**Corollary 3.** *(Simplified bounds of the optimal reverse variance)*

*When $\lambda_n^2 = \tilde{\beta}_n$, $\sigma_n^{*2}$ has the following lower and upper bounds:*

$$\tilde{\beta}_n \leq \sigma_n^{*2} \leq \frac{\beta_n}{\alpha_n}.$$

*If we further assume $q(\boldsymbol{x}_0)$ is a bounded distribution in $[a, b]^d$, where $d$ is the dimension of data, then $\sigma_n^{*2}$ can be further upper bounded by*

$$\sigma_n^{*2} \leq \tilde{\beta}_n + \frac{\overline{\alpha}_{n-1}\beta_n^2}{\overline{\beta}_n^2}\left(\frac{b-a}{2}\right)^2.$$

As for the shorter forward process defined in Eq. (13), it also includes the DDPM as a special case when $\lambda_{\tau_{k-1}|\tau_k}^2 = \tilde{\beta}_{\tau_{k-1}|\tau_k}$, where $\tilde{\beta}_{\tau_{k-1}|\tau_k} := \frac{\overline{\beta}_{\tau_{k-1}}}{\overline{\beta}_{\tau_k}}\beta_{\tau_k|\tau_{k-1}}$. Similar to Corollary 2, the optimal mean and variance of its reverse process can also be simplified for DDPMs by letting $\lambda_{\tau_{k-1}|\tau_k}^2 = \tilde{\beta}_{\tau_{k-1}|\tau_k}$. Formally, the simplified optimal mean and variance are

$$\boldsymbol{\mu}_{\tau_{k-1}|\tau_k}^*(\boldsymbol{x}_{\tau_k}) = \frac{1}{\sqrt{\alpha_{\tau_k|\tau_{k-1}}}}(\boldsymbol{x}_{\tau_k} + \beta_{\tau_k|\tau_{k-1}} \nabla_{\boldsymbol{x}_{\tau_k}} \log q_{\tau_k}(\boldsymbol{x}_{\tau_k})),$$

$$\sigma_{\tau_{k-1}|\tau_k}^{*2} = \frac{\beta_{\tau_k|\tau_{k-1}}}{\alpha_{\tau_k|\tau_{k-1}}}(1 - \beta_{\tau_k|\tau_{k-1}} \mathbb{E}_{q_{\tau_k}(\boldsymbol{x}_{\tau_k})} \frac{||\nabla_{\boldsymbol{x}_{\tau_k}} \log q_{\tau_k}(\boldsymbol{x}_{\tau_k})||^2}{d}).$$

Besides, Theorem 3 can also be simplified for DDPMs. We list the simplified result in Corollary 4.

**Corollary 4.** *(Simplified decomposed optimal KL)*

*When $\lambda_n^2 = \tilde{\beta}_n$, the KL divergence between the subprocess and its optimal reverse process is*

$$D_{\mathrm{KL}}(q(\boldsymbol{x}_0, \boldsymbol{x}_{\tau_1}, \cdots, \boldsymbol{x}_{\tau_K})||p^*(\boldsymbol{x}_0, \boldsymbol{x}_{\tau_1}, \cdots, \boldsymbol{x}_{\tau_K})) = \frac{d}{2}\sum_{k=2}^{K} J(\tau_{k-1}, \tau_k) + c,$$

$$where\ J(\tau_{k-1}, \tau_k) = \log(1 - \beta_{\tau_k|\tau_{k-1}} \mathbb{E}_{q_{\tau_k}(\boldsymbol{x}_{\tau_k})} \frac{||\nabla_{\boldsymbol{x}_{\tau_k}} \log q_{\tau_k}(\boldsymbol{x}_{\tau_k})||^2}{d}),$$

*and $c$ is a constant unrelated to the trajectory $\tau$.*

## E  EXTENSION TO DIFFUSION PROCESS WITH CONTINUOUS TIMESTEPS

Song et al. (2020b) generalizes the diffusion process to continuous timesteps by introducing a stochastic differential equation (SDE) $d\boldsymbol{z} = f(t)\boldsymbol{z}dt + g(t)d\boldsymbol{w}$. Without loss of generality, we consider the parameterization of $f(t)$ and $g(t)$ introduced by Kingma et al. (2021)

$$f(t) = \frac{1}{2}\frac{d\log\overline{\alpha}_t}{dt}, \quad g(t)^2 = \frac{d\overline{\beta}_t}{dt} - \frac{d\log\overline{\alpha}_t}{dt}\overline{\beta}_t,$$

where $\overline{\alpha}_t$ and $\overline{\beta}_t$ are scalar-valued functions satisfying some regular conditions (Kingma et al., 2021) with domain $t \in [0, 1]$. Such a parameterization induces a diffusion process on continuous timesteps $q(\boldsymbol{x}_0, \boldsymbol{z}_{[0,1]})$, s.t.,

$$q(\boldsymbol{z}_t|\boldsymbol{x}_0) = \mathcal{N}(\boldsymbol{z}_t|\sqrt{\overline{\alpha}_t}\boldsymbol{x}_0, \overline{\beta}_t\boldsymbol{I}), \quad \forall t \in [0, 1],$$
$$q(\boldsymbol{z}_t|\boldsymbol{z}_s) = \mathcal{N}(\boldsymbol{z}_t|\sqrt{\alpha_{t|s}}\boldsymbol{z}_s, \beta_{t|s}\boldsymbol{I}), \quad \forall 0 \le s < t \le 1,$$

where $\alpha_{t|s} := \overline{\alpha}_t/\overline{\alpha}_s$ and $\beta_{t|s} := \overline{\beta}_t - \alpha_{t|s}\overline{\beta}_s$.

### E.1 ANALYTIC ESTIMATE OF THE OPTIMAL REVERSE VARIANCE

Kingma et al. (2021) introduce $p(\boldsymbol{z}_s|\boldsymbol{z}_t) = \mathcal{N}(\boldsymbol{z}_s|\boldsymbol{\mu}_{s|t}(\boldsymbol{z}_t), \sigma^2_{s|t})$ $(s < t)$ to reverse from timestep $t$ to timestep $s$, where $\sigma^2_{s|t}$ is fixed to $\frac{\overline{\beta}_s}{\overline{\beta}_t}\beta_{s|t}$. In contrast, we show that $\sigma^2_{s|t}$ also has an optimal solution in an analytic form of the score function under the sense of KL minimization. According to Lemma 9 and Lemma 11, we have

$$\boldsymbol{\mu}^*_{s|t}(\boldsymbol{z}_t) = \mathbb{E}_{q(\boldsymbol{z}_s|\boldsymbol{z}_t)}[\boldsymbol{z}_s] = \frac{1}{\sqrt{\alpha_{t|s}}}(\boldsymbol{z}_t + \beta_{t|s}\nabla_{\boldsymbol{z}_t}\log q(\boldsymbol{z}_t)),$$

$$\sigma^{*2}_{s|t} = \mathbb{E}_q \frac{\text{tr}(\text{Cov}_{q(\boldsymbol{z}_s|\boldsymbol{z}_t)}[\boldsymbol{z}_s])}{d} = \frac{\beta_{t|s}}{\alpha_{t|s}}(1 - \beta_{t|s}\mathbb{E}_{q(\boldsymbol{z}_t)}\frac{||\nabla_{\boldsymbol{z}_t}\log q(\boldsymbol{z}_t)||^2}{d}).$$

Thereby, both the optimal mean and variance have a closed form expression w.r.t. the score function. In this case, we first estimate the expected mean squared norm of the score function by $\Gamma_t$ for $t \in [0, 1]$, where

$$\Gamma_t = \mathbb{E}_{q(\boldsymbol{z}_t)}\frac{||\boldsymbol{s}_t(\boldsymbol{z}_t)||^2}{d}.$$

Notice that there are infinite timesteps in $[0, 1]$. In practice, we can only choose a finite number of timesteps $0 = t_1 < \cdots < t_N = 1$ and calculate $\Gamma_{t_n}$. For a timestep $t$ between $t_{n-1}$ and $t_n$, we can use a linear interpolation between $\Gamma_{t_{n-1}}$ and $\Gamma_{t_n}$. Then, we can estimate $\sigma^{*2}_{s|t}$ by

$$\hat{\sigma}^2_{s|t} = \frac{\beta_{t|s}}{\alpha_{t|s}}(1 - \beta_{t|s}\Gamma_t).$$

### E.2 ANALYTIC ESTIMATION OF THE OPTIMAL REVERSE TRAJECTORY

Now we consider optimize the trajectory $0 = \tau_1 < \cdots < \tau_K = 1$ in the sense of KL minimization

$$\min_{\tau_1, \cdots, \tau_K} D_{\text{KL}}(q(\boldsymbol{x}_0, \boldsymbol{z}_{\tau_1}, \cdots, \boldsymbol{z}_{\tau_K})||p^*(\boldsymbol{x}_0, \boldsymbol{z}_{\tau_1}, \cdots, \boldsymbol{z}_{\tau_K})).$$

Similar to Theorem 3, the optimal KL is

$$D_{\text{KL}}(q(\boldsymbol{x}_0, \boldsymbol{z}_{\tau_1}, \cdots, \boldsymbol{z}_{\tau_K})||p^*(\boldsymbol{x}_0, \boldsymbol{z}_{\tau_1}, \cdots, \boldsymbol{z}_{\tau_K})) = \frac{d}{2}\sum_{k=2}^{K} J(\tau_{k-1}, \tau_k) + c,$$

where $J(\tau_{k-1}, \tau_k) = \log(1 - \beta_{\tau_k|\tau_{k-1}}\mathbb{E}_q \frac{||\nabla_{\boldsymbol{z}_{\tau_k}}\log q(\boldsymbol{z}_{\tau_k})||^2}{d})$ and $c$ is unrelated to $\tau$. The difference is that $J(s, t)$ is defined on a continuous range $0 \le s < t \le 1$ and the DP algorithm is not directly applicable. However, we can restrict $J(s, t)$ on a finite number of timesteps $0 = t_1 < \cdots < t_N = 1$. Then we can apply the DP algorithm (see Algorithm 1) to the restricted $J(s, t)$.

# F  EXPERIMENTAL DETAILS

## F.1  DETAILS OF SCORE-BASED MODELS

The CelebA 64x64 pretrained score-based model is provided in the official code (`https://github.com/ermongroup/ddim`) of Song et al. (2020a). The LSUN Bedroom pretrained score-based model is provided in the official code (`https://github.com/hojonathanho/diffusion`) of Ho et al. (2020). Both of them have a total of $N = 1000$ timesteps and use the linear schedule (Ho et al., 2020) as the forward noise schedule.

The ImageNet 64x64 pretrained score-based model is the unconditional $L_{\text{hybrid}}$ model provided in the official code (`https://github.com/openai/improved-diffusion`) of Nichol & Dhariwal (2021). The model includes both the mean and variance networks, where the mean network is trained with Eq. (5) as the standard DDPM (Ho et al., 2020) and the variance network is trained with the $L_{\text{vb}}$ objective. We only use the mean network. The model has a total of $N = 4000$ timesteps and its forward noise schedule is the cosine schedule (Nichol & Dhariwal, 2021).

The CIFAR10 score-based models are trained by ourselves. They have a total of $N = 1000$ timesteps and are trained with the linear forward noise schedule and the cosine forward noise schedule respectively. We use the same U-Net model architecture to Nichol & Dhariwal (2021). Following Nichol & Dhariwal (2021), we train 500K iterations with a batch size of 128, use a learning rate of 0.0001 with the AdamW optimizer (Loshchilov & Hutter, 2017) and use an exponential moving average (EMA) with a rate of 0.9999. We save a checkpoint every 10K iterations and select the checkpoint according to the FID results on 1000 samples generated under the reverse variance $\sigma_n^2 = \beta_n$ and full timesteps.

## F.2  LOG-LIKELIHOOD AND SAMPLING

Following Ho et al. (2020), we linearly scale the image data consisting of integers in $\{0, 1, \cdots, 255\}$ to $[-1, 1]$, and discretize the last reverse Markov transition $p(\boldsymbol{x}_0|\boldsymbol{x}_1)$ to obtain discrete log-likelihoods for image data.

Following Ho et al. (2020), at the end of sampling, we only display the mean of $p(\boldsymbol{x}_0|\boldsymbol{x}_1)$ and discard the noise. This is equivalent to setting a clipping threshold of zero for the noise scale $\sigma_1$. Inspired by this, when sampling, we also clip the noise scale $\sigma_2$ of $p(\boldsymbol{x}_1|\boldsymbol{x}_2)$, such that $\mathbb{E}|\sigma_2\epsilon| \leq \frac{2}{255}y$, where $\epsilon$ is the standard Gaussian noise and $y$ is the maximum tolerated perturbation of a channel. It improves the sample quality, especially for our analytic estimate (see Appendix G.4). We clip $\sigma_2$ for all methods compared in Table 2, and choose $y \in \{1, 2\}$ according to the FID score. We find $y = 2$ works better on CIFAR10 (LS) and CelebA 64x64 with Analytic-DDPM. For other cases, we find $y = 1$ works better.

We use the official implementation of FID to pytorch (`https://github.com/mseitzer/pytorch-fid`). We calculate the FID score on 50K generated samples on all datasets. Following Nichol & Dhariwal (2021), the reference distribution statistics are computed on the full training set for CIFAR10 and ImageNet 64x64. For CelebA 64x64 and LSUN Bedroom, the reference distribution statistics is computed on 50K training samples.

## F.3  CHOICE OF THE NUMBER OF MONTE CARLO SAMPLES AND CALCULATION OF $\Gamma$

We use a maximal $M$ without introducing too much computation. Specifically, we set $M = 50000$ on CIFAR10, $M = 10000$ on CelebA 64x64 and ImageNet 64x64 and $M = 1000$ on LSUN Bedroom by default without a sweep. All of the samples are from the training dataset. We use the default settings of $M$ for all results in Table 1, Table 2 and Table 3.

We only calculate $\Gamma$ in Eq. (8) **once** for a pretrained model, and $\Gamma$ is reused during inference under different settings (e.g., trajectories of smaller $K$) in Table 1, Table 2 and Table 3.

### F.4 IMPLEMENTATION OF THE EVEN TRAJECTORY

We follow Nichol & Dhariwal (2021) for the implementation of the even trajectory. Given the number of timesteps $K$ of a trajectory, we firstly determine the stride $a = \frac{N-1}{K-1}$. Then the $k$th timestep is determined as $\text{round}(1 + a(k - 1))$.

### F.5 EXPERIMENTAL DETAILS OF TABLE 3

In Table 3, the results of DDPM, DDIM and Analytic-DPM are based on the same score-based models (i.e., those listed in Section F.1). We get the results of Improved DDPM by running its official code and unconditional $L_{\text{hybrid}}$ models (`https://github.com/openai/improved-diffusion`). As shown in Table 4, on the same dataset, the sizes as well as the averaged time of a single function evaluation of these models are almost the same.

Table 4: Model size and the averaged time to run a model function evaluation with a batch size of 10 on one GeForce RTX 2080 Ti.

|  | CIFAR10 | CelebA 64x64 | LSUN Bedroom |
|---|---|---|---|
| DDPM, DDIM, Analytic-DPM | 200.44 MB / 29 ms | 300.22 MB / 50 ms | 433.63 MB / 438 ms |
| Improved DDPM | 200.45 MB / 30 ms | Missing model | 433.64 MB / 439 ms |

The DDPM and DDIM results on CIFAR10 are based on the quadratic trajectory following Song et al. (2020a), which gets better FID than the even trajectory. The Analytic-DPM result is based on the DDPM forward process on LSUN Bedroom, and based on the DDIM forward process on other datasets. These choices achieve better efficiency than their alternatives.

## G    ADDITIONAL RESULTS

### G.1    VISUALIZATION OF REVERSE VARIANCES AND VARIATIONAL BOUND TERMS

Figure 1 visualizes the reverse variances and $L_{\mathrm{vb}}$ terms on CIFAR10 with the linear forward noise schedule (LS). In Figure 2, we show more DDPM results on CIFAR10 with the cosine forward noise schedule (CS), CelebA 64x64 and ImageNet 64x64.

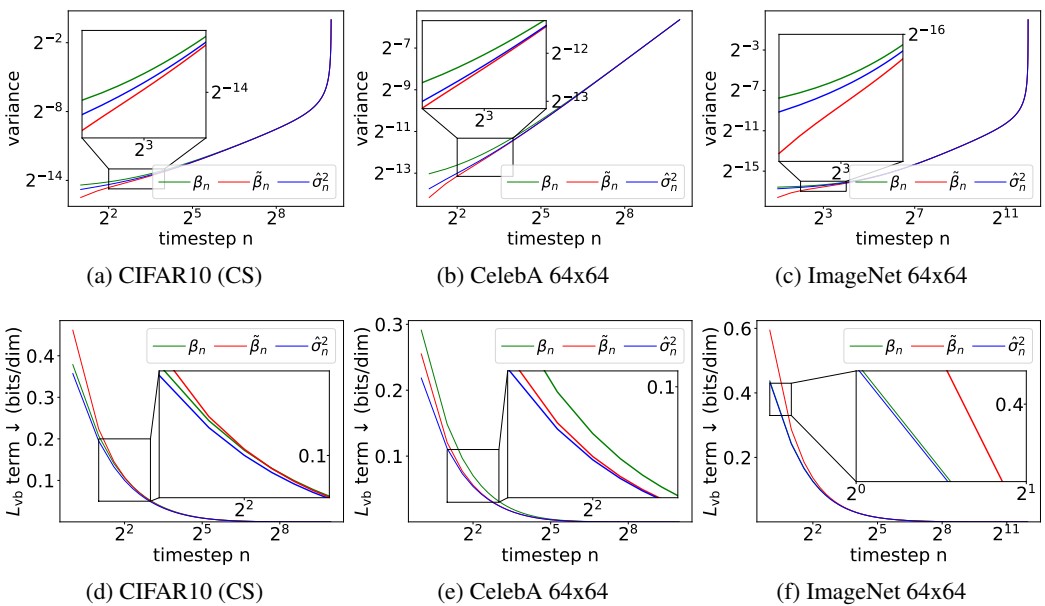

Figure 2: Comparing our analytic estimate $\hat{\sigma}_n^2$ and prior works with handcrafted variances $\beta_n$ and $\tilde{\beta}_n$. (a-c) compare the values of the variance of different timesteps. (d-e) compare the term in $L_{\mathrm{vb}}$ corresponding to each timestep. The value of $L_{\mathrm{vb}}$ is the area under the corresponding curve.

### G.2    ABLATION STUDY ON THE NUMBER OF MONTE CARLO SAMPLES

We show that only a small number of Monte Carlo (MC) samples $M$ in Eq. (8) is enough for a small MC variance. As shown in Figure 3, the values of $\Gamma_n$ with $M = 100$ and $M = 50000$ Monte Carlo samples are almost the same in a single trial. To explicitly see the variance, in Figure 4 and Figure 5, we plot the mean, the standard deviation and the relative standard deviation (RSD) (i.e., the ratio of the standard deviation to the mean) of a single Monte Carlo sample $\frac{||s_n(x_n)||^2}{d}$, $x_n \sim q_n(x_n)$ and $\Gamma_n$ with different $M$ respectively on CIFAR10 (LS). In all cases, the RSD decays fast as $n$ increases. When $n$ is small (e.g., $n = 1$), using $M = 10$ Monte Carlo samples can ensure that the RSD of $\Gamma_n$ is below 0.1, and using $M = 100$ Monte Carlo samples can ensure that the RSD of $\Gamma_n$ is about 0.025. When $n > 100$, the RSD of a single Monte Carlo sample is below 0.05, and using only $M = 1$ Monte Carlo sample can ensure the RSD of $\Gamma_n$ is below 0.05. Overall, a small $M$ like 10 and 100 is sufficient for a small Monte Carlo variance.

Furthermore, we show that Analytic-DPM with a small $M$ like 10 and 100 has a similar performance to that with a large $M$. As shown in Figure 6 (a), using $M = 100$ or $M = 50000$ almost does not affect the likelihood results on CIFAR10 (LS). In Table 5 (a), we show results with even smaller $M$ (e.g., $M = 1, 3, 10$). Under both the NLL and FID metrics, $M = 10$ achieves a similar result to that of $M = 50000$. The results are similar on ImageNet 64x64, as shown in Figure 6 (b) and Table 5 (b). Notably, the expected performance of FID is almost not influenced by the choice of $M$.

As a result, Analytic-DPM consistently improves the baselines using a much smaller $M$ (e.g., $M = 10$), as shown in Table 6.

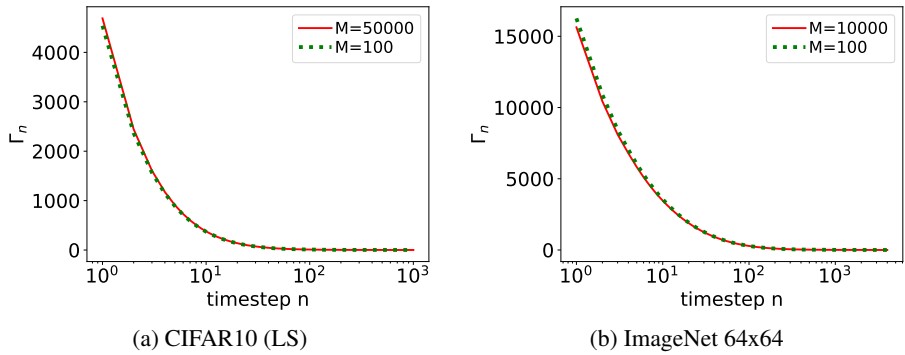

(a) CIFAR10 (LS)  (b) ImageNet 64x64

Figure 3: The value of $\Gamma_n$ in a single trial with different number of Monte Carlo samples $M$.

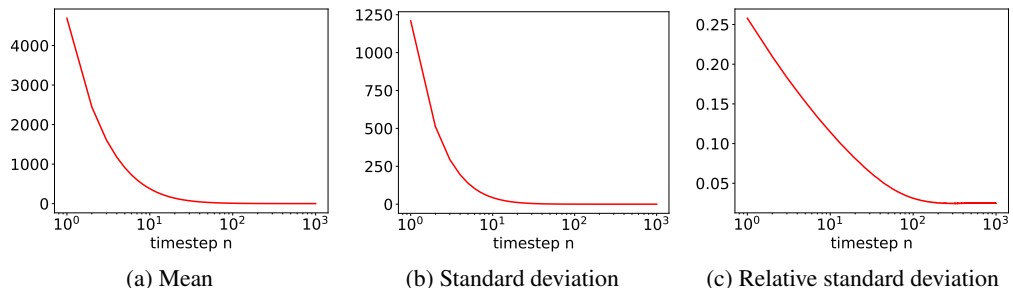

(a) Mean  (b) Standard deviation  (c) Relative standard deviation

Figure 4: The mean, the standard deviation and the relative standard deviation (RSD) (i.e., the ratio of the standard deviation to the mean) of a single Monte Carlo sample $\frac{||\boldsymbol{s}_n(\boldsymbol{x}_n)||^2}{d}$, $\boldsymbol{x}_n \sim q_n(\boldsymbol{x}_n)$ at different $n$ on CIFAR10 (LS). These values are estimated by 50000 samples.

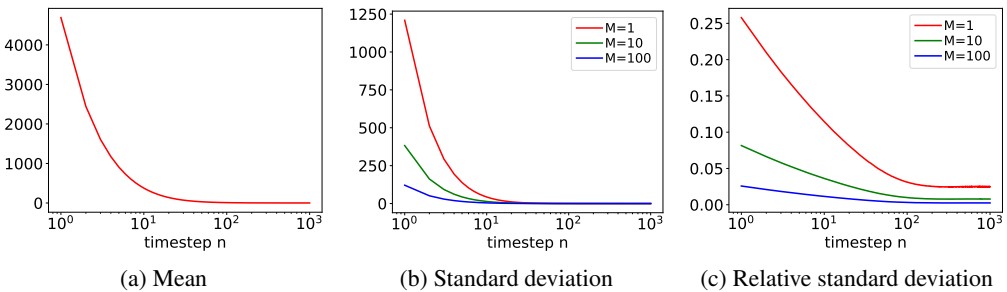

(a) Mean  (b) Standard deviation  (c) Relative standard deviation

Figure 5: The mean, the standard deviation and the relative standard deviation (RSD) (i.e., the ratio of the standard deviation to the mean) of $\Gamma_n$ with different number of Monte Carlo samples $M$ at different $n$ on CIFAR10 (LS). These values are directly calculated from the mean, the standard deviation and the RSD of $\frac{||\boldsymbol{s}_n(\boldsymbol{x}_n)||^2}{d}$, $\boldsymbol{x}_n \sim q_n(\boldsymbol{x}_n)$ presented in Figure 4.

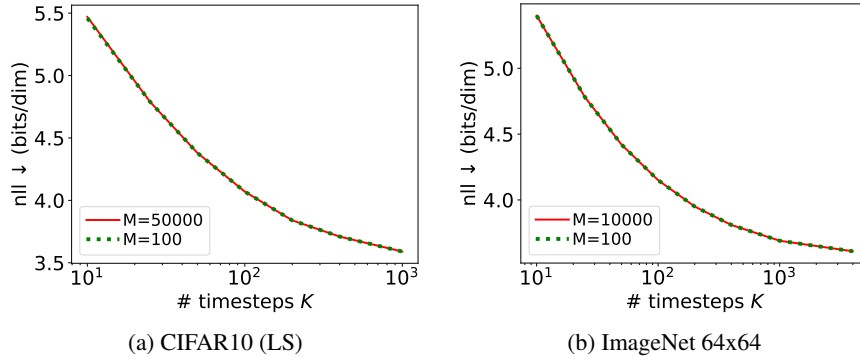

(a) CIFAR10 (LS)  (b) ImageNet 64x64

Figure 6: The curves of NLL v.s. the number of timesteps $K$ in a trajectory with different number of Monte Carlo samples $M$, evaluated under $\sigma_n^2 = \hat{\sigma}_n^2$ and the even trajectory.

Table 5: The negative log-likelihood (NLL) and the FID results of Analytic-DPM with different number of Monte Carlo samples $M$. The results with $M = 1, 3, 10, 100$ are averaged by 5 runs. All results are evaluated under the DDPM forward process and the even trajectory. We use $K = 10$ for CIFAR10 (LS) and $K = 25$ for ImageNet 64x64.

| (a) CIFAR10 (LS) | | | (b) ImageNet 64x64 | | |
|---|---|---|---|---|---|
| | NLL ↓ | FID ↓ | | NLL ↓ | FID ↓ |
| $M = 1$ | 6.220±1.126 | 34.05±4.97 | $M = 1$ | 4.943±0.162 | 31.59±5.11 |
| $M = 3$ | 5.689±0.424 | 34.29±2.88 | $M = 3$ | 4.821±0.055 | 31.98±1.19 |
| $M = 10$ | 5.469±0.005 | 33.69±2.10 | $M = 10$ | 4.791±0.017 | 31.93±1.02 |
| $M = 100$ | 5.468±0.004 | 34.63±0.68 | $M = 100$ | 4.785±0.003 | 31.93±0.69 |
| $M = 50000$ | 5.471 | 34.26 | $M = 10000$ | 4.783 | 32.56 |

Table 6: The NLL and FID comparison between Analytic-DDPM with $M = 10$ Monte Carlo samples and DDPM. Results are evaluated under the even trajectory on CIFAR10 (LS).

| # timesteps $K$ | 10 | 25 | 50 | 100 | 200 | 400 |
|---|---|---|---|---|---|---|
| NLL ↓ | | | | | | |
| Analytic-DDPM ($M = 10$) | 5.47 | 4.80 | 4.38 | 4.07 | 3.85 | 3.71 |
| DDPM | 6.99 | 6.11 | 5.44 | 4.86 | 4.39 | 4.07 |
| FID ↓ | | | | | | |
| Analytic-DDPM ($M = 10$) | 33.69 | 11.99 | 7.24 | 5.39 | 4.19 | 3.58 |
| DDPM | 44.45 | 21.83 | 15.21 | 10.94 | 8.23 | 4.86 |

## G.3   TIGHTNESS OF THE BOUNDS

In Section 3.1 and Appendix C, we derive upper and lower bounds of the optimal reverse variance. In this section, we show these bounds are tight numerically in practice. In Figure 7, we plot the combined upper bound (i.e., the minimum of the upper bounds in Eq. (11) and Eq. (12)) and the lower bound on CIFAR10. As shown in Figure 7 (a,c), the two bounds almost overlap under the full-timesteps ($K=N$) trajectory. When the trajectory has a smaller number of timesteps (e.g., $K=100$), the two bounds also overlap when the timestep $\tau_k$ is large. These results empirically validate that our bounds are tight, especially when the timestep is large.

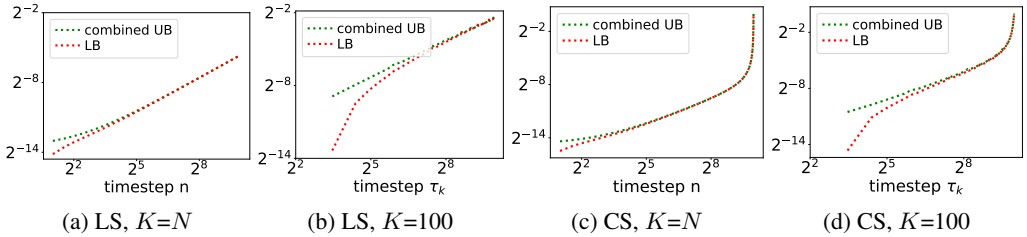

(a) LS, $K=N$      (b) LS, $K=100$      (c) CS, $K=N$      (d) CS, $K=100$

Figure 7: The combined upper bound (UB) and the lower bound (LB) under full-timesteps ($K=N$) and 100-timesteps ($K=100$) trajectories on CIFAR10 (LS) and CIFAR10 (CS).

In Figure 8, we also plot the two upper bounds in Eq. (11) and Eq. (12) individually. The upper bound in Eq. (11) is tighter when the timestep is small and the other one is tighter when the timestep is large. Thereby, both upper bounds contribute to the combined upper bound.

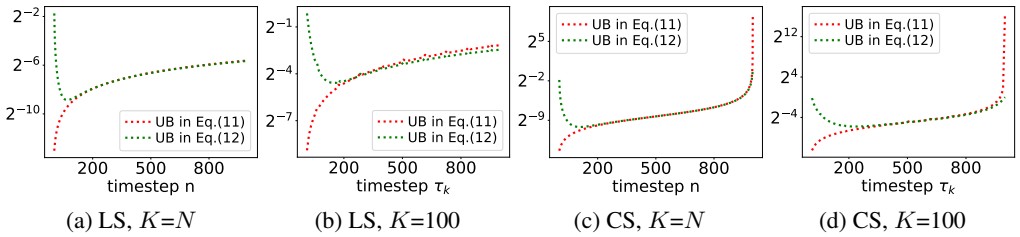

(a) LS, $K=N$      (b) LS, $K=100$      (c) CS, $K=N$      (d) CS, $K=100$

Figure 8: The upper bounds (UB) in Eq. (11) and Eq. (12) under full-timesteps ($K=N$) and 100-timesteps ($K=100$) trajectories on CIFAR10 (LS) and CIFAR10 (CS).

To see how these bounds work in practice, in Figure 9, we plot the probability that $\hat{\sigma}_n^2$ is clipped by the bounds in Theorem 2 with different number of Monte Carlo samples $M$ on CIFAR10 (LS). For all $M$, the curves of ratio v.s. $n$ are similar and the estimate is clipped more frequently when $n$ is large. This is as expected because when $n$ is large, the gap between the upper bound in Eq. (12) and the lower bound in Eq. (11) tends to zero. The results also agree with the plot of the bounds in Figure 7. Besides, the similarity of results between different $M$ implies that the clipping by bounds occurs mainly due to the error of the score-based model $s_n(x_n)$, instead of the randomness in Monte Carlo methods.

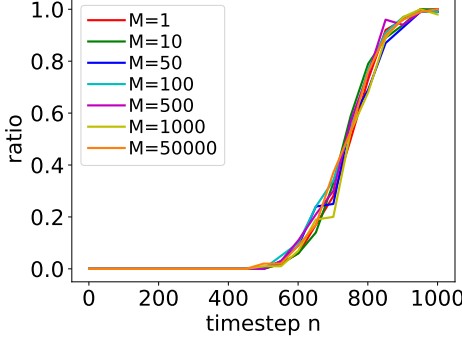

Figure 9: The probability that $\hat{\sigma}_n^2$ is clipped by the bounds in Theorem 2 with different number of Monte Carlo samples $M$ on CIFAR10 (LS). The probability is estimated by the ratio of $\hat{\sigma}_n^2$ being clipped in 100 independent trials. The results are evaluated with full timesteps $K = N$.

### G.4 ABLATION STUDY ON THE CLIPPING OF $\sigma_2$ DESIGNED FOR SAMPLING

This section validates the argument in Appendix F.2 that properly clipping the noise scale $\sigma_2$ in $p(\boldsymbol{x}_1|\boldsymbol{x}_2)$ leads to a better sample quality. As shown in Figure 10 and Figure 11, it greatly improves the sample quality of our analytic estimate. The curves of clipping and no clipping overlap as $K$ increases, since $\sigma_2$ is below the threshold for a large $K$.

Indeed, as shown in Table 7, the clipping threshold designed for sampling in Appendix F.2 is 1 to 3 orders of magnitude smaller than the combined upper bound in Theorem 2 (i.e., the minimum of the upper bounds in Eq. (11) and Eq. (12)) when $K$ is small.

As shown in Figure 12, clipping $\sigma_2$ also slightly improves the sample quality of the handcrafted reverse variance $\sigma_n^2 = \beta_n$ used in the original DDPM (Ho et al., 2020). As for the other two variances, i.e., $\sigma_n^2 = \tilde{\beta}_n$ in the original DDPM and $\sigma_n^2 = \lambda_n^2 = 0$ in the original DDIM (Song et al., 2020a), their $\sigma_2$ generally don't exceed the threshold and thereby clipping doesn't affect the result.

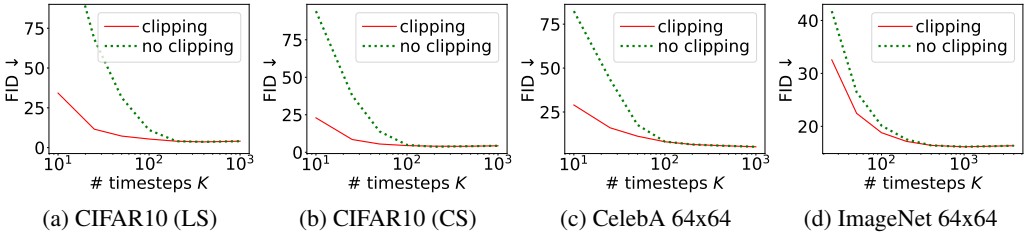

| (a) CIFAR10 (LS) | (b) CIFAR10 (CS) | (c) CelebA 64x64 | (d) ImageNet 64x64 |

Figure 10: Ablation study on clipping $\sigma_2$, evaluated under Analytic-DDPM.

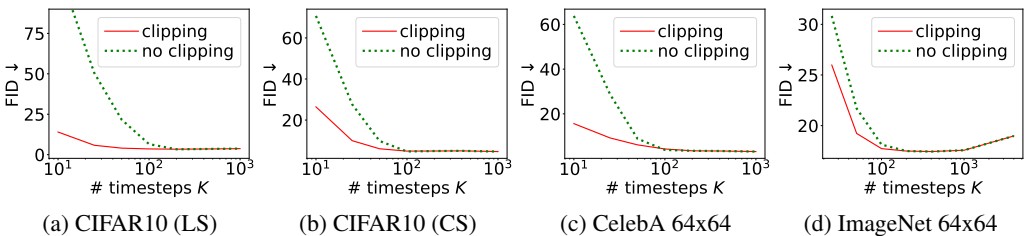

| (a) CIFAR10 (LS) | (b) CIFAR10 (CS) | (c) CelebA 64x64 | (d) ImageNet 64x64 |

Figure 11: Ablation study on clipping $\sigma_2$, evaluated under Analytic-DDIM.

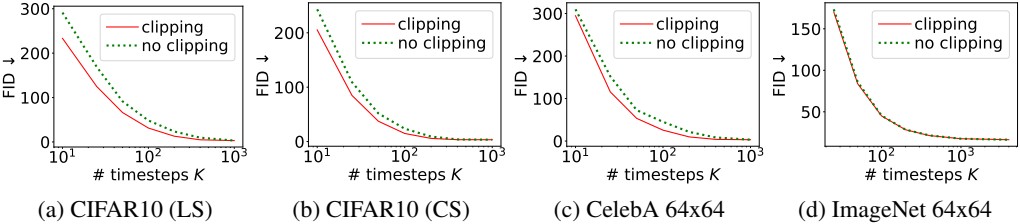

| (a) CIFAR10 (LS) | (b) CIFAR10 (CS) | (c) CelebA 64x64 | (d) ImageNet 64x64 |

Figure 12: Ablation study on clipping $\sigma_2$, evaluated under DDPM with $\sigma_n^2 = \beta_n$.

Table 7: Comparing the values of (i) the threshold in Appendix F.2 used to clip $\sigma_2^2$ designed for sampling, (ii) the combined upper bound in Theorem 2 when $n = 2$, (iii) the lower bound in Theorem 2 when $n = 2$ and (iv) our analytic estimate $\hat{\sigma}_2^2$. We show comparison results on different datasets and different forward processes when $K$ is small.

| Model \ # timesteps $K$ | | 10 | 25 | 50 | 100 |
|---|---|---|---|---|---|
| **CIFAR10 (LS)** | | | | | |
| DDPM | Threshold ($y$=2) | $3.87 \times 10^{-4}$ | $3.87 \times 10^{-4}$ | $3.87 \times 10^{-4}$ | $3.87 \times 10^{-4}$ |
| | Upper bound | $1.45 \times 10^{-1}$ | $2.24 \times 10^{-2}$ | $6.20 \times 10^{-3}$ | $2.10 \times 10^{-3}$ |
| | Lower bound | $9.99 \times 10^{-5}$ | $9.96 \times 10^{-5}$ | $9.84 \times 10^{-5}$ | $9.55 \times 10^{-5}$ |
| | $\hat{\sigma}_2^2$ | $8.70 \times 10^{-3}$ | $2.99 \times 10^{-3}$ | $1.32 \times 10^{-3}$ | $6.54 \times 10^{-4}$ |
| DDIM | Threshold ($y$=1) | $9.66 \times 10^{-5}$ | $9.66 \times 10^{-5}$ | $9.66 \times 10^{-5}$ | $9.66 \times 10^{-5}$ |
| | Upper bound | $1.37 \times 10^{-1}$ | $1.96 \times 10^{-2}$ | $4.82 \times 10^{-3}$ | $1.36 \times 10^{-3}$ |
| | Lower bound | $0$ | $0$ | $0$ | $0$ |
| | $\hat{\sigma}_2^2$ | $8.17 \times 10^{-3}$ | $2.54 \times 10^{-3}$ | $9.66 \times 10^{-4}$ | $3.73 \times 10^{-4}$ |
| **CIFAR10 (CS)** | | | | | |
| DDPM | Threshold ($y$=1) | $9.66 \times 10^{-5}$ | $9.66 \times 10^{-5}$ | $9.66 \times 10^{-5}$ | $9.66 \times 10^{-5}$ |
| | Upper bound | $3.56 \times 10^{-2}$ | $6.15 \times 10^{-3}$ | $1.85 \times 10^{-3}$ | $6.80 \times 10^{-4}$ |
| | Lower bound | $4.12 \times 10^{-5}$ | $4.10 \times 10^{-5}$ | $4.04 \times 10^{-5}$ | $3.89 \times 10^{-5}$ |
| | $\hat{\sigma}_2^2$ | $3.90 \times 10^{-3}$ | $1.28 \times 10^{-3}$ | $5.61 \times 10^{-4}$ | $2.75 \times 10^{-4}$ |
| DDIM | Threshold ($y$=1) | $9.66 \times 10^{-5}$ | $9.66 \times 10^{-5}$ | $9.66 \times 10^{-5}$ | $9.66 \times 10^{-5}$ |
| | Upper bound | $3.33 \times 10^{-2}$ | $5.22 \times 10^{-3}$ | $1.37 \times 10^{-3}$ | $4.18 \times 10^{-4}$ |
| | Lower bound | $0$ | $0$ | $0$ | $0$ |
| | $\hat{\sigma}_2^2$ | $3.61 \times 10^{-3}$ | $1.06 \times 10^{-3}$ | $3.95 \times 10^{-4}$ | $1.53 \times 10^{-4}$ |
| **CelebA 64x64** | | | | | |
| DDPM | Threshold ($y$=2) | $3.87 \times 10^{-4}$ | $3.87 \times 10^{-4}$ | $3.87 \times 10^{-4}$ | $3.87 \times 10^{-4}$ |
| | Upper bound | $1.45 \times 10^{-1}$ | $2.24 \times 10^{-2}$ | $6.20 \times 10^{-3}$ | $2.10 \times 10^{-3}$ |
| | Lower bound | $9.99 \times 10^{-5}$ | $9.96 \times 10^{-5}$ | $9.84 \times 10^{-5}$ | $9.55 \times 10^{-5}$ |
| | $\hat{\sigma}_2^2$ | $4.04 \times 10^{-3}$ | $1.54 \times 10^{-3}$ | $7.54 \times 10^{-4}$ | $4.06 \times 10^{-4}$ |
| DDIM | Threshold ($y$=1) | $9.66 \times 10^{-5}$ | $9.66 \times 10^{-5}$ | $9.66 \times 10^{-5}$ | $9.66 \times 10^{-5}$ |
| | Upper bound | $1.37 \times 10^{-1}$ | $1.96 \times 10^{-2}$ | $4.82 \times 10^{-3}$ | $1.36 \times 10^{-3}$ |
| | Lower bound | $0$ | $0$ | $0$ | $0$ |
| | $\hat{\sigma}_2^2$ | $3.74 \times 10^{-3}$ | $1.26 \times 10^{-3}$ | $5.17 \times 10^{-4}$ | $2.11 \times 10^{-4}$ |

| Model \ # timesteps $K$ | | 25 | 50 | 100 | 200 |
|---|---|---|---|---|---|
| **ImageNet 64x64** | | | | | |
| DDPM | Threshold ($y$=1) | $9.66 \times 10^{-5}$ | $9.66 \times 10^{-5}$ | $9.66 \times 10^{-5}$ | $9.66 \times 10^{-5}$ |
| | Upper bound | $5.93 \times 10^{-3}$ | $1.84 \times 10^{-3}$ | $6.44 \times 10^{-4}$ | $2.61 \times 10^{-4}$ |
| | Lower bound | $9.85 \times 10^{-6}$ | $9.81 \times 10^{-6}$ | $9.72 \times 10^{-6}$ | $9.51 \times 10^{-6}$ |
| | $\hat{\sigma}_2^2$ | $1.40 \times 10^{-3}$ | $6.05 \times 10^{-4}$ | $2.77 \times 10^{-4}$ | $1.39 \times 10^{-4}$ |
| DDIM | Threshold ($y$=1) | $9.66 \times 10^{-5}$ | $9.66 \times 10^{-5}$ | $9.66 \times 10^{-5}$ | $9.66 \times 10^{-5}$ |
| | Upper bound | $5.46 \times 10^{-3}$ | $1.59 \times 10^{-3}$ | $5.03 \times 10^{-4}$ | $1.77 \times 10^{-4}$ |
| | Lower bound | $0$ | $0$ | $0$ | $0$ |
| | $\hat{\sigma}_2^2$ | $1.28 \times 10^{-3}$ | $5.17 \times 10^{-4}$ | $2.12 \times 10^{-4}$ | $9.11 \times 10^{-5}$ |

### G.5 SAMPLE QUALITY COMPARISON BETWEEN DIFFERENT TRAJECTORIES

While the optimal trajectory (OT) significantly improves the likelihood results, it doesn't lead to better FID results. As shown in Figure 13, the even trajectory (ET) has better FID results. Such a behavior essentially roots in the different natures of the two metrics and has been investigated in extensive prior works (Ho et al., 2020; Nichol & Dhariwal, 2021; Song et al., 2021; Vahdat et al., 2021; Watson et al., 2021; Kingma et al., 2021).

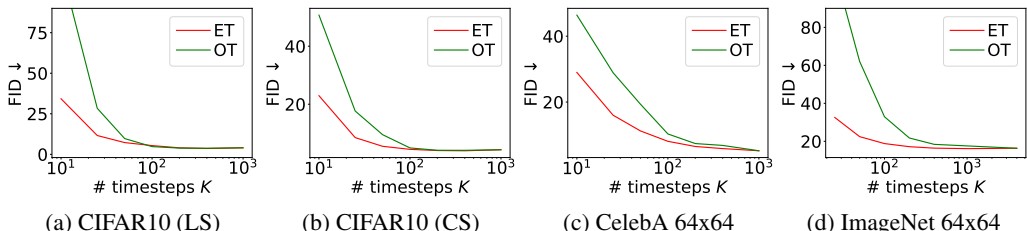

| (a) CIFAR10 (LS) | (b) CIFAR10 (CS) | (c) CelebA 64x64 | (d) ImageNet 64x64 |

Figure 13: FID results with ET and OT, evaluated under Analytic-DDPM.

### G.6 ADDITIONAL LIKELIHOOD COMPARISON

We compare our Analytic-DPM to Improved DDPM (Nichol & Dhariwal, 2021) that predicts the reverse variance by a neural network. The comparison is based on the ImageNet 64x64 model described in Appendix F.1. As shown in Table 8, with full timesteps, Analytic-DPM achieves a NLL of 3.61, which is very close to 3.57 achieved by predicting the reverse variance in Improved DDPM. Besides, we also notice that the ET reduces the log-likelihood performance of Improved DDPM when $K$ is small, and this is consistent with what Nichol & Dhariwal (2021) report. In contrast, our Analytic-DPM performs well with the ET.

Table 8: Negative log-likelihood (bits/dim) ↓ under the DDPM forward process on ImageNet 64x64. All are evaluated under the even trajectory (ET).

| Model \ # timesteps $K$ | 25 | 50 | 100 | 200 | 400 | 1000 | 4000 |
|---|---|---|---|---|---|---|---|
| Improved DDPM | 18.91 | 8.46 | 5.27 | 4.24 | 3.86 | **3.68** | **3.57** |
| Analytic-DDPM | **4.78** | **4.42** | **4.15** | **3.95** | **3.81** | 3.69 | 3.61 |

### G.7 CELEBA 64x64 RESULTS WITH A SLIGHTLY DIFFERENT IMPLEMENTATION OF THE EVEN TRAJECTORY

Song et al. (2020a) use a slightly different implementation of the even trajectory on CelebA 64x64. They choose a different stride $a = \text{int}(\frac{N}{K})$, and the $k$th timestep is determined as $1 + a(k - 1)$. As shown in Table 9, under the setting of Song et al. (2020a) on CelebA 64x64, our Analytic-DPM still improves the original DDIM consistently and improves the original DDPM in most cases.

### G.8 COMPARISON TO OTHER CLASSES OF GENERATIVE MODELS

While DPMs and their variants serve as the most direct baselines to validate the effectiveness of our method, we also compare with other classes of generative models in Table 10. Analytic-DPM achieves competitive sample quality results among various generative models, and meanwhile significantly reduces the efficiency gap between DPMs and other models.

Table 9: FID ↓ on CelebA 64x64, following the even trajectory implementation of Song et al. (2020a). [†]Original results in Song et al. (2020a). [‡]Our reproduced results.

| Model \ # timesteps $K$ | 10 | 20 | 50 | 100 | 1000 |
|---|---|---|---|---|---|
| CelebA 64x64 | | | | | |
| DDPM, $\sigma_n^2 = \tilde{\beta}_n$[†] | 33.12 | 26.03 | 18.48 | 13.93 | 5.98 |
| DDPM, $\sigma_n^2 = \tilde{\beta}_n$[‡] | 33.13 | 25.95 | 18.61 | 13.92 | 5.95 |
| DDPM, $\sigma_n^2 = \beta_n$[†] | 299.71 | 183.83 | 71.71 | 45.20 | **3.26** |
| DDPM, $\sigma_n^2 = \beta_n$[‡] | 299.88 | 185.21 | 71.86 | 45.15 | **3.21** |
| Analytic-DDPM | **25.88** | **17.40** | **10.98** | **7.95** | 5.21 |
| DDIM, $\sigma_n^2 = \lambda_n^2 = 0$[†] | 17.33 | 13.73 | 9.17 | 6.53 | 3.51 |
| DDIM, $\sigma_n^2 = \lambda_n^2 = 0$[‡] | 17.38 | 13.72 | 9.17 | 6.51 | 3.40 |
| Analytic-DDIM | **12.74** | **9.50** | **5.96** | **4.14** | **3.13** |

Table 10: Comparison to other classes of generative models on CIFAR10. We show the FID results, the number of model function evaluations (NFE) to generate a single sample and the time to generate 10 samples with a batch size of 10 on one GeForce RTX 2080 Ti.

| Method | FID↓ | NFE ↓ | Time (s) ↓ |
|---|---|---|---|
| Analytic-DPM, $K = 25$ (ours) | 5.81 | 25 | 0.73 |
| DDPM, $K = 90$ (Ho et al., 2020) | 6.12 | 90 | 2.64 |
| DDIM, $K = 30$ (Song et al., 2020a) | 5.85 | 30 | 0.88 |
| Improved DDPM, $K = 45$ (Nichol & Dhariwal, 2021) | 5.96 | 45 | 1.37 |
| SNGAN (Miyato et al., 2018) | 21.7 | 1 | - |
| BigGAN (cond.) (Brock et al., 2018) | 14.73 | 1 | - |
| StyleGAN2 (Karras et al., 2020a) | 8.32 | 1 | - |
| StyleGAN2 + ADA (Karras et al., 2020a) | 2.92 | 1 | - |
| NVAE (Vahdat & Kautz, 2020) | 23.5 | 1 | - |
| Glow (Kingma & Dhariwal, 2018) | 48.9 | 1 | - |
| EBM (Du & Mordatch, 2019) | 38.2 | 60 | - |
| VAEBM (Xiao et al., 2020) | 12.2 | 16 | - |

### G.9 SAMPLES

In Figure 14-17, we show Analytic-DDIM constrained on a short trajectory of $K = 50$ timesteps can generate samples comparable to these under the best FID setting.

In Figure 18-21, we also show samples of both Analytic-DDPM and Analytic-DDIM constrained on trajectories of different number of timesteps $K$.

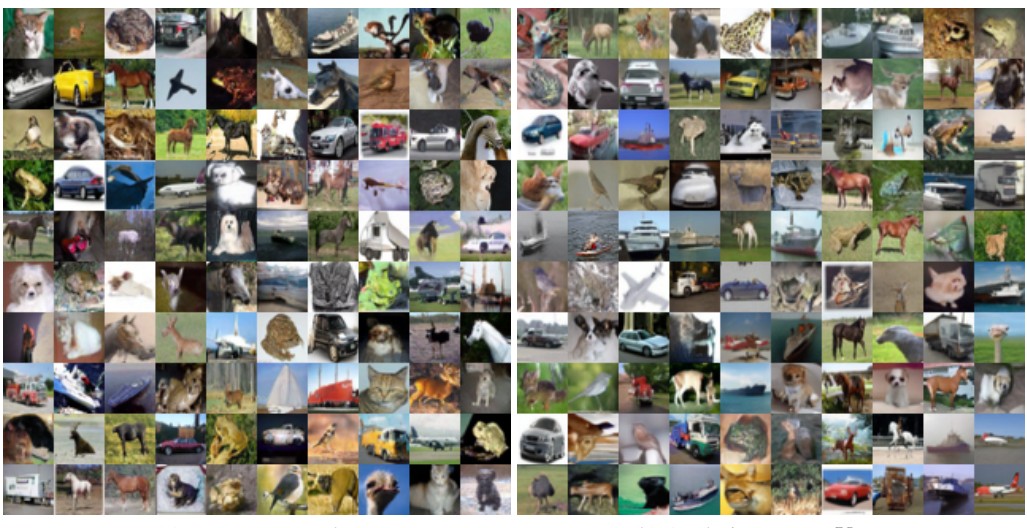

(a) Best FID samples           (b) Analytic-DDIM, $K = 50$

Figure 14: Generated samples on CIFAR10 (LS).

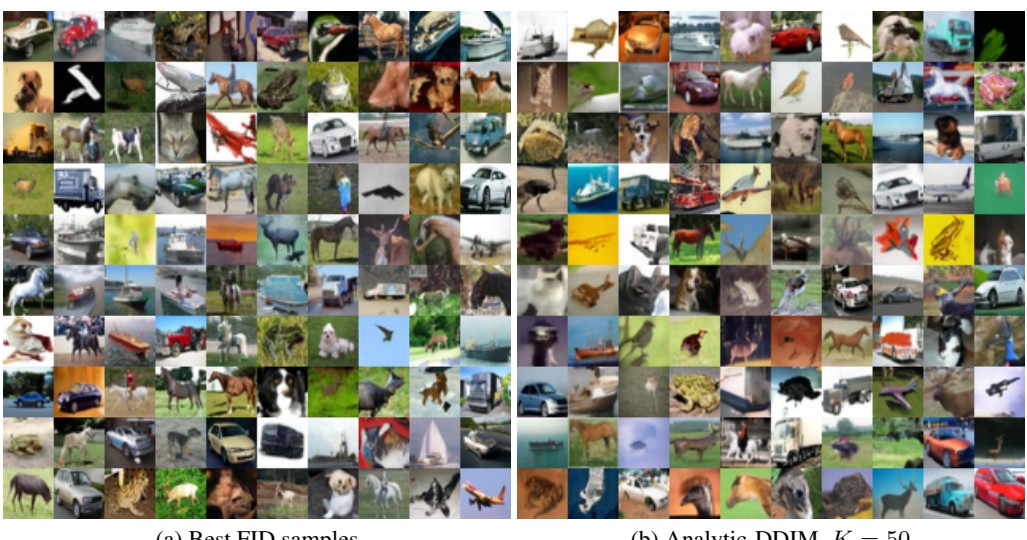

(a) Best FID samples           (b) Analytic-DDIM, $K = 50$

Figure 15: Generated samples on CIFAR10 (CS).

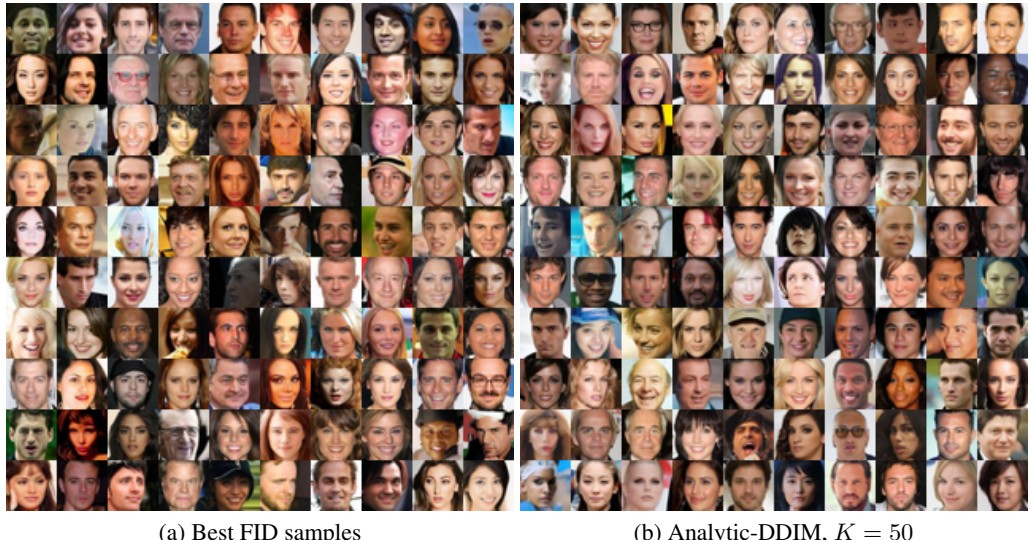

(a) Best FID samples                 (b) Analytic-DDIM, $K = 50$

Figure 16: Generated samples on CelebA 64x64.

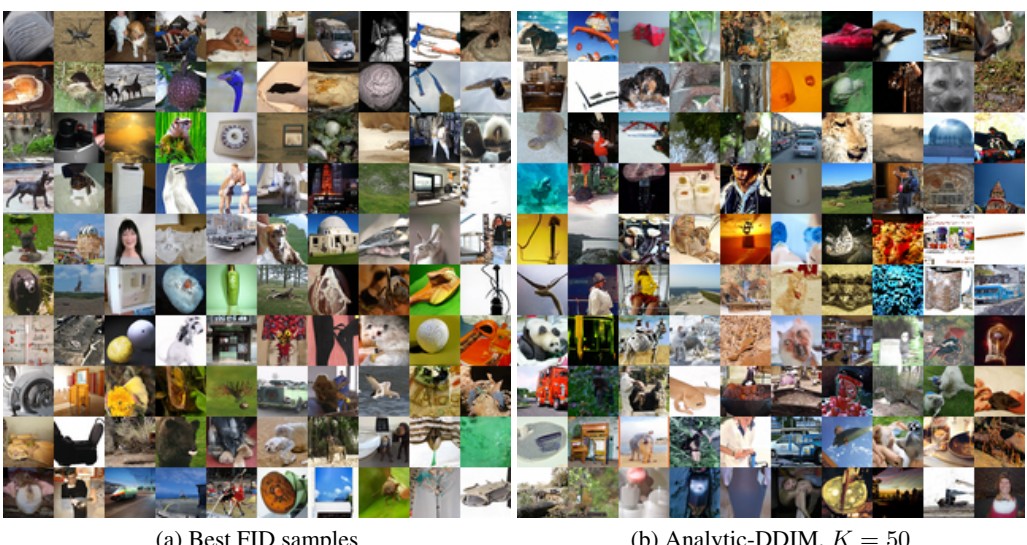

(a) Best FID samples                 (b) Analytic-DDIM, $K = 50$

Figure 17: Generated samples on ImageNet 64x64.

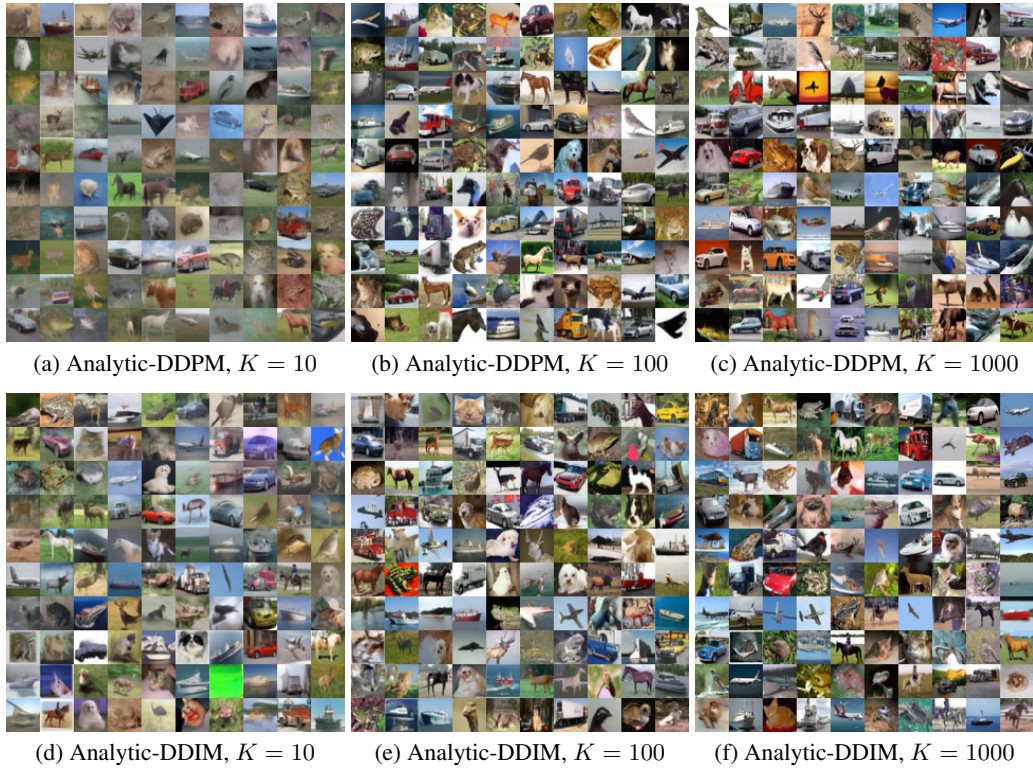

(a) Analytic-DDPM, $K = 10$     (b) Analytic-DDPM, $K = 100$     (c) Analytic-DDPM, $K = 1000$

(d) Analytic-DDIM, $K = 10$     (e) Analytic-DDIM, $K = 100$     (f) Analytic-DDIM, $K = 1000$

Figure 18: Generated samples on CIFAR10 (LS).

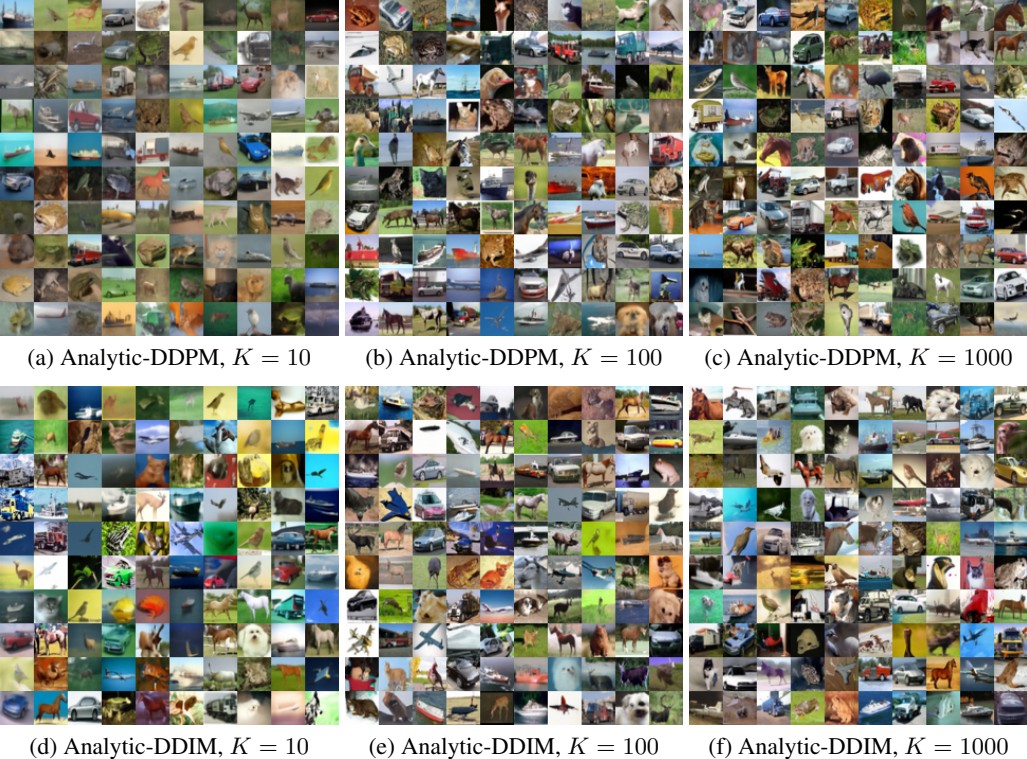

(a) Analytic-DDPM, $K = 10$     (b) Analytic-DDPM, $K = 100$     (c) Analytic-DDPM, $K = 1000$

(d) Analytic-DDIM, $K = 10$     (e) Analytic-DDIM, $K = 100$     (f) Analytic-DDIM, $K = 1000$

Figure 19: Generated samples on CIFAR10 (CS).

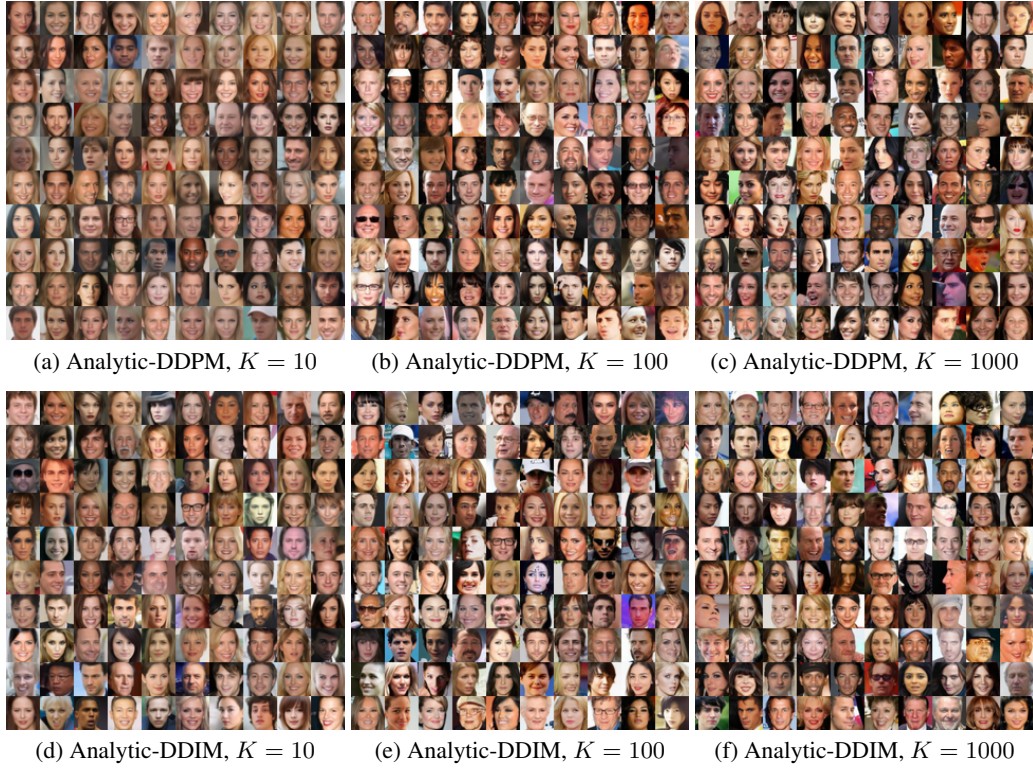

(a) Analytic-DDPM, $K = 10$    (b) Analytic-DDPM, $K = 100$    (c) Analytic-DDPM, $K = 1000$

(d) Analytic-DDIM, $K = 10$    (e) Analytic-DDIM, $K = 100$    (f) Analytic-DDIM, $K = 1000$

Figure 20: Generated samples on CelebA 64x64.

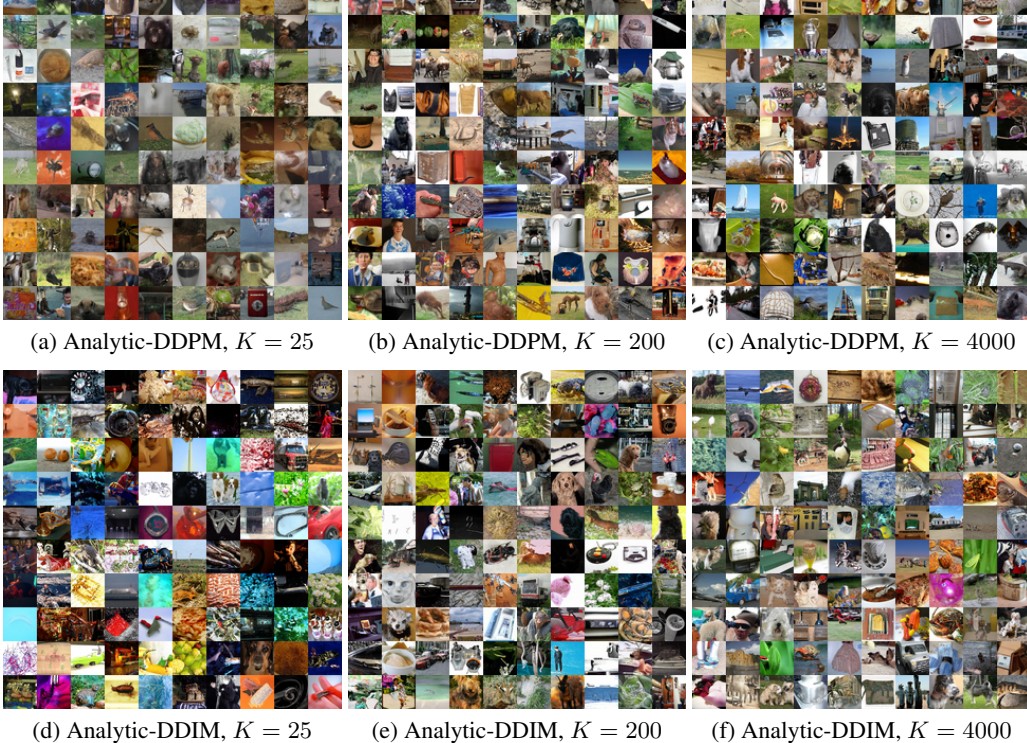

(a) Analytic-DDPM, $K = 25$    (b) Analytic-DDPM, $K = 200$    (c) Analytic-DDPM, $K = 4000$

(d) Analytic-DDIM, $K = 25$    (e) Analytic-DDIM, $K = 200$    (f) Analytic-DDIM, $K = 4000$

Figure 21: Generated samples on ImageNet 64x64.

# H   ADDITIONAL DISCUSSION

## H.1   THE EXTRA COST OF THE MONTE CARLO ESTIMATE

The extra cost of the Monte Carlo estimate $\Gamma$ is small compared to the whole inference cost. In fact, the Monte Carlo estimate requires $MN$ additional model function evaluations. During inference, suppose we generate $M_1$ samples or calculate the log-likelihood of $M_1$ samples with $K$ timesteps. Both DPMs and Analytic-DPMs need $M_1 K$ model function evaluations. Employing the same score-based models, the relative additional cost of Analytic-DPM is $\frac{MN}{M_1 K}$. As shown in Appendix G.2, a very small $M$ (e.g., $M = 10, 100$) is sufficient for Analytic-DPM, making the relative additional cost small if not negligible. For instance, on CIFAR10, let $M = 10$, $N = 1000$, $M_1 = 50000$ and $K \geq 10$, we obtain $\frac{MN}{M_1 K} \leq 0.02$ and Analytic-DPM still consistently improves the baselines as presented in Table 6.

Further, the additional calculation of the Monte Carlo estimate occurs only **once** given a pretrained model and training dataset, since we can save the results of $\Gamma = (\Gamma_1, \cdots, \Gamma_N)$ in Eq.(8) and reuse it among different inference settings (e.g., trajectories of various $K$). The reuse is valid, because the marginal distribution of a shorter forward process $q(\boldsymbol{x}_0, \boldsymbol{x}_{\tau_1}, \cdots, \boldsymbol{x}_{\tau_K})$ at timestep $\tau_k$ is the same as that of the full-timesteps forward process $q(\boldsymbol{x}_{0:N})$ at timestep $n = \tau_k$. Indeed, in our experiments (e.g., Table 1,2), $\Gamma$ is shared across different selections of $K$, trajectories and forward processes. Moreover, in practice, $\Gamma$ can be calculated offline and deployed together with the pretrained model and the online inference cost of Analytic-DPM is exactly the same as DPM.

## H.2   THE STOCHASTICITY OF THE VARIATIONAL BOUND AFTER PLUGGING THE ANALYTIC ESTIMATE

In this part, we write $L_{\mathrm{vb}}$ as $L_{\mathrm{vb}}(\sigma_n^2)$ to emphasize its dependence on the reverse variance $\sigma_n^2$.

When calculating the variational bound $L_{\mathrm{vb}}(\sigma_n^2)$ (i.e., the negative ELBO) of Analytic-DPM, we will plug $\hat{\sigma}_n^2$ into the variational bound and get $L_{\mathrm{vb}}(\hat{\sigma}_n^2)$. Since $\hat{\sigma}_n^2$ is calculated by the Monte Carlo method, $L_{\mathrm{vb}}(\hat{\sigma}_n^2)$ is a stochastic variable. A natural question is that whether $L_{\mathrm{vb}}(\hat{\sigma}_n^2)$ is a stochastic bound of $L_{\mathrm{vb}}(\mathbb{E}[\hat{\sigma}_n^2])$, which can be judged by the Jensen's inequality if $L_{\mathrm{vb}}$ is convex or concave. However, this is generally not guaranteed, as stated in Proposition 2.

**Proposition 2.** $L_{\mathrm{vb}}(\sigma_n^2)$ *is neither convex nor concave w.r.t.* $\sigma_n^2$.

*Proof.* Since $\sigma_n^2$ only influences the $n$-th term $L_n$ in the variational bound $L_{\mathrm{vb}}$, where

$$L_n = \begin{cases} \mathbb{E}_q D_{\mathrm{KL}}(q(\boldsymbol{x}_{n-1}|\boldsymbol{x}_n, \boldsymbol{x}_0)||p(\boldsymbol{x}_{n-1}|\boldsymbol{x}_n)) & 2 \leq n \leq N \\ -\mathbb{E}_q \log p(\boldsymbol{x}_0|\boldsymbol{x}_1) & n = 1 \end{cases},$$

we only need to study the convexity of $L_n$ w.r.t. $\sigma_n^2$.

When $2 \leq n \leq N$,

$$L_n = \frac{d}{2}\left( \frac{\lambda_n^2}{\sigma_n^2} - 1 + \log \frac{\sigma_n^2}{\lambda_n^2} + \frac{1}{\sigma_n^2} \mathbb{E}_q \frac{||\tilde{\boldsymbol{\mu}}(\boldsymbol{x}_n, \boldsymbol{x}_0) - \boldsymbol{\mu}_n(\boldsymbol{x}_n)||^2}{d} \right).$$

Let $A = \lambda_n^2 + \mathbb{E}_q \frac{||\tilde{\boldsymbol{\mu}}(\boldsymbol{x}_n, \boldsymbol{x}_0) - \boldsymbol{\mu}_n(\boldsymbol{x}_n)||^2}{d}$, then $L_n$ as a function of $\sigma_n^2$ is convex when $0 < \sigma_n^2 < 2A$ and concave when $2A < \sigma_n^2$. Thereby, $L_{\mathrm{vb}}(\sigma_n^2)$ is neither convex nor concave w.r.t. $\sigma_n^2$. $\qquad\square$

Nevertheless, in this paper, $L_{\mathrm{vb}}(\hat{\sigma}_n^2)$ is a stochastic upper bound of $L_{\mathrm{vb}}(\sigma_n^{*2})$ because $L_{\mathrm{vb}}(\sigma_n^{*2})$ is the optimal. The bias of $L_{\mathrm{vb}}(\hat{\sigma}_n^2)$ w.r.t. $L_{\mathrm{vb}}(\sigma_n^{*2})$ is due to the Monte Carlo method as well as the error of the score-based model. The former can be reduced by increasing the number of Monte Carlo samples. The latter is irreducible if the pretrained model is fixed, which motivates us to clip the estimate, as discussed in Section 3.1.

## H.3   COMPARISON TO OTHER GAUSSIAN MODELS AND THEIR RESULTS

The reverse process of DPMs is a Markov process with Gaussian transitions. Thereby, it is interesting to compare it with other Gaussian models, e.g., the expectation propagation (EP) with the Gaussian process (GP) (Kim & Ghahramani, 2006).

Both EP and Analytic-DPM use moment matching as a key step to find analytic solutions of $D_{\mathrm{KL}}(p_{target}||p_{opt})$ terms. However, to our knowledge, the relation between moment matching and DPMs has not been revealed in prior literature. Further, compared to EP, we emphasize that it is highly nontrivial to calculate the second moment of $p_{target}$ in DPMs because $p_{target}$ involves an unknown and potentially complicated data distribution.

In EP with GP (Kim & Ghahramani, 2006), $p_{target}$ is the product of a single likelihood factor and all other approximate factors for tractability. In fact, the form of the likelihood factor is chosen such that the first two moments of $p_{target}$ can be easily computed or approximated. For instance, the original EP (Minka, 2001) considers Gaussian mixture likelihood (or Bernoulli likelihood for classification) and the moments can be directed obtained by the properties of Gaussian (or integration by parts). Besides, at the cost of the tractability, there is no converge guarantee of EP in general.

In contrast, $p_{target}$ in this paper is the conditional distribution $q(\boldsymbol{x}_{n-1}|\boldsymbol{x}_n)$ of the corresponding joint distribution $q(\boldsymbol{x}_{0:N})$ defined by the forward process. Note that the moments of $q(\boldsymbol{x}_{n-1}|\boldsymbol{x}_n)$ are nontrivial to calculate because it involves an unknown and potentially complicated data distribution. Technically, in Lemma 13, we carefully use the law of total variance conditioned on $\boldsymbol{x}_0$ and convert the second moment of $q(\boldsymbol{x}_{n-1}|\boldsymbol{x}_n)$ to that of $q(\boldsymbol{x}_0|\boldsymbol{x}_n)$, which surprisingly can be expressed as the score function as proven in Lemma 11.

## H.4 FUTURE WORKS

In our work, we mainly focus on image data. It would be interesting to apply Analytic-DPM to other data modalities, e.g. speech data (Chen et al., 2020). As presented in Appendix E, our method can be applied to continuous DPMs, e.g., variational diffusion models (Kingma et al., 2021) that learn the forward noise schedule. It is appealing to see how Analytic-DPM works on these continuous DPMs. Finally, it is also interesting to incorporate the optimal reverse variance in the training process of DPMs.

