# OpenReview forum: "Analytic-DPM: an Analytic Estimate of the Optimal Reverse Variance in Diffusion Probabilistic Models"
_ICLR.cc/2022/Conference — ICLR 2022 Oral_

### Official Review · Reviewer_ejLx · 2021-10-26

**Correctness:** 4
**Technical Novelty And Significance:** 4
**Empirical Novelty And Significance:** 3
**Recommendation:** 8
**Confidence:** 4

**Main Review:**

Strengths:
- Paper has single well-executed idea (optimal variance can be computed as a function of score)
- Improvements over generally fair base lines are achieved by only post-processing; no additional training needed
- Paper is generally well-written and straightforward to read
- I did not check all Lemmas (appendix) in detail, however, it seems that their proofs are generally very rigorous and detailed.
- The bounds in Theorem 2 are quite nice. Given that DPMs are often used for images, the specific bound for the data distributions supported in the $d$-dimensional cube are quite applicable.

Weaknesses:
- For $K<100$, FID scores of the proposed method greatly rely on a trick of clipping the variance of the step $n=2$ (can be seen in appendix G.4). This seems to be a crucial element and should be discussed more in the main paper. In particular, I would like to see how this clipping compares to the bounds (and the clipping) of the optimal score wrt Theorem 2.
- The post-processing method can be quite expensive. For example, on ImageNet the best values are achieved using $MK= 400000$ additional ($M=100, K=N=4000$) function evaluations. For even trajectories (ET), the obvious solution would be to simply use $K \ll N$ (results for this are shown in paper), however, if I am not mistaken, for the optimal trajectory (OT), computed using the DP algorithm from [4], $MN$ functions have to be evaluated for any $K$ (see appendix B and eq (14)). Therefore, in my opinion, the comparison of DDPM and Analytic=DDPM in the OT setting of Table 1 is unfair.

Suggestions:
- It would be helpful to understand how often the estimator is actually clipped. I suggest to compute $R$ (maybe $R=100$) estimators for, say $M=[1,10, 50, 100, 500, 1000]$, and plot he number of the ratio of estimators that was clipped over $M$. This could be done for a few different instantiations of $n$ (I guess there will be more clipping for $n$ being small).
- the estimator $J$ in (14) is biased even when the correct score is known ($J$ is the log of an unbiased (Monte Carlo) estimator); it might be nice to mention this fact.
- It would be nice to see even lower $M$ in the ablations in G.2. Instead of only indicating variance by plotting the estimator in Figure 3, it would be nice to see the estimated variance of the sampler directly.
- Please state the number of Monte Carlo samples used for results in Tables 1,2,3.
- To me it seems that the optimal variance could also be used for training, using for example only $M=1$. I would be curious to see how this performs compared to using the optimal variance only as a guide for post-processing. I greatly encourage the authors to try this (in case there are no major problems I am missing)

**Summary Of The Paper:**

In Sec. 2, the authors review the general framework of Diffusion Probabilistic Models~(DPMs) with non-Markovian forward processes from [1] which has DDPMs [2] and DDIMs [1] as special cases. They emphasize that the variance of the reverse process, such as in DDPM and DDIM, are generally hand-crafted.

In Sec. 3, the authors present their main theorem: the optimal variance of the reverse process at step $n$ is a linear function of the expected squared norm of the score at step $n$. As a post-processing scheme of learned models, they then propose to replace the commonly used hand-crafted variances by an estimator of their optimal variance: in particular, the score is replaced by the learned score model and the expectation is replaced by a Monte Carlo estimate (with $M$ samples) thereof. In the remainder of this section, the authors derive bounds for the optimal variance and state that in practice their estimator is clipped according to these bounds.

In Sec. 4, the authors repeat the derivation of an optimal variance and bounds thereof when only a subset (of size $K$) of the $N$ timesteps are used for inference as is very common in the literature [1, 3, 4]. Furthermore, they show how their estimator in this setting can be used to still compute the optimal subset according to the commonly used DP algorithm introduced in [4].

In Sec. 6, they apply their post-processing scheme to models trained on CIFAR-10, CelebA 64 and ImageNet 64 and show that it generally leads to improved likelihood and FID scores. They show that their post-procecssing scheme consistently outperforms suboptimal hand-crafted choices (such as in DDPM and DDIM).

References:

[1] Song et al. Denoising Diffusion Implicit Models. ICLR 2021.

[2] Ho et al. Denoising Diffusion Probabilistic Models. NeurIPS 2020.

[3] Dhariwal & Nichol. Diffusion Models Beat GANs on Image Synthesis. arXiv:2105.05233.

[4] Watson et al. Learning to Efficiently Sample from Diffusion Probabilistic Models. arXiv:2106.03802.

**Summary Of The Review:**

Overall, I vote for accepting. The paper provides an important insight in DPMs and shows improved results for pretrained models by a simple post-processing technique. My major concern about the paper is that OT [4] does not work well in combination with their method, which in my opinion makes the work slightly less significant. I hope the authors can address this concern in the rebuttal period.

**Post discussion period update:**
I strongly vote for accepting (and also changed the correctness score from 3 to 4). All of my concerns, questions, and suggestions have been addressed by the authors in the discussion period. I thank the authors for this productive reviewing process.

---

> ### Author Response · Authors · 2021-11-15
> **Response to reviewer ejLx**
>
> We thank reviewer ejLx for the acknowledgement of our contributions and the valuable comments.
>
> ## Q1: Compare the clipping at $n=2$ to bounds in Theorem 2
>
> Thanks for the suggestion. We compare them as suggested and the former is 1 to 3 orders of magnitude smaller than the latter, when $K$ is small (e.g., $K<100$). Please see Table 7 of Appendix G.4 in the updated revision for details. We have added more discussion to the main paper (see Section 6) in the updated version.
>
>
>
>
> ## Q2: The post-processing method can be quite expensive and the fairness under the OT setting
> Thanks for the valuable comment.
>
> * Suppose we perform inference on $M_1$ samples. The relative additional cost of Analytic-DPM is $\frac{M N}{M_1 K}$ (please see details in our response to common concern 2). In our experiments, we found that a smaller $M$ (e.g., $M=10$) is sufficient for Analytic-DPM (please see our response to common concern 1 (1.2)), making the relative additional cost small if not negligible. For instance, on CIFAR10, let $M=10$, $N=1000$, $M_1=50000$ and $K\ge 10$, we obtain $\frac{M N}{M_1 K}\le 0.02$ and Analytic-DPM still consistently improves the baselines as presented in Table 2*.
>
> * Further, the additional calculation of the Monte Carlo estimate $\Gamma$ occurs only **once** given a pretrained model and training dataset, since we can save the results of $\Gamma = (\Gamma_1, \cdots, \Gamma_N)$ in Eq.(8) and reuse it among different inference settings (e.g., trajectories of various $K$). The reuse is valid, because the marginal distribution of a shorter forward process $q(x_0, x_{\tau_1}, \cdots, x_{\tau_K})$ at timestep $\tau_k$ is the same as that of the full-timesteps forward process $q(x_{0:N})$ at timestep $n=\tau_k$. Indeed, in our experiments (e.g., Table 1,2), $\Gamma$ is shared across different selections of $K$, trajectories and forward processes.
>
> * Finally, the DDPM with OT [4] also requires a post-processing of $M N$ additional function evaluations (see Section 4.3 in [4]) before getting the optimal trajectory. Thereby, under the OT setting, DDPM and Analytic-DDPM have the same function evaluations, and the comparison is fair.
>
> Please see more details in the common concern 2.
>
> [4] Watson et al. Learning to Efficiently Sample from Diffusion Probabilistic Models. arXiv:2106.03802.
>
> ## Q3: How often the estimate is actually clipped
>
> Thanks for the valuable suggestion. We have plotted the ratio of estimates that were clipped over different $M$ and $n$ in Figure 9 (Appendix G.3 in the revised version). For all $M$, the curves of the ratio v.s. $n$ are similar, and the estimate is clipped more frequently when $n$ is large. This is as expected because when $n$ is large, the gap between the upper bound in Eq. (12) and the lower bound in Eq. (11) tends to zero. The results also agree with the plot of the bounds in Figure 5 (Appendix G.3 in the original paper).
>
>
> ## Q4: The estimate is biased even when the correct score is known
>
> Thanks for the suggestion. We have explicitly mentioned it in Section 4.
>
>
> ## Q5: Lower $M$ and plotting variance
>
> Thanks for the suggestion. We added the results of Analytic-DPM over a wide range of $\{1,3,10,100,10000,50000\}$.
> Under both the NLL and FID metrics, $M=10$ achieves similar results to that of $M=50000$. The results are presented in Table 1*. Please see details in our response to common concern 1 (1.2).
>
> Besides, we plotted the standard deviations of the estimate when $M=1,10,100$ and the results agree with those in Figure 3 in the original paper. We have added the new results to Appendix G.2 in Figure 4\&5 in the revised version. See details in our response to common concern 1 (1.3).
>
> ## Q6: The number of Monte Carlo samples used for results in Tables 1,2,3
>
> Thanks for the suggestion. We set $M=50000$ on CIFAR10, $M=10000$ on CelebA 64x64 and ImageNet 64x64 and $M=1000$ on LSUN Bedroom by default without a sweep. See details in our response to common concern 1 (1.1).
>
> ## Q7: The optimal variance could also be used for training
>
> Thanks for the insightful suggestion. It is a promising future work and we have added a discussion in Section H.4.

---

> > ### Comment · Reviewer_ejLx · 2021-11-18
> > **Response to the authors**
> >
> > I thank the authors for clarifying most of my questions as well as incorporating my suggestions in their paper.
> >
> > I only want to follow up on one point: the authors say that "[...] the DDPM with OT [4] also requires a post-processing of $MN$ additional function evaluations (see Section 4.3 in [4]) before getting the optimal trajectory. Thereby, under the OT setting, DDPM and Analytic-DDPM have the same function evaluations, and the comparison is fair."
> >
> > I don't understand why a plain DDPM would need $MN$ function evaluations. The $M$ refers to the number of Monte Carlo samples to compute $\Gamma_n$ (which is specific to Analytic-DDPM) so it would not apply to plain DDPM? Wouldn't DDPM only need $N$ function evaluations? Could the authors clarify?
> >
> > [4] Watson et al. Learning to Efficiently Sample from Diffusion Probabilistic Models. arXiv:2106.03802.

---

> > > ### Author Response · Authors · 2021-11-19
> > > **Thanks for the valuable question**
> > >
> > > Thanks for the valuable question.
> > >
> > > Indeed, in order to get the optimal trajectory for DDPM, we need to calculate every term $L(t, s)$ (see Eq.(17) in [4]) appeared in the variational bound. $L(t, s)$ has an expectation term and this term is estimated by $M$ Monte Carlo samples. Empirically, the $M$ samples are drawn from the training dataset. Here we also use $M$ to denote the number of Monte Carlo samples as Analytic-DPM.
> > >
> > >
> > > Formally, $L(t, s)=\mathbb{E}_q KL(q(x_s|x_t,x_0)||p(x_s|x_t))$ when $s>0$, which can be written as
> > >
> > > $$
> > > L(t, s)=\frac{d}{2} [  C_{t,s} + A_{t,s} \mathbb{E}_q || x_0 - \frac{1}{\sqrt{\bar{\alpha}}_t}(x_t + \bar\beta_t s_t (x_t) ) ||^2 / d ],
> > > $$
> > >
> > >
> > > where $C_{t,s},A_{t,s}$ are only related to the forward variance $\beta_1,\cdots,\beta_N$ and  $\mathbb{E}_q || x_0 - \frac{1}{\sqrt{\bar\alpha_t}}(x_t + \bar\beta_t s_t (x_t) ) ||^2 / d$ is the expectation term to estimate. Similarly, the expectation term also appears in $L(t, s)$ when $s=0$.
> > >
> > > The expectation term is estimated using $M$ Monte Carlo samples:
> > >
> > > $$
> > > \Phi_t = \frac{1}{M} \sum_{m=1}^M || x_{0,m} - \frac{1}{\sqrt{\bar\alpha_t}}(x_{t,m} + \bar\beta_t s_t (x_{t,m}) ) ||^2 / d, \quad  x_{0,m}, x_{t,m} \sim q(x_0,x_t).
> > > $$
> > >
> > > Thereby, each $\Phi_t$ requires $M$ function evaluations. To calculate $\Phi_t$ for all $t=1,\cdots,N$, we need a total of $MN$ function evaluations.
> > >
> > > As a result, the post-processing of DDPM also requires $MN$ function evaluations.

---

> > > > ### Comment · Reviewer_ejLx · 2021-11-19
> > > > **Response to the authors**
> > > >
> > > > I thank the authors again for clarifying: please see **post discussion period update** in my **Summary Of The Review**.

---

> > > > > ### Author Response · Authors · 2021-11-19
> > > > > **Thanks for the update**
> > > > >
> > > > > Thanks for the update. We highly appreciate it.

---

### Official Review · Reviewer_a5eg · 2021-10-30

**Correctness:** 4
**Technical Novelty And Significance:** 3
**Empirical Novelty And Significance:** 3
**Recommendation:** 8
**Confidence:** 4

**Main Review:**

Strengths:

1. The paper is clearly written and well organized. Contents are easy to follow. There are many technical details for readers to understand the results, such as the derivation of Theorem 1.

2. The analytic results are interesting and novel. According to the introduction and the related work sections, the optimal forms of the reverse process of DPM didn't appear in the previous DPM work and I believe it will help people in the field of DPM better understand this type of models. The bias analysis and the bound of the variance are helpful to understand the estimate and improve the estimate performance. The authors also make detailed discussions on these results, which are very helpful.

3. The experiment results are strong. The authors not only validate their analytic forms but also present the outperformance of their methods. The experiment results show significant improvements in their methods.

Weaknesses:

1. The DPM is a Gaussian-distribution-based simple model and similar analytic results can exist in similar models. The Gaussian model considered in this paper has a simple Markov property, which as a result has decomposable probability. The optimization is via forward KL divergence. Therefore, for example, the classical result on the expectation propagation algorithm with the Gaussian process can simplify the derivation: the decomposable form of the probability guarantees the forward KL decomposable, and minimizing the forward KL is equivalent to moment matching. So, I am concerned that the contribution is not as much as what was introduced in the paper. I also believe that such kind of connection will be helpful to figure out the possible further directions along this line and also save energy to get new results. I suggest authors add some discussions on the connection to other Gaussian models and their results.

**Summary Of The Paper:**

The paper studies an estimate of the reverse process of a diffusion probabilistic model (DPM). The reverse process is usually estimated by minimizing the KL divergence between the forward and the reverse process. Authors present that the optimal mean and variance of the reverse process have analytic forms w.r.t. the score function. The form of the optimal mean coincides with the parameterization in the previous work and justifies a reweighted variant of the variational bound proposed in the recent work. Different from the handcrafted strategies employed in other work, the authors propose a novel estimation for the variance and analyze its bias. To reduce the bias, bounds of the optimal reverse variance are analyzed and the estimate is clipped based on the bound. Furthermore, the KL divergence between the forward and the optimal reserve process also has an analytical form and as a result, the authors propose the optimal trajectory which has minimal KL value. Finally, the authors present the relationship between the score function and the data covariance matrix and assess the proposed approach in the experiments. The experiment results show that the analytic results improve the efficiency, likelihood, and sample quality.

**Summary Of The Review:**

The paper contributes interesting analytic results to the DPM models and the methods perform well in practice. However, the DPM is a simple Gaussian model and similar results can appear in other similar models.

---

> ### Author Response · Authors · 2021-11-15
> **Response to reviewer a5eg**
>
> We thank reviewer a5eg for the positive comments and valuable suggestions.
>
> ## Q1: On the connection to other Gaussian models and their results
>
> We appreciate the reviewer for the valuable comment. Below, we directly compare our theoretical results to existing work, especially expectation propagation (EP) with Gaussian process (GP) (e.g., [1*]).
>
> It is true that both EP and Analytic-DPM use moment matching as a key step to find analytic solutions of $KL(p_{target}||p_{opt})$ terms, and we provide a full proof of moment matching for completeness. However, to our knowledge, the connection of moment matching and DPMs has not been revealed in prior literature.
> Further, compared to EP, we emphasize that it is highly nontrivial to calculate the second moment of $p_{target}$ in DPMs because $p_{target}$ involves an unknown and potentially complicated data distribution.
>
>
> * In EP with GP (e.g., [1*]), $p_{target}$ is the product of a single likelihood factor and all other approximate factors for tractability. In fact, the form of the likelihood factor is chosen such that the first two moments of $p_{target}$ can be easily computed or approximated.
> For instance, the original EP [2*] considers Gaussian mixture likelihood (or Bernoulli likelihood for classification) and the moments can be directed obtained by the properties of Gaussian (or integration by parts). Besides, at the cost of the tractability, there is no converge guarantee of EP in general.
>
> * In contrast, $p_{target}$ in our paper is the conditional distribution $q(x_{n-1}|x_n)$ of the corresponding joint distribution $q(x_{0:N})$ defined by the forward process.
> Note that the moments of $q(x_{n-1}|x_n)$ are nontrivial to calculate because it involves an unknown and potentially complicated data distribution.
> Technically, in Lemma 13, we carefully use the law of total variance conditioned on $x_0$ and convert the second moment of $q(x_{n-1}|x_n)$ to that of $q(x_0|x_n)$,
> which can be expressed as the score function surprisingly as proven in Lemma 11. This is regarded as a novel and insightful contribution to the literature of DPMs by reviewers FHw9, Xtgn and Qdy8.
>
>
> We have revised Section 3 to emphasize our technical contributions and added a comparison to EP with GP in Appendix H.3.
>
>
> [1*] Hyun-Chul Kim and Zoubin Ghahramani. Bayesian gaussian process classification with the em-epalgorithm.
>
> [2*] Thomas Peter Minka. A family of algorithms for approximate Bayesian inference.

---

> > ### Comment · Reviewer_a5eg · 2021-11-20
> > **Thanks for your reply**
> >
> > Thanks for your replies to all reviews and I have read all of them. The Monte Carlo approximation in other reviews is very common in the Bayesian community when estimating a term that has no closed-form expression and I have no concern about it. The above reply is helpful and I realize that the specific study on the conditional Gaussian with a more complicated mean is non-trivial and can be very helpful for a series of models. As other reviewers point out, this paper is novel in the community of DPMs, so I upgrade my recommendation. Also, I would keep suggesting that it is helpful to discuss some Bayesian works which also rely on similar measures between p_target and p_opt and obtain analytic results. Please also take the related works about the Gaussian Markov process into account, because it is very related to DPMs. The difference or similarities will be helpful for future works along this line.

---

> > > ### Author Response · Authors · 2021-11-20
> > > **Thanks for the update!**
> > >
> > > Thank you very much for the valuable suggestions and the update on the score. We highly appreciate it.

---

### Official Review · Reviewer_Qdy8 · 2021-11-03

**Correctness:** 3
**Technical Novelty And Significance:** 4
**Empirical Novelty And Significance:** 3
**Recommendation:** 8
**Confidence:** 3

**Details Of Ethics Concerns:**

Performant generative models can be misused in situations such as deep fake.

**Main Review:**

Strength:
1. the paper derives the optimal reverse variance for diffusion probabilistic models as well as its lower and upper bound. This leads to more efficient DPMs with better performance compared to existing DPM variants. It also leads to new insights into DPMs such as why the way the reversed variance is chosen by existing work is not ideal.

2. The paper is clearly presented. The technical results reported in the paper are non-trivially obtained. The relationship between the paper and existing works is well discussed and well-motivated.

3. Experimental results compared to other variants of DPM are well discussed. It also demonstrates the advantage of the proposed method compared to existing DPM variants.

Weakness:
1. Although proposition 1 establishes the relationship between data covariance and score function, it is unclear to me what is the practical implication of proposition 1.

2. In the experiment, it is unclear to me whether timestep is a good metric to measure efficiency in Table 3. Does each method spend roughly the same time at each timestep?

3. While the authors compare the proposed method with other existing variants of DPMs. Is there any reason why the comparison between the proposed method and other classes of generative models such as GAN should be conducted or not?



**Summary Of The Paper:**

The paper proposed a modification of the diffusion probabilistic models (DPMs) called analytic-DPM that is based on an analytic estimate of the optimal reverse variance. Using this analytic estimate, the proposed method can achieve fast and performant inference through the Monte Carlo method and pre-trained score-based model. The derivation of the optimal reverse analytic mean and variance is proven to be associated with the score function. Upper bounds and lower bounds are provided for the optimal reverse variance, the relationship between the data covariance and the score function is shown. Experimental results are provided by comparing the proposed method with existing variants of DPMs, through both negative log-likelihood and FID as metrics. The experimental results suggest that the proposed method can potentially provide better performance more efficiently compared to alternatives.

**Summary Of The Review:**

Correctness: I think the paper is mostly correct. I am not sure if using timestep is a good metric to demonstrate the efficiency of the proposed method. Since efficiency is a major aspect of the proposed method, it would be desirable to make a clarification on this issue.

Novelty and significance: I think the derivation of the optimal reverse variance is novel and insightful. The lower bound and upper bound of the optimal reverse variance is also useful in practice. While the proposed method is performant compared to existing DPM variants, these existing variants achieve better or comparable FID or negative log-likelihood with enough timesteps. These strong baselines suggest the (potentially limited) headroom left for improvement for the proposed method.

Overall, I think the paper solves an interesting problem in DPM.

---

> ### Author Response · Authors · 2021-11-15
> **Response to reviewer Qdy8**
>
> We thank reviewer Qdy8 for the acknowledgement of our contributions and the valuable comments.
>
> ## Q1: Practical implication of Proposition 1
>
> Thanks for the question. Currently, Proposition 1 is purely theoretical and its practical implication is unclear. We have clarified this in Section 5 in the revised version.
>
>
> ## Q2: Whether timestep is a good metric to measure efficiency in Table 3
>
> Since the score-based models are nearly the same for all methods compared in Table 3, it is natural to compare the number of timesteps, or equivalently the number of model function evaluations, for efficiency. In fact, we have validated that the averaged time of a single model function evaluation in compared methods are almost the same, as presented in Appendix F.5 (Table 4) in the revised version. Also see our response to common concern 2 for more discussion about the efficiency.
>
> ## Q3: Should the comparison to other classes of generative models be conducted or not
>
> Thanks for the suggestion. The primary focus of our work is on improving the performance and efficiency of DPMs. Thereby, DPMs and their variants serve as the most direct baselines to validate the effectiveness of the proposed method. Despite this, we have added a new table (Table 10) in Appendix G.8 of the updated version to compare with other classes of generative models including GAN, VAE, Flow and EBM. As shown in Table 10, Analytic-DPM achieves competitive sample quality results among various generative models, and meanwhile significantly reduces the efficiency gap between DPMs and other models.

---

### Official Review · Reviewer_Xtgn · 2021-11-05

**Correctness:** 4
**Technical Novelty And Significance:** 3
**Empirical Novelty And Significance:** 3
**Recommendation:** 8
**Confidence:** 3

**Main Review:**

**Strengths**
* Strong theoretical motivation and strong empirical results to support it.
* Nicely written, does a great job putting prior work in the context of the new insights on optimal reverse mean and variance.
* Despite the abundance of theory / derivations, the paper still remains accessible.

**Weaknesses**

I am convinced by the paper in its current form. But if I had to list something:
* It would be great to see applications of this method to other data modalities, such as for example speech / sound generation.
* The performance of Analytical-DPMs on methods that learn forward process variance schedules (e.g. VDMs) would also be great to see.

**Summary Of The Paper:**

The paper proposes a theoretically grounded method for estimating *optimal* reverse process variances for DDPMs and DDIMs. This method can be applied to a trained DDPM / DDIM after the fact and lead to improved likelihoods and faster sampling (when used together with optimal trajectory search). The proposed method and theoretical insights also perform strongly in practice across a range of models and datasets.

**Summary Of The Review:**

The paper provides valuable insights into optimal reverse process variance of DDPMs and DDIMs, and makes connections between the proposed optimal variance and previous handcrafted choices, etc. The improved understanding of these model classes could have been sufficient to recommend acceptance. However, the empirical results, especially those around faster sampling are also strong and convincing. DDPMs / DDIMs achieve high sample quality and it's primarily their sampling speed that prevents practical application off this model class in real-world systems. This works makes a significant step towards enabling faster sampling for this model class.

---

> ### Author Response · Authors · 2021-11-15
> **Response to reviewer Xtgn**
>
> We thank reviewer Xtgn for the acknowledgement of our contributions and the valuable comments.
>
> ## Q1: Application of this method to other data modalities
>
>
> Thanks for the suggestion. It would be interesting to apply Analytic-DPM to other data modalities, e.g. speech data [1*]. We leave it for future work and have added a discussion in Appendix H.4.
>
> ## Q2: Performance of Analytic-DPMs on methods that learn forward process variance schedules
>
> Thanks for the suggestion. As presented in Appendix E, our method can be applied to VDMs and we're trying to reproduce VDMs. We leave it for future work.
>
> [1*] Nanxin Chen, Yu Zhang, Heiga Zen, Ron J Weiss, Mohammad Norouzi, and William Chan. Wave-grad: Estimating gradients for waveform generation.

---

### Official Review · Reviewer_FHw9 · 2021-11-06

**Correctness:** 3
**Technical Novelty And Significance:** 3
**Empirical Novelty And Significance:** 3
**Recommendation:** 8
**Confidence:** 4

**Main Review:**

**novelty, significance**

The result on optimal variance is new, to the best of my knowledge. The authors also do a reasonable job in convincing me that a Monte Carlo estimate of it is useful for inference. Given the recent progress and interest on DPMs, I think this work will also have reasonable significance and influence.

**presentation**

Authors do a reasonable job on the writing and presenting experimental results. Past works are adequately and appropriately cited, to the best of my knowledge.

**Technical quality and correctness**

One thing I'll add here is that once the Monte Carlo estimate of $\sigma_t^2$ is plugged into the bound computation, it seems we end up with a stochastic lower bound of the ELBO (assuming the loss is concave in $\sigma_t^2$). The important thing here to note perhaps is that bias is introduced. To put it more concretely, say the quantity being estimated is $\mathbb{E}[ f(\sigma_t^2) ]$, where I've written the bound as the expectation of the loss $f$ as a function of $\sigma_t^2$. The estimator $f(\hat{\sigma}_t^2)$ is now a stochastic lower bound on the original quantity by Jensen's, since
$$\mathbb{E} [f(\hat{\sigma}_t^2)] \le \mathbb{E} [f( \mathbb{E} [ \hat{\sigma}_t^2 ] ) ]  = \mathbb{E} [f( \sigma_t^2)].$$

I think there should be some discussion about this. I'm assuming $f$ is concave in $\sigma_t^2$, mostly reasoning from past bounds, but authors should perhaps make parts of the discussion more precise.

**Experimental results**

Authors do a reasonable job in evaluating their method. One particular point I didn't get is how $M$ (number of samples for estimating the expected score) is chosen. The are a couple of potential issues here.
- Selecting $M$ requires additional hyperparameter tuning, potentially; the tuning procedure should be reported.
- How results depend on $M$ isn't entirely clear just from reading the main text (maybe there's some discussion in the appendix, but I didn't have time to read all content in the appendix). Ideally, some discussions should appear in the main text.
- Large $M$ incurs more compute cost during inference -- while this seems less an issue when inference is run on GPUs (since most scenarios, I'd guess, there's enough cores to parallelize the Monte Carlo samples), this could be an issue for CPU inference. How does the run-time in practice compare in this case? Note while practical systems don't tend to run training on CPUs, inference on CPUs is still quite common.

**Summary Of The Paper:**

The paper studies diffusion probabilistic models, and derives the optimal mean and variance (as functions of the expected data score) for the reverse process. Authors then propose to plug in a Monte Carlo estimate of the the variance for the reverse process and experimentally show how this leads to improved results with trained and/or pretrained models in terms of FID and NLL. In addition, authors combine their approach with recent work that optimizes for "knot" locations given a fixed number of knots for faster sampling.

**Summary Of The Review:**

Authors study choosing the optimal variance for the reverse process in DPMs and propose to Monte Carlo estimate it for improved inference.  Technical quality, writing, and experiments are mostly good with the two minor caveats I described above.

---

> ### Author Response · Authors · 2021-11-15
> **Response to reviewer FHw9**
>
> We thank reviewer FHw9 for the acknowledgement of our contributions and the insightful comments.
> We have updated our paper by adding a discussion about the bias caused by the Monte Carlo method, and clarifying the issues about the tuning procedure of $M$, sensitivity of $M$ and the additional computation cost.
> Below, we provided a point-to-point response to all comments.
>
>
> ## Q1: $f(\hat{\sigma}_t^2)$ as a stochastic lower bound of $\mathbb{E}[f(\sigma_t^2)]$ and the potential bias
>
> Thanks for the valuable comment. Generally, $f(\hat{\sigma}_t^2)$ is not a stochastic lower bound of $\mathbb{E}[f(\sigma_t^2)]$, since $f$ is not concave in $\sigma_t^2$. Actually, the ELBO can be written as a function of $\sigma_t^2$ in the form of $f(\sigma_t^2) = B(\frac{A}{\sigma_t^2} + \log \sigma_t^2) + C$, where $A, B, C$ are constants unrelated to $\sigma_t^2$ and $A, B > 0$. $f(\sigma_t^2)$ is convex when $0 < \sigma_t^2 < 2A$ and concave when $2A < \sigma_t^2$. Please refer to Proposition 2 of Appendix H.2 in the revised version for a formal proof.
>
> However, in this paper,
> $f(\hat{\sigma}_t^2)$ is a stochastic lower bound of $\mathbb{E}[f(\sigma_t^{*2})]$ because $\mathbb{E}[f(\sigma_t^{*2})]$ is the optimal ELBO.
> Plugging in $\hat{\sigma}_t^2$, the bias of $\mathbb{E}[f(\hat{\sigma}_t^2)]$ w.r.t. $\mathbb{E}[f( \sigma_t^{*2})]$ is due to the Monte Carlo method as well as the error of the score-based model.
> The former can be reduced by increasing the number of Monte Carlo samples. The latter is irreducible if the pretrained model is fixed, which motivates us to clip the estimate, as discussed in Section 3.1. We have revised the main text in Section 3 and added a detailed discussion in Appendix H.2 of the updated version.
>
>
> ## Q2: The tuning procedure of $M$
>
> Thanks for the suggestion. We do not tune $M$ in the original paper. In fact, we use a maximal $M$ without introducing too much computation.
> See more details in the common concern 1 (1.1).
>
> ## Q3: How results depend on $M$
>
> Thanks for the suggestion. Based on the original results of varying $M$ in Appendix G.2 and newly added experiments in Table 5 of Appendix G.2 (or see Table 1* in the common concern 1 (1.2)), we conclude that Analytic-DPM is not sensitive to $M$ and usually a small $M$ (e.g. $M=10$) is sufficient for good results.
>
> ## Q4: Large $M$ incurs more computation cost during inference
>
> Thanks for the valuable comment. As mentioned in the response to Q3, usually a small $M$ is sufficient for accurate inference. Suppose that we have to use a large $M$ in the setting mentioned in the comment (i.e., inference using CPUs). Given a pretrained model and training dataset, we can calculate $\Gamma = (\Gamma_1, \cdots, \Gamma_N)$
> offline (i.e., on GPUs) and deploy it together with the pretrained model. Consequently, the online inference cost of Analytic-DPM is exactly the same as DPM. In fact, in the paper, we calculate $\Gamma$ first and reuse it throughout our experiments (e.g., over different selections of $K$, trajectories and forward processes).
> Please see more details in our response to the common concern 2.

---

> > ### Comment · Reviewer_FHw9 · 2021-11-15
> > **Thanks**
> >
> > Thanks for the updates. I read the author's response and will keep my recommendation.

---

### Author Response · Authors · 2021-11-15
**Common concern 1**

We thank all the reviewers for their appreciation of our novel contributions as well as the valuable comments, which help to further improve. Below, we first address some common concerns. Then, we address the individual comments to each reviewer.

## Common concern 1 (from reviewers FHw9, ejLx): The tuning procedure of $M$ and more experiments on varying $M$

We clarify how $M$ is selected in the original paper and add more experiments on varying $M$.

### 1.1 The tuning procedure of $M$ in the original paper

We did not tune $M$ in the original paper. In fact, we used a maximal $M$ without introducing too much computation. Specifically, we set $M=50,000$ on CIFAR10, $M=10,000$ on CelebA 64x64 and ImageNet 64x64 and $M=1,000$ on LSUN Bedroom by default without a sweep.
All of the samples are from the training dataset. We used the default settings of $M$ for all results in Tables 1, 2 and 3. We have added the experimental details in Appendix F.3 in the revised version.


### 1.2. How results depend on $M$

In Appendix G.2 of the original submission, we evaluated Analytic-DPM with $M=100$ and found that it has a small variance and achieves  almost the same NLL results as those of $M=50,000$.

Following the reviewers' suggestion, we added the results of Analytic-DPM over a wide range of $M \in \\{1,3,10,100,10000,50000\\}$.
Under both the NLL and FID metrics, $M=10$ achieves similar results as those of $M=50,000$ on CIFAR10 (LS) and those of $M=10,000$ on ImageNet 64x64. The results are presented in Table 1* below.
We have also added the results to Appendix G.2 in Table 5 in the revised version.


| CIFAR10 (LS) | NLL $\downarrow$ | FID $\downarrow$ |  ImageNet 64x64 | NLL $\downarrow$ | FID $\downarrow$ |
|  ----        | ----        | ----        |  ----          | ----       | ----        |
| $M=1$        | 6.220±1.126 | 34.05±4.97 | $M=1$          | 4.943±0.162 | 31.59±5.11 |
| $M=3$        | 5.689±0.424 | 34.29±2.88 | $M=3$          | 4.821±0.055 | 31.98±1.19 |
| $M=10$       | 5.469±0.005 | 33.69±2.10 | $M=10$         | 4.791±0.017 | 31.93±1.02 |
| $M=100$      | 5.468±0.004 | 34.63±0.68 | $M=100$        | 4.785±0.003 | 31.93±0.69 |
| $M=50000$    | 5.471       | 34.26      | $M=10000$      | 4.783       | 32.56      |

Table 1*: FID and NLL results of Analytic-DPM with different $M$. We use $K=10$ for CIFAR10 (LS) and $K=25$ for ImageNet 64x64.


### 1.3. The variance of the estimate $\Gamma_n$ over different $M$

We plotted the standard deviations of the estimate $\Gamma_n$ at different timesteps $n$ when $M\in \\{1,10,100\\}$.
In all cases, the variance decays fast as $n$ increases. Further, $M=10$ is sufficient to achieve a small standard deviation relative to the mean (e.g. less than 10\% of the mean) for all $n$.
See Figures 4\&5 in Appendix G.2 of the revised version for the results.

---

### Author Response · Authors · 2021-11-15
**Common concern 2**


## Common concern 2 (from reviewers FHw9, ejLx): The extra cost of the Monte Carlo estimate $\Gamma_n$

* The extra cost of the Monte Carlo estimate is small compared to the whole inference cost.
In fact, the Monte Carlo estimate requires $M N$ additional function evaluations.
During inference, suppose we generate $M_1$ samples or calculate the log-likelihood of $M_1$ samples with $K$ timesteps. Both DPMs and Analytic-DPMs need $M_1 K$ function evaluations. Employing the same score-based models, the relative additional cost of Analytic-DPM is $\frac{M N}{M_1 K}$. In our experiments, we found that a very small $M$ (e.g., $M=10$ or $100$) is sufficient for Analytic-DPM (see our response to common concern 1 (1.2 and 1.3)), making the relative additional cost small if not negligible. For instance, on CIFAR10, let $M=10$, $N=1000$, $M_1=50000$ and $K\ge 10$, we obtain $\frac{M N}{M_1 K}\le 0.02$ and  Analytic-DPM still consistently improves the baselines as presented in Table 2* below. We have also added the results to Appendix G.2 in Table 6 in the revised version.


* Further, the additional calculation of the Monte Carlo estimate occurs only **once** given a pretrained model and training dataset, since we can save the results of $\Gamma = (\Gamma_1, \cdots, \Gamma_N)$ in Eq.(8) and reuse it among different inference settings (e.g., trajectories of various $K$). The reuse is valid, because the marginal distribution of the shorter forward process $q(x_0, x_{\tau_1}, \cdots, x_{\tau_K})$ at timestep $\tau_k$ is the same as that of the full-timesteps forward process $q(x_{0:N})$ at timestep $n=\tau_k$. Indeed, in our experiments (e.g., Table 1,2), $\Gamma$ is shared across different selections of $K$, trajectories and forward processes. Moreover, in practice, $\Gamma$ can be calculated offline and deployed together with the pretrained model and the online inference cost of Analytic-DPM is exactly the same as DPM.

We have updated Section 3 and added Appendix H.1 in the revised version to address this comment.


| # timesteps $K$ | 10 |  25 | 50 | 100 |  200 | 400 |
| ---- | ---- | ---- | ---- | ---- | ---- | ---- |
| NLL $\downarrow$ |
| Analytic-DDPM ($M=10$) | 5.47 | 4.80 | 4.38 | 4.07 | 3.85 | 3.71 |
| DDPM        | 6.99 | 6.11 | 5.44 | 4.86 | 4.39 | 4.07 |
| FID $\downarrow$ |
| Analytic-DDPM ($M=10$) | 33.69 | 11.99 | 7.24 | 5.39 | 4.19 | 3.58 |
| DDPM        | 44.45 | 21.83 | 15.21 | 10.94 | 8.23 | 4.86 |

Table 2*: The NLL and FID comparison between Analytic-DDPM with $M=10$ Monte Carlo samples and DDPM on CIFAR10 (LS).

---

### Author Response · Authors · 2021-11-15
**Summary of the revision**

We sincerely thank all the reviewers for the valuable comments, which help to improve the quality of our work. We summarize the revision in the updated version as follows:
* We revised Section 3 to emphasize our technical contributions
* We moved Proposition 1 to Appendix A.5, and added more discussion of it in Section 5
* We moved some experimental details to the main paper (see Section 6) and added missing ones (see Appendix F.3\&F.5)
* We added more experiments on the number of Monte Carlo samples (see Appendix G.2)
* We added more experiments on the bounds of Theorem 2 (see Appendix G.3)
* We showed values of clipping thresholds at $n=2$ (see Table 7)
* We added a comparison to other classes of generative models (see Appendix G.8)
* We added discussion on the extra cost of the Monte Carlo estimate (see Appendix H.1)
* We added discussion on the stochasticity of the variational bound (see Appendix H.2)
* We added a comparison to other Gaussian models (see Appendix H.3)
* We added discussion on future works (see Appendix H.4)

---

### Decision · Program_Chairs · 2022-01-20

**Decision:**

Accept (Oral)

**Comment:**

This paper presents an analytic approach for estimating the optimal reverse variance schedule given a pre-trained score-based model. The experimental results demonstrated the efficacy of the proposed method on several datasets across different sampling budgets. Given the recent interest in score-based generative models, I believe that the paper will find applications in various domains. I am pleased to recommend it for acceptance.